# Electric pulse-tuned piezotronic effect for interface engineering

Qiuhong Yu [1,2,7], Rui Ge [1,3,7], Juan Wen [1], Qi Xu [1], Zhouguang Lu[4], Shuhai Liu [1] ✉ & Yong Qin [1,5,6] ✉

Investigating interface engineering by piezoelectric, flexoelectric and ferro-electric polarizations in semiconductor devices is important for their applications in electronics, optoelectronics, catalysis and many more. The interface engineering by polarizations strongly depends on the property of interface barrier. However, the fixed value and uncontrollability of interface barrier once it is constructed limit the performance and application scenarios of interface engineering by polarizations. Here, we report a strategy of tuning piezotronic effect (interface barrier and transport controlled by piezoelectric polarization) reversibly and accurately by electric pulse. Our results show that for Ag/HfO$_2$/ $n$-ZnO piezotronic tunneling junction, the interface barrier height can be reversibly tuned as high as 168.11 meV by electric pulse, and the strain (0–1.34‰) modulated current range by piezotronic effect can be switched from 0–18 nA to 44–72 nA. Moreover, piezotronic modification on interface barrier tuned by electric pulse can be up to 148.81 meV under a strain of 1.34‰, which can totally switch the piezotronic performance of the electronics. This study provides opportunities to achieve reversible control of piezotronics, and extend them to a wider range of scenarios and be better suitable for micro/ nano-electromechanical systems.

Ever since the wide range applications of functional nanodevices, investigation of interface physics and interface engineering in semiconductors[1] for tunable electronics, optoelectronics and bioelectronics is becoming increasingly important[2–4]. Recently, electronics fabricated by utilizing piezoelectric, flexoelectric, ferroelectric polarization potential as a gate voltage to tune/control the carrier transport behavior of interface[5] is of particular interest and, respectively, named piezotronics[6–9], flexoelectronics[10,11], and ferroelectric electronics[12–14], which are distinctly different from those achieved by traditional CMOS technologies in principle, design, and applications. These emerging types of interface engineering by local polarization have triggered plenty of new applications and attracted worldwide attention as a promising approach for human-CMOS interfacing[9,15,16], high-performance memory[12,13], next generation of sensor and transducers[6,7], and many more.

The fundamental principle of these emerging sciences stems from regulations of interface barrier by local polarization[17], that can establish the relationship between signals such as external stimulus and the interface electrical characteristics. Figure 1a illustrates the typical piezotronics, and the corresponding energy bands at interfaces modulated by strain-induced piezoelectric polarization. When a strain is applied on the piezoelectric semiconductor, piezoelectric charges and the corresponding piezoelectric potential will appear at the interface, which will bend the energy band and thus control the

[1]Institute of Nanoscience and Nanotechnology, School of Materials and Energy, Lanzhou University, Lanzhou, Gansu, China. [2]Henan Key Laboratory of Photoelectric Energy Storage Materials and Applications, School of Physics and Engineering, Henan University of Science and Technology, Luoyang, Henan, China. [3]School of Advanced Materials and Nanotechnology, Xidian University, Xi'an, Shaanxi, China. [4]Department of Materials Science and Engineering, Southern University of Science and Technology, Shenzhen, Guangdong, China. [5]MIIT Key Laboratory of Complex-field Intelligent Exploration, Beijing Institute of Technology, Beijing, China. [6]Advanced Research Institute of Multidisciplinary Sciences, Beijing Institute of Technology, Beijing, China. [7]These authors contributed equally: Qiuhong Yu, Rui Ge. ✉e-mail: liushuhai1991@live.cn; qinyong@lzu.edu.cn

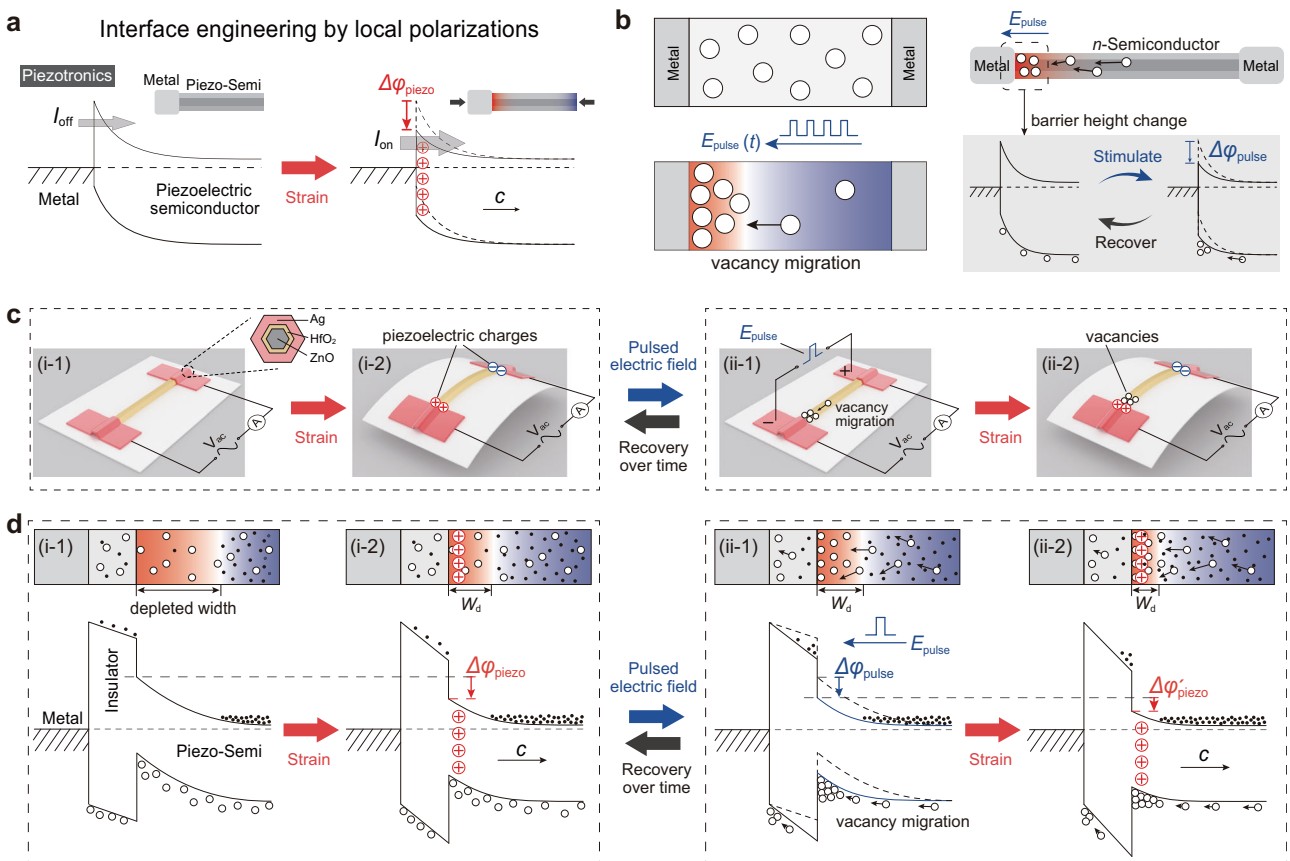

**Fig. 1 | Working principle of the electric pulse-tuned piezotronic effect on Ag/HfO₂/n-ZnO piezotronic tunneling junction (PTJ). a** Energy band diagrams of interface engineering by local polarization in piezotronics. **b** Effect of electric pulse on the oxygen vacancies in semiconductors (left) and the corresponding modification of interface barrier height (Δφ_pulse) (right). **c** Introducing a strain (i) and an electric pulse (ii) to the Ag/HfO₂/n-ZnO PTJ to tune the interface barrier. **d** Corresponding energy bands of Ag/HfO₂/n-ZnO PTJ in (**c**). (i) The modulation of

interface barrier by piezoelectric polarization ranges from $\varphi_0/2$ to $(\varphi_0 - \Delta\varphi_{piezo})/2$, where $\varphi_0/2$ is the initial interface barrier height. (ii) The electric pulse-induced oxygen vacancy migration changes the interface barrier by $\Delta\varphi_{pulse}/2$, and thus tune the piezotronic modulation range from $(\varphi_0 - \Delta\varphi_{pulse})/2$ to $(\varphi_0 - \Delta\varphi_{pulse} - \Delta\varphi'_{piezo})/2$. The red color and blue color in (**b**) and (**d**) represent positive potential and negative potential, respectively.

electrical transport of the interface. As can be seen, the construction of interface barrier is the key in achieving interface engineering by local polarization[3]. Generally, the height of the interface barrier is a core parameter that must be carefully designed in these tunable electronics[18–20], because it directly determines the current and its tunable range, thus their application scenarios. High interface barrier usually results in a lower working current with relatively small tunable range but remarkable sensitivity, whereas low interface barrier means higher working current with a big tunable range but greater shielding effect of free carriers on polarization charges, leading to a weakened interface engineering[3]. Until now, in the studies of regulating interface barrier by local polarization, the interface barrier in electronics is completely fixed and cannot be changed once it is constructed[20]. Finding strategies to tune the interface barrier thus the current and its tunable range to be suitable for various application scenario has important scientific and engineering significance in tunable electronics including piezotronics, flexoelectronics, and ferroelectric electronics.

Interface barrier is determined by the work function of metal, and the electron affinity together with the carrier concentration of semiconductor[20]. In order to control the interface barrier without change the constructed interface structure, great effects have been made to tune the carrier concentration of semiconductors. For example, material doping/alloying[20,21], surface modification[20], and internal nano-pores' constructing[22] can well control the carrier

concentration, but these methods are usually irreversible and difficult to accurately control. Most recently, utilizing asymmetric electric field is proved to be an effective approach to reversibly tune the interface characteristics (e.g., interface symmetry[23] and vacancy distribution/concentration[24]). These findings provide the possibility of reversibly and accurately regulating the interface properties to tune the interface barrier and current, which shows potential in achieving tunable piezotronics, flexoelectronics, and ferroelectric electronics.

Here, we demonstrate an electric pulse-tuned piezotronic effect, showing that the interface barrier and hence the interface engineering by piezoelectric polarization can be reversibly and accurately regulated by the electric pulse-induced rearrangement of vacancies. For Ag/HfO₂/n-ZnO piezotronic tunneling junction, the interface barrier height can be reversibly tuned by electric pulse up to 168.11 meV, corresponding to a change of about 643 times in the working current and internal resistance. The piezotronic modification range of working current can be switched from 0.068–18.3 nA to 43.7–71.9 nA (corresponding to strain from 0.00‰ to −1.34‰) by electric pulse of 80 V, and recovers in 35 minutes. Moreover, the piezotronic modification on interface barrier can also be tuned by electric pulse up to 148.81 meV under a compressive strain of 1.34‰. These findings give an in-depth understanding for the coupling effect between the electric field-induced redistribution of vacancies and local polarization-controlled interface engineering, and provide the opportunities to achieve reversible control of piezotronics.

## Results

### Working principle

As illustrated in Fig. 1b (left), vacancies in semiconductors will migrate along the applied electric pulse. Taking metal/*n*-semiconductor/metal structure as an example, the electric pulse-induced rearrangement of vacancies leads to asymmetric charge distribution and corresponding potential distribution, which in turn will change the height of the interface barrier formed between the metal and the *n*-semiconductor by $\Delta\varphi_{pulse}$ (right side of Fig. 1b). This regulation will gradually recover to the original state over time. In this way, we can use electric pulse to tune the height of interface barrier without breaking the interface that has already been constructed.

Based on the above principle, we fabricated an Ag/HfO$_2$/*n*-ZnO piezotronic tunneling junction[6] (PTJ) as a demonstration illustrated in Fig. 1c, since the piezotronic tunneling junction has the more excellent piezoelectric polarization regulation performance compared with the Schottky junction (Supplementary Note 1). Here, we choose HfO$_2$ with a thickness of about 1.8 nm as the insulation layer and Ag as the electrode to fabricate the Ag/HfO$_2$/*n*-ZnO PTJ for this structure possessing good performance on piezotronic modification of electrical transport (Supplementary Note 2, and Supplementary Figs. 1–4 and Supplementary Table 1). Material characterizations (Supplementary Note 3 and Supplementary Figs. 6–8) show that the ZnO nanowire possesses a typical hexagonal geometry with clean surface and good wurtzite crystallinity, and the interface formed between HfO$_2$ and ZnO has few interface traps. The piezoelectricity and polarization *c*-axis in wurtzite ZnO can be measured through the piezoresponse force microscopy (PFM) and the piezoelectric nanogenerator (PENG) measurement, respectively (Supplementary Note 4, 5 and Supplementary Figs. 9, 10). As bending the substrate, a strain will be induced along ZnO nanowire to make it produce opposite piezoelectric charges at both terminals of nanowire (Fig. 1c-i) that acts as a gate voltage to control the charge transport of interface (piezotronic effect[3]). As shown in Fig. 1c-ii, with electric pulse stimulation, oxygen vacancies migration occurs in ZnO nanowire to tune the barrier height at the tunneling interface, which will then determine the range of working current modulated by piezotronic effect.

The corresponding energy bands of Ag/HfO$_2$/*n*-ZnO tunneling junction are schematically depicted in Fig. 1d. If the piezoelectric polar *c*-axis points into the piezoelectric semiconductor, the tunneling junction under a compressive strain can utilize the strain-induced positive piezoelectric charge and the corresponding piezoelectric potential at the interface to lower the barrier height (by $\Delta\varphi_{piezo}/2$) and the barrier width in parallel by making the surface of semiconductor less depleted (Fig. 1d-i), and hence synergistically modulate the electrical transport of interface[6] (Supplementary Note 6 and Supplementary Figs. 11 and 12). This synergistically modulation makes the Ag/HfO$_2$/*n*-ZnO tunneling junction with a high strain sensitivity, which is the advantage of using Ag/HfO$_2$/*n*-ZnO tunneling junction here. What needs attention is that the modulation of interface barrier by piezoelectric polarization ranges from $\varphi_0/2$ to $(\varphi_0 - \Delta\varphi_{piezo})/2$, where $\varphi_0/2$ is the initial interface barrier height. But for a particular tunneling junction, once it is fabricated, the interface barrier is fixed and can't be tuned, it is more impossible to be continuously tuned. This initially fixed barrier height and uncontrollability are not beneficial to expand the performance and application scenarios of piezotronics. As shown in Fig. 1d-ii, under the driving force induced by electric pulse (pointing to the left), oxygen vacancies will migrate toward the interface from the inside of semiconductors and then aggregate at the interface. Because the oxygen vacancies are positively ionized, the aggregation of vacancies at the interface will initially lower the interface barrier by $\Delta\varphi_{pulse}/2$, and thus tune the piezotronic modulation range from $(\varphi_0 - \Delta\varphi_{pulse})/2$ to $(\varphi_0 - \Delta\varphi_{pulse} - \Delta\varphi'_{piezo})/2$, which can control the working current range of PTJ. The key working principle here lies in utilizing electric pulse stimulation to achieve tunable barrier of

tunneling junction and thus regulate the piezotronic effect in the tunneling junction (Supplementary Note 7 and Supplementary Fig. 13).

### Electric pulse-tuned interface barrier

In order to clarify the influence of electric pulse on the interface barrier, we carried out electrical transport and surface potential measurements in Fig. 2. A set procedure was used to perform the electric pulse stimulation on PTJ device, and the current signals was captured by electrical measurements (Supplementary Note 8 and Supplementary Figs. 14 and 15). As shown in Fig. 2a, b, the current flowing through the device is gradually enhanced with the increase of the electric pulse voltage, and shows a trend of attenuation with time. Through the linear fitting analysis (Fig. 2c) and the recovery ratio plotting (Fig. 2d) of the current with time in the period numbered 1 to 5 in Fig. 2b and Supplementary Fig. 16, we found that the recovery rate of current gradually decreases with the increasing electric pulse voltage. In other words, the current will become larger and last longer with a higher electric pulse voltage. Some other ways, such as proper annealing of piezoelectric semiconductors (e.g., *n*-ZnO), may be also able to achieve a longer retention (Supplementary Note 9 and Supplementary Fig. 17).

In Fig. 2e, we studied the electrical transport of the tunneling junction after electric pulse (+80 V) of various durations with a frequency of 10 Hz (Supplementary Fig. 18). It need to be mentioned that the periodic pulse voltage of 80 V will not damage the tunneling junction, which still work normally after the pulse voltage stimulation (Supplementary Note 10 and Supplementary Figs. 19 and 20). It can be seen from Fig. 2e and Supplementary Fig. 22a that the current gradually increases to a stable value as the stimulation time increases, implying that the interface barrier is indeed tuned by the applied electric pulse. This tunable interface barrier is also verified by an in situ measurement utilizing Atomic Force Microscope (AFM) (Supplementary Note 11 and Supplementary Fig. 23). Under electric pulse with different voltage, the plotting of current over time (Fig. 2f) shows that the higher the electric pulse, the larger the tuned current and the slower its recovery, which is consistent with the previous analysis in Fig. 1c, d and the results in Supplementary Fig. 22b, c. The increased current means a decreased interface barrier, which attributes to the oxygen vacancy migration driven by the electric pulse. As the electric pulse is stopped, the current gradually decreases to its original magnitude due to the diffusion of oxygen vacancies back to its original state. Moreover, by increasing the pulse frequency (cycle number per second) and the pulse duration, we can further improve the impact of electric pulse on the current response and increase the recovery time (Supplementary Note 10 and Supplementary Fig. 21). It is worth noting that we used sweeping voltages of ±1 V here and in subsequent *I–V* characteristics, because higher sweeping voltage may degrade the piezotronic effect and cause the damage of tunneling junction (Supplementary Note 12 and Supplementary Figs. 24 and 25).

Additionally, we also measured the surface potential of ZnO near the interface in situ by Kelvin Probe Force Microscopy (KPFM) over time just after an electric pulse (Supplementary Note 13). Figure 2g and Supplementary Fig. 26a exhibit the evolutions of surface potential distribution of ZnO at the same position at different time. Through a statistical analysis of the surface potential of ZnO (Fig. 2h) and substrate (Fig. 2i) (details in Supplementary Figs. 26–28), we can find that both fit to normal distribution and that of the substrate is stable near zero, indicating the reliability of KPFM measurements. By subtracting the surface potential of ZnO and that of substrate, the net surface potential of ZnO (in reference to the substrate) is found to decrease gradually with time. The measured surface potential corresponds to the electron energy difference between conduction band bottom and vacuum level[20]. A decreasing surface potential means an increasing conduction band bottom in energy profile and an increasing concentration of positive charges near the interface

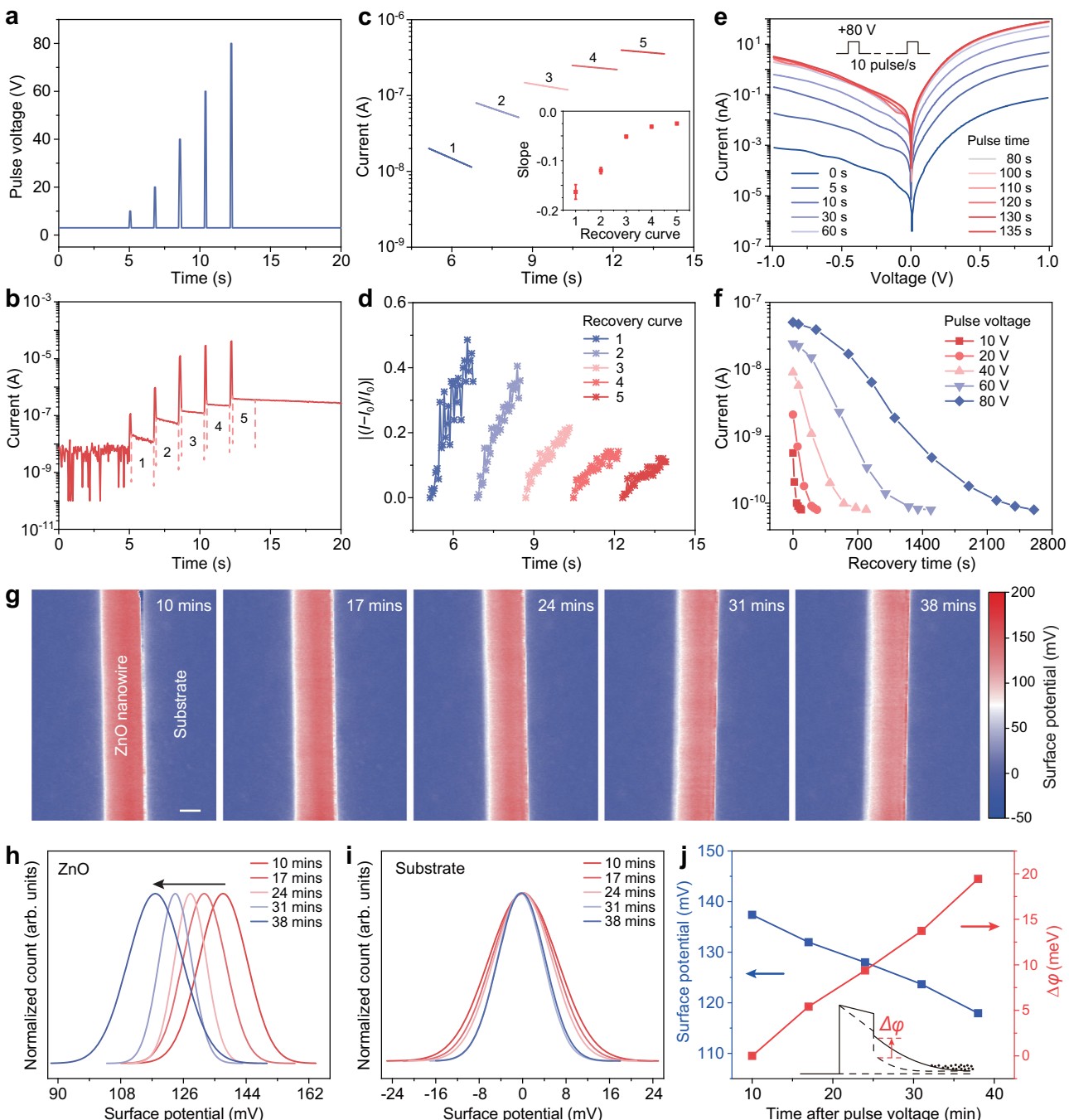

**Fig. 2 | Electric pulse-tuned interface barrier. a** Electric pulse voltage of 10, 20, 40, 60, and 80 V on Ag/HfO₂/*n*-ZnO PTJ. The duration of each pulse is 0.05 s, and the time interval between two electric pulses is about 1.59 s. **b** Current response corresponding to (**a**). **c**, **d** Linear fitting and recovery ratio plotting of the five recovery curves numbered 1 to 5 in (**b**). Inset of (**c**) shows the slope corresponding to the recovery curve. **e** *I*–*V* characteristics of PTJ after electric pulses of various durations. **f** Plotting of the current over time after electric pulses of different voltage. **g** Surface potential of ZnO near the interface in situ by Kelvin Probe Force Microscopy (KPFM) over time just after an electric pulse. **h**, **i** Statistical analysis of the surface potential of ZnO and substrate from (**g**). **j** Net surface potential of ZnO and the corresponding change of barrier height as a function of time, indicating an increasing oxygen vacancy concentration and a decreasing barrier height occur with the electric pulse. The data in (**j**) is calculated from the data in (**h**) and (**i**).

(Supplementary Note 13 and Supplementary Fig. 29). The net surface potential of ZnO and the corresponding change of barrier height as a function of time are plotted in Fig. 2j, indicating an increasing concentration of oxygen vacancies and a decreasing barrier height occur with the electric pulse. The previous electrical transport characteristics together with the in situ finding here further confirm that the interface barrier can be effectively tuned by the electric pulse-induced rearrangement of oxygen vacancies.

## Electric pulse-tuned piezotronic effect

As illustrated in Fig. 3a-i, the Ag/HfO₂/*n*-ZnO PTJ without electric pulse can respond to an applied strain. By bending PET substrate, an axial strain is induced along the length of ZnO nanowire, leading to positive and negative piezoelectric charges at left end and right end, respectively. Considering the extremely small diameter of nanowire compared with the thickness of PET substrate, the strain can be accurately evaluated from the bending degree of the substrate[25]

(Supplementary Note 14 and Supplementary Fig. 30). Based on piezotronic effect, the strain-induced piezoelectric charges and the corresponding potential at the interface can modulate the interface barrier and thus control the electrical transport. The strain-dependent $I$–$V$ characteristics of PTJ is measured in Fig. 3b. Under a forward bias, the $I$–$V$ curve is dominated by the reversely biased interface barrier in the left end of ZnO nanowire, where strain-induced positive piezoelectric charges are produced. As can be seen, the forward current dramatically increases as the compressive strain increases from 0.00‰ to 1.34‰. It should be noted that the control range of piezotronic effect on the working current is about 0.068–18.3 nA.

As a comparison, when an electric pulse is applied to cause oxygen vacancy migration inside the ZnO nanowire, there exists accumulation of oxygen vacancies near the left interface barrier (Fig. 3a-ii), which can effectively tune the interface barrier. The strain-dependent $I$–$V$ characteristics of PTJ after an electric pulse of 80 V for 30 s is also measured in Fig. 3c. The forward current also increases with increased compressive strain, which is consistent with the case without electric pulse. The difference is that the piezotronic control range of working current has changed to 43.7–71.9 nA (Supplementary Fig. 31). That's to say, the working current range of PTJ is effectively tuned by the applied electric pulse. It is worth noting that the electric pulse-tuned strain-dependent $I$–$V$ characteristics will gradually recover to its original state in Fig. 3d and Supplementary Fig. 32, which is the same as that in Fig. 3b. This recovery indicates the reversibility of the electric pulse-tuned piezotronic effect. It is noteworthy that the effective interface barrier height can be reversibly tuned by electric pulse up to 168.11 meV, corresponding to a change of about 643 times in the working current and internal resistance.

As can be seen, there are two processes in the above experiment. The first process involves the stimulation of electric pulse on the tunneling junction, which drives the vacancy migration inside the $n$-ZnO and thus achieves tunable barrier of the tunneling junction. Because the thickness of HfO$_2$ is about 1.8 nm, there must be some electrons tunneling from the Ag into the $n$-ZnO to neutralize the migration of vacancies. The second process involves the piezotronic effect on the post-stimulated Ag/HfO$_2$/$n$-ZnO. During this process, the electron tunneling is the dominant transport, which is modulated by the strain-induced piezoelectric polarization and the post-stimulation-induced vacancy migration. The electron tunneling will run through the two processes. It should be noted that the vacancy will not pass through the tunneling junction, because if the vacancy under the stimulation of electric pulse passes through the tunneling junction, the $I$–$V$ characteristics in Fig. 3d will not fully recover to its original state.

In addition, we further analysize the modulation of electric pulse on piezotronic effect. By comparing the electrical transport under the same strain, we plot the variation of current change ratio defined as $(I_{after}/I_{before})_{electric\ pulse}$ caused by electric pulse (Supplementary Note 15) with the applied strain in Fig. 3e, based on the summarized current in Supplementary Fig. 33. As can be found that the $(I_{after}/I_{before})_{electric\ pulse}$ under forward bias and reverse bias keep consistent with each other and both decrease with the increase of strain, indicating that the influence of electric pulse on current becomes weaker as the modulation by piezoelectric polarization becomes stronger. Furthermore, we can also calculate the relationship between the piezotronic modification on the effective barrier height and the strain before/after the electric pulse under forward/reverse bias (Fig. 3f). By subtracting the change of effective barrier height modulated by piezotronic effect after and before electric pulse, we can find that the influence of electric pulse on piezotronic effect (piezotronic modification on interface barrier) is more and more obvious with the increase of strain, as shown in Fig. 3g. This means that the piezotronic effect can be effectively tuned by the electric pulse. It's worth to note that the calculated change of barrier height is smaller than the actual value due to the side contact

between the nanowire and the electrode, the real electric pulse regulation capability on piezotronic effect should be stronger (Supplementary Notes 16, 17 and Supplementary Fig. 34). Our results show that for Ag/HfO$_2$/$n$-ZnO PTJ, the piezotronic modification on the interface barrier height can be tuned by electric pulse up to 132.24 meV for forward bias (+80 V) and 148.81 meV for reverse bias (−80 V) under a compressive strain of 1.34‰, providing an opportunity to achieve tunable piezotronics.

By applying different electric pulse voltages to the same device (Fig. 4a), we studied the influence of pulse voltage on the electric pulse-tuned piezotronic effect. The electrical transport characteristics of PTJ under various pulse voltages are measured under strain free (Fig. 4b) and strain (−1.00‰) (Fig. 4c). As can be seen, the current is enhanced by increasing pulse voltage both under strain and without strain (Supplementary Fig. 35). The current change ratio $(I_{after}/I_{before})_{electric\ pulse}$ as a function of pulse voltage is also evaluated here in Fig. 4d, indicating an increased $(I_{after}/I_{before})_{electric\ pulse}$ with the increase of pulse voltage. It is worth noting that the $(I_{after}/I_{before})_{electric\ pulse}$ with strain is less than that without strain, which is consistent with the previously obtained conclusion in Fig. 3e. Figure 4e summaries the piezotronic-modulated working current range of PTJ as a function of the electric pulse voltage. The lower and upper limits of the working current (corresponding to strain ranged from 0.00‰ to −1.00‰) increase with the increase of pulse voltage, and particularly, the upper limit increases exponentially with pulse voltage. This indicates the effective tunability of electric pulse on the working current range of the piezotronic tunneling junction. Additionally, based on semiconductor theory, the change of effective barrier height can be roughly calculated in Fig. 4f. It can be easily found that as the pulse voltage increases, the change of barrier height tuned by electric pulse increases. The gray shadow area in Fig. 4f represents the tunable regions of electric pulse (0 to 80 V) on the piezotronic modification of interface barrier height.

We also checked the influence of post-annealing of as-prepared ZnO nanowires on the device performances (Supplementary Note 18). The post-annealing with temperature ranged from 0 to 400 °C tends to improve the conductivity of devices, indicating an increased carrier concentration and vacancy density inside the ZnO. As can be seen from Supplementary Figs. 36–37, modulations on electrical transport by piezoelectric polarization and electric pulse stimulation are both weakened by increasing post-annealing temperature. These weakened modulations result from the stronger electrostatic shielding effect of carriers. Due to the increased vacancy density, the vacancy gradient caused by electric pulse is smaller, so the recovery time is slightly increased with the increased annealing temperature. The post-annealing maybe a method to achieve long retention of the modulation by electric pulse.

## Regulation mechanism of electric pulse on the back-to-back PTJs

Combined with the above investigations of surface potential (Fig. 2), the electric pulse-tuned piezotronic effect (Figs. 3 and 4), and the theory of piezotronic effect on the tunneling junction (Supplementary Note 19 and Supplementary Fig. 38), the regulation mechanism of electric pulse on the piezotronic effect can be revealed. Detailed piezotronic modification on the energy band for back-to-back PTJs that tuned by electric pulse is illustrated in Fig. 5. Without electric pulse, PTJ (Ag/HfO$_2$/$n$-ZnO in this work) will be formed at both terminals of the piezoelectric semiconductor (ZnO nanowire). Due to the different contact conditions in experiments, the interface barriers at two terminals may be slightly different from each other. As a case schematically diagramed in Fig. 5a, the barrier height and the depleted width of tunneling junction at the left terminal is larger than these at the right terminal. As shown in Fig. 5b, upon compressive strain, the piezoelectric semiconductor with $c$-axis pointing right will produce

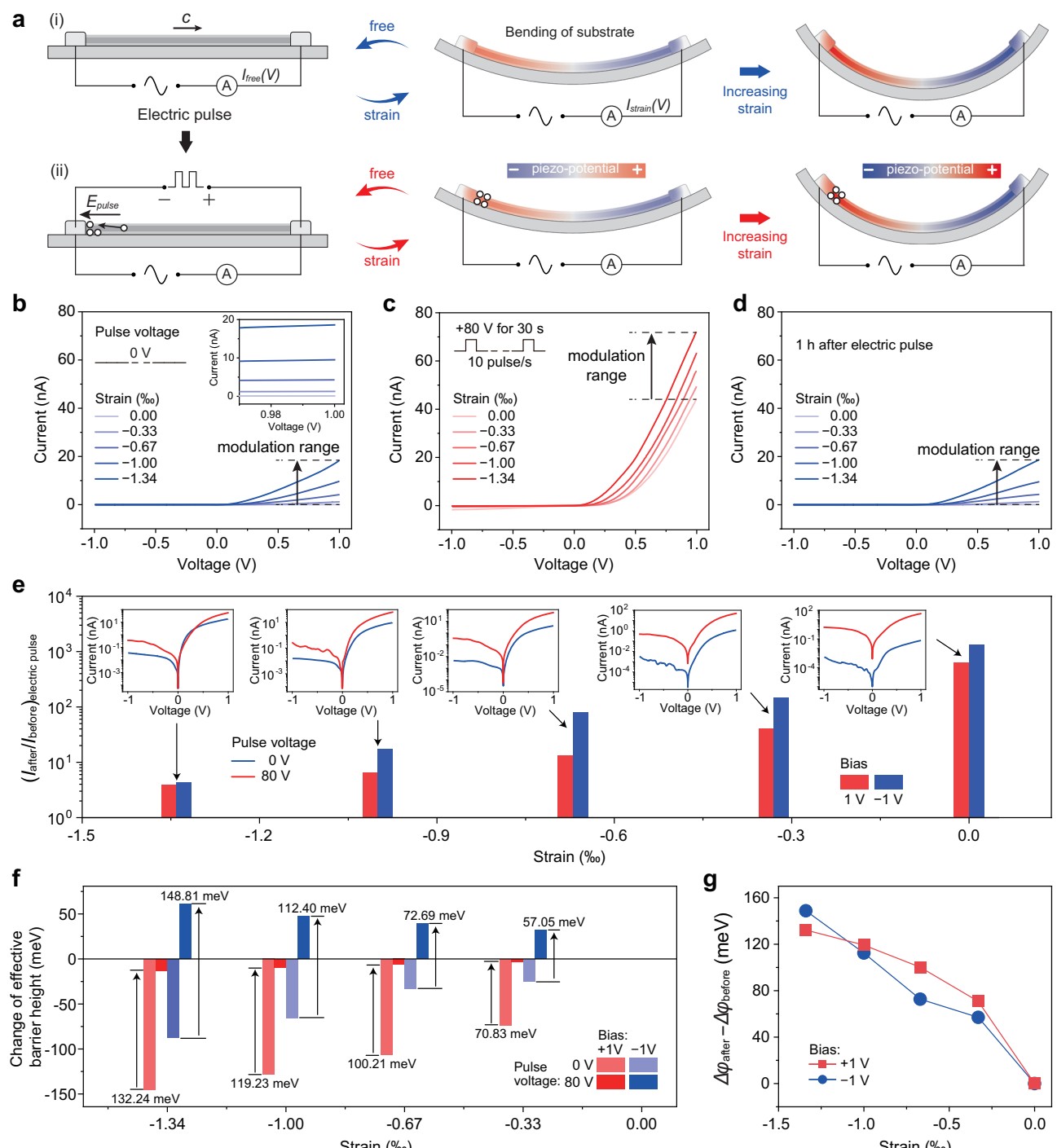

**Fig. 3 | Electric pulse-tuned piezotronic effect. a** Schematic diagrams of the influences of strain and electric pulse on Ag/HfO$_2$/$n$-ZnO PTJ. The red color and blue color in (**a**) (right) represent positive potential and negative potential, respectively. (i) PTJ without stimulation of electric pulse responds to an applied strain. (ii) When an electric pulse is applied to cause migration of oxygen vacancies inside the ZnO nanowire, there exists an accumulation of oxygen vacancies near the left interface barrier, which can effectively tune the interface barrier. **b**–**d** Strain-dependent $I$–$V$ curves of PTJ before (**b**), immediately after (**c**), and 1 h after (**d**)

an electric pulse of 80 V for 30 s. **e** Current change ratio ($I_{after}/I_{before}$)$_{electric pulse}$ of PTJ caused by electric pulses under different applied strain. **f** Relationship between the piezotronic modification on effective barrier height and the strain, before and after electric pulse under forward bias and reverse bias. **g** Effect of electric pulse on piezotronic effect by subtracting the change of effective barrier height modulated by piezotronic effect after and before electric pulse, which shows that the piezotronic effect can be tuned by the electric pulse and becomes more and more obvious with the increase of strain.

positive and negative piezoelectric charges at the left terminal and right terminal, respectively. Based on piezotronic effect, the barrier height and depleted width at the left terminal will both decrease with the increase of strain, whereas those at the right terminal increase. Different from the piezotronic effect utilizing strain-induced

piezoelectric polarization to modulate the interface barrier, the electric pulse tunes the interface barrier by driving the redistribution of vacancies. As subjected to a leftward electric pulse, the interface barrier and depleted width at the left terminal will decrease due to the electric pulse-induced oxygen vacancy migration (Fig. 5c). It is worth

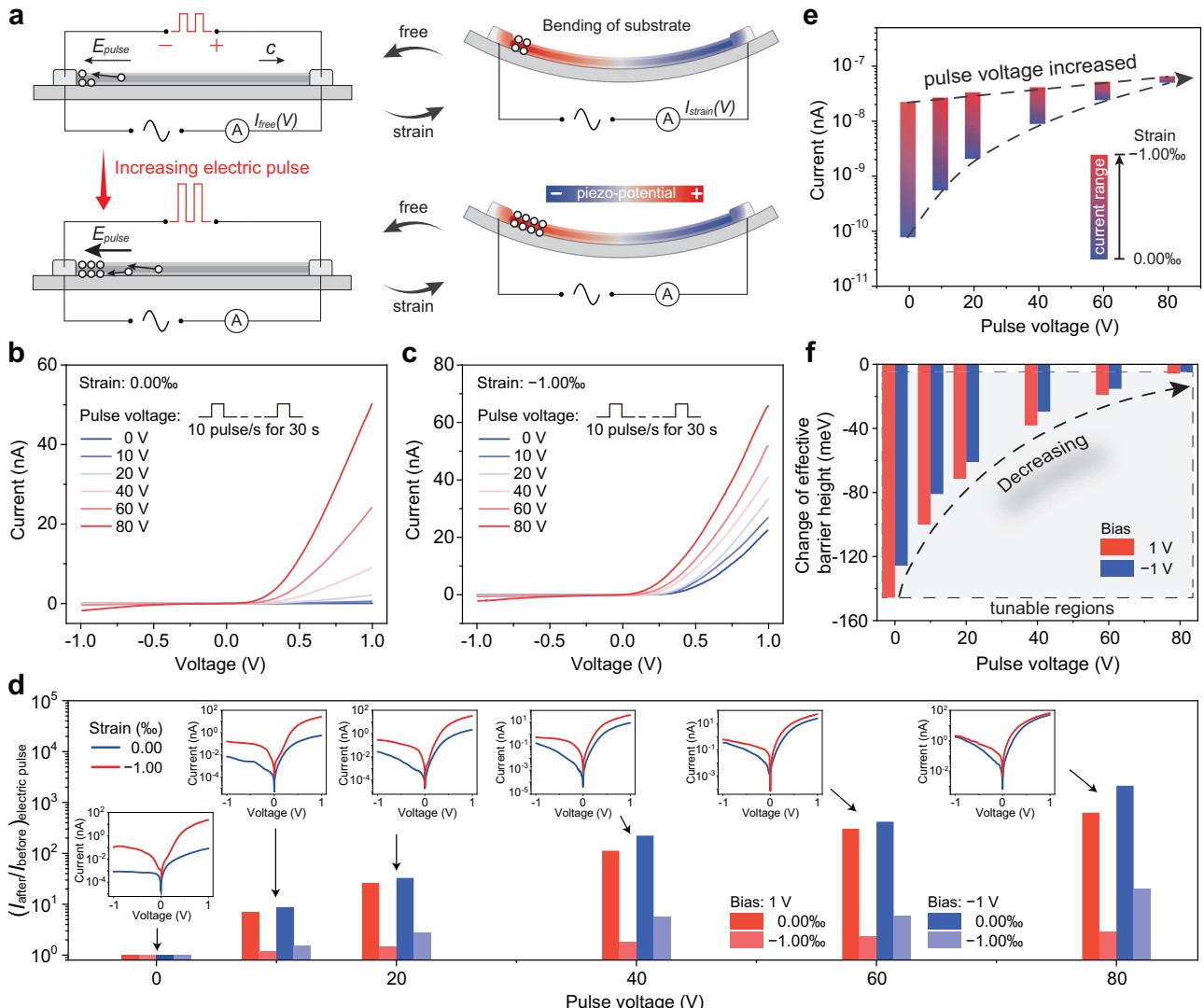

**Fig. 4 | Influence of electric pulse voltage on piezotronic effect. a** Schematic diagrams of applying electric pulse voltages to the same device. The red color and blue color in (**a**) (right) represent positive potential and negative potential, respectively. **b**, **c** Electrical transport of PTJ under various pulse voltages measured under strain free (**b**) and strain of −0.10% (**c**), respectively. **d** Current change ratio $(I_{after}/I_{before})_{electric\ pulse}$ as a function of pulse voltage. **e** Piezotronic-modulated working current range of PTJ as a function of electric pulse voltage. **f** Piezotronic effect induced change of effective barrier height under biases of 1 V and −1 V as a function of the electric pulse voltage. Gray shadow area in (**f**) represents the tunable region of electric pulse on piezotronic modification of interface barrier height.

noting that the interface barrier pointed by the electric pulse can be tuned by $\Delta\varphi_{pulse}/2$, while the other one is almost unchanged (due to the almost unchanged oxygen vacancy concentration at the right terminal). Meanwhile, based on the left interface barrier tuned by electric pulse, the strain-induced piezoelectric polarization can further modulate the interface barriers at two terminals (Fig. 5d). In other words, we can utilize electric pulse to initialize the interface barrier that is further modulated by applying strain based on piezotronic effect, thus to tune the piezotronic modification range of working current of back-to-back PTJs.

To in-depth understand the role of electric pulse on back-to-back PTJs, we further investigated the basic characteristics of electric pulse-tuned piezotronic effect by switching the direction of electric pulse. After applying a positive pulse (+80 V, red line) or a negative pulse (−80 V, blue line) to the device with back-to-back PTJs (Fig. 6a), the $I$–$V$ characteristics of the device are then measured and compared with that before electric pulse stimulation (0 V, black line in Fig. 6a) in Fig. 6b. It can be clearly found that the positive pulse significantly increases the forward current and slightly increases the reverse

current; on the contrary, the negative pulse can significantly increase the reverse current and slightly increase the forward current. This indicates that only the interface barrier that is pointed by the electric pulse can be tuned effectively by the electric pulse, while the other one remains unregulated, which is consistent with the previous discussion in Fig. 5. Figure 6c illustrates the corresponding energy bands of the back-to-back PTJs stimulated by +80 V (left), 0 V (middle), and −80 V (right) electric pulses. Next, by applying compressive strain, we studied the piezotronic effect in these three cases. Figure 6d–f exhibits the corresponding piezotronic modifications of the electrical transport after electric pulses of +80 V, 0 V, and −80 V, respectively. As can be seen, through the stimulation by electric pulse, we can completely change the electrical transport characteristics of the device (Fig. 6b), thereby controlling the regulation range of piezotronic effect (Fig. 6d–f). The asymmetric modulation of $I$–$V$ curves in Fig. 6e also indicates the dominant role of piezotronic effect on the strain-controlled electrical transport, rather than some other factors such as piezoresistive effect or interface traps/defects (Supplementary Notes 20, 21 and Supplementary Figs. 39, 40). Figure 6g, h summarize

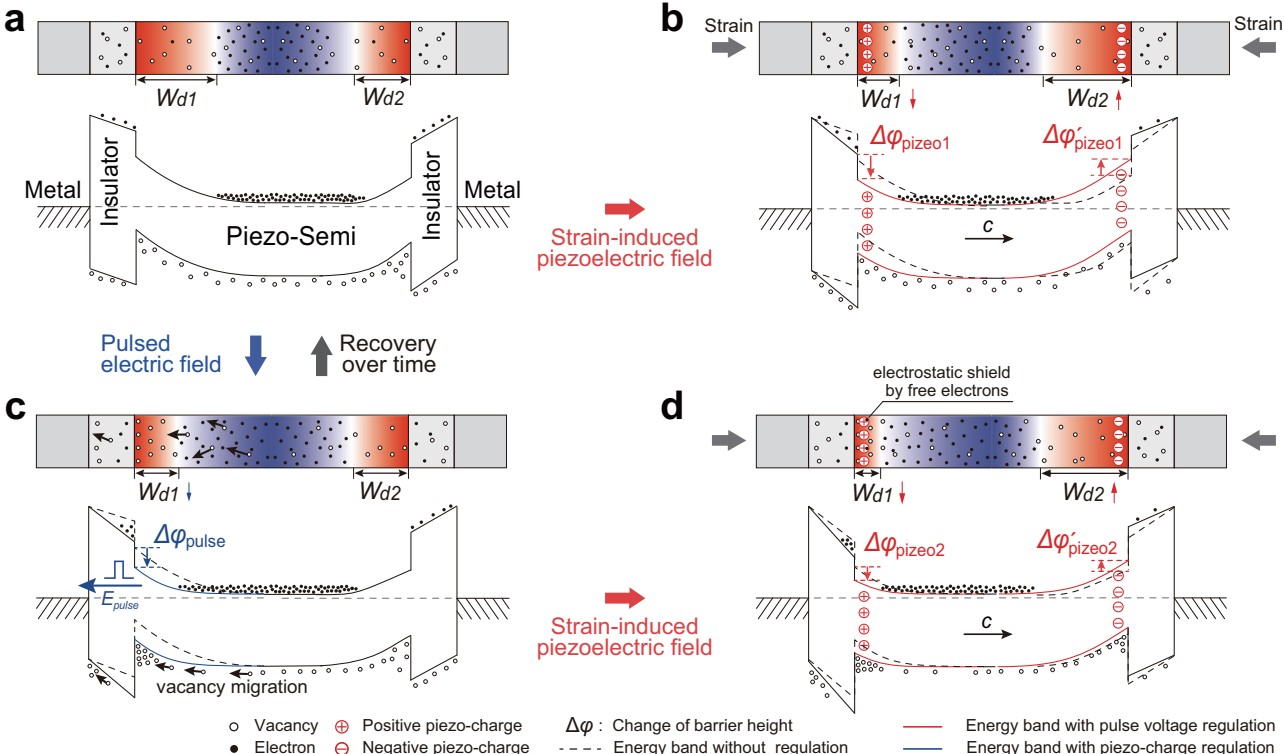

**Fig. 5 | Electric pulse-tuned piezotronic effect on the energy band of back-to-back PTJs. a** Energy band for tunneling junctions formed at both terminals of the piezoelectric semiconductor (ZnO nanowire) without strain and electric pulse. Barrier height and depleted width ($W_d$) of PTJ at the left terminal is larger than these at the right terminal. **b** Energy band tuned by a compressive strain. Based on piezotronic effect, barrier heights at the left and right terminals are, respectively, regulated by $\Delta\varphi_{piezo1}$ and $\Delta\varphi'_{piezo1}$. **c** Energy band tuned by electric pulse. The interface barrier and depleted width at left terminal will decreased due to the electric pulse-induced oxygen vacancy migration. The interface barrier pointed by the electric pulse can be tuned by $\Delta\varphi_{pulse}/2$, while the other one is almost unchanged (due to the almost unchanged concentration of oxygen vacancies at the right terminal). **d** Energy band modulated by the strain-induced piezoelectric polarization based on the left interface barrier tuned by electric pulse. Based on piezotronic effect, the barrier heights at the left and right terminals are regulated by $\Delta\varphi_{piezo2}$ and $\Delta\varphi'_{piezo2}$, respectively. In the schematic diagram of the ZnO nanowire, the red color and blue color represent positive potential and negative potential, respectively.

the currents and the changes of effective barrier height as a function of the applied strain, indicating that the electric pulse plays a significant role in regulating the electrical transport and the piezotronic effect (strain-controlled electrical transport by piezoelectric polarization here).

## Discussion

In summary, we have carefully demonstrated an electric pulse-tuned piezotronic effect, in which the interface engineering by local piezoelectric polarization can be reversibly and accurately regulated by electric pulse. For Ag/HfO$_2$/$n$-ZnO piezotronic tunneling junction, the interface barrier height can be reversibly tuned up to 168.11 meV by +80 V electric pulses, corresponding to a change of 643 times in the working current and internal resistance, and the strain (0.00–0.134%) modulated working current range by piezotronic effect can be shifted from 0.068–18.3 nA to 43.7–71.9 nA, which can be further improved by optimizing the material options and device structure. Additionally, piezotronic modification on interface barrier can also be tuned by electric pulse up to 148.81 meV, which can reversibly and accurately switch the piezotronic performance of an electronics and achieve electric pulse-tuned piezotronic electronics. This work not only provides in-depth understanding for the coupling between electric field-induced redistribution of vacancies and local polarization-controlled interface engineering, but also exhibits the possibility to realize tunable piezotronics with a non-destructive technique, which makes it has great application potential for micro/nano-electromechanical systems and many more.

## Methods

### Synthesis of ZnO nano/microwires

ZnO nano/microwires were prepared by chemical vapor deposition (CVD) through a vapor-solid process. The specific experimental process is as follows: First, grind 2.0 g ZnO powder and 0.4 g graphite powder evenly in an agate mortar. Then, put the obtained uniform mixture into an alumina boat and place it together at the center of the furnace tube. Then, 240 standard-state cubic centimeters per minute (sccm) inert argon and 12 sccm oxygen were introduced into the furnace tube. After the heating procedure, 1150 °C for 40 min, and then 1250 °C for 40 min, ZnO nano/microwires used in the experiment were grown successfully near the outlet of the furnace tube.

### Fabrication of nanometer-thick HfO$_2$ insulation layer

The grown process of HfO$_2$ layer on the surface of ZnO nano/microwires was in Savannah G2 S200 PEALD system with tetrakis(ethylmethylamino)hafnium (TDMAHf, Hf[N(CH$_3$)$_2$]$_4$) and H$_2$O as precursors. The specific operation method is as follows: first of all, we selected a certain number of ZnO nano/microwires to place on the glass sheet after repeated ultrasonic treatment with acetone and ethanol, and then put them together in the ALD reaction chamber. Next, we need to run the program set in the system. The key process parameters were: pressure -0.34 Torr, temperature -150 °C, 20 sccm N$_2$ carrier gas, 0.015 s cycle-1 H$_2$O, 0.1 s cycle-1 TDMAHf, 10 s purge time. Finally, controllable preparation of HfO$_2$ insulation layer of about 1.8 nm can be realized. Since the PTJ with a 1.8 nm HfO$_2$ possesses the best piezotronic modification of electrical transport, the thickness of

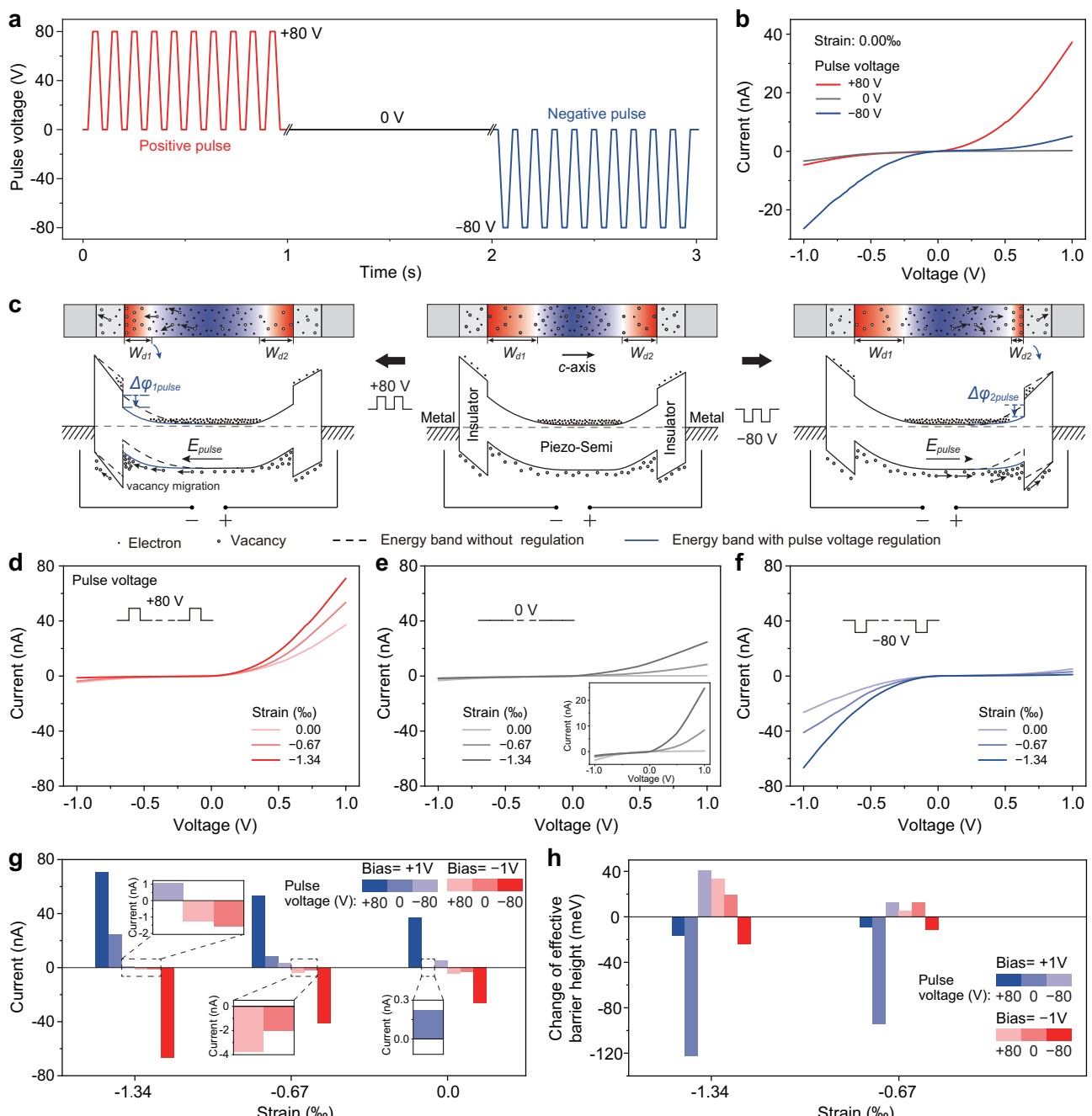

**Fig. 6 | Effect of electric pulse polarity on piezotronic effect. a** Positive pulse (+80 V, red line), 0 V pulse (black line), and negative pulse (−80 V, blue line) applied to the back-to-back PTJs. The pulse frequency is 10 Hz. **b** *I–V* characteristics of back-to-back PTJs after electric pulse stimulations (+80 V, 0 V, and −80 V for 30 s, respectively) under strain free. The positive pulse significantly increases the forward current and slightly increases the reverse current; the negative pulse can significantly increase the reverse current and slightly increase the forward current. **c** Corresponding energy bands of the back-to-back PTJs stimulated by +80 V (left),

0 V (middle), and −80 V (right) electric pulses. In the schematic diagram of PTJs, the red color and blue color represent positive potential and negative potential, respectively. **d–f** Piezotronic modifications of the electrical transport after electric pulses of +80 V, 0 V, and −80 V, respectively. **g, h** Currents and changes of effective barrier height as a function of the applied strain, indicating that the electric pulse plays a significant role in regulating the electrical transport and the strain-controlled electrical transport by piezotronic effect.

about 1.8 nm is chosen. In addition, our further experiments indicate that the HfO$_2$ can also be replaced with A$_2$lO$_3$, SiO$_2$, and ZrO$_2$. More detailed discussions can be found in the Supplementary Note 2, Supplementary Figs. 1–4 and Supplementary Table 1.

**Fabrication of Ag/HfO$_2$/n-ZnO piezotronic tunneling junction**[6]
In the process of device preparation, we first cut the polyethylene terephthalate (PET) films to a size of 3.00 cm × 0.50 cm × 0.15 mm, and

used them as the support substrate of the piezotronic tunneling junction after cleaning with acetone, alcohol, and deionized water. ZnO nano/microwires which have been successfully deposited with insulation layer by ALD are laid in the middle of PET substrate. Then, using the RZF-300 thermal evaporation coating system, the Ag electrode as one of the most widely used electrodes for constructing interface barrier in piezotronics is prepared at both ends of the nanowire/microwire by the thermal evaporation method. After being fixed by

silver paste, the source and drain electrodes are led out by copper wire. Finally, a thin layer of polydimethylsiloxane (PDMS) is used to package the whole device to prevent ZnO oxidation and device damage under repeated mechanical strain. Our simulations show that the PDMS has little impact on the performance of the piezotronic tunneling junction, which is the reason we chose PDMS as the encapsulation layer. (Supplementary Note 2, Supplementary Fig. 5 and Supplementary Table 2).

## Material characterizations

The morphology of nano/microwires and the thickness of $HfO_2$ layer deposited on ZnO were characterized by transmission electron microscopy (TEM) (JEM-2100F) and scanning electron microscopy (SEM) (SU8020, Hitachi). The X-ray diffraction spectrum of materials were characterized by Bruker D8 ADVANCE. The piezoelectricity of ZnO nano/microwires was investigated using AFM (Cypher ES, Asylum Research) with PFM (piezoresponse force microscopy) modes.

## Electrical measurements

The equipments used in the electrical measurements mainly include low noise preamplifiers (SR570, SR560), Keithley 2636b digital source meter and DC linear motor. We used low noise preamplifiers (SR570, SR560) to measure the output voltage and current, and use PCI-6259 (National Instruments Corporation of America) to collect data. Keithley 2636b digital source meter can be used to input pulse voltage stimulation signal and output pulse current response signal. As an alternative, triboelectric nanogenerator can also be used to input pulse voltage stimulation signal (Supplementary Note 10). The strain of the Ag/$HfO_2$/$n$-ZnO PTJs is driven by a DC linear motor. When measuring $I–V$ characteristics, the sweeping voltage applied on devices is between −1 and +1 V.

## Reporting summary

Further information on research design is available in the Nature Portfolio Reporting Summary linked to this article.

## Data availability

The data that support the findings of this study are available from the corresponding author upon request.

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

## Acknowledgements

This research was supported by the National Natural Science Foundation of China (Nos. 52102173, 52302173, and U21A20175), the Natural Science Foundation of Gansu Province of China (No. 23JRRA1101), the National Key R & D Project from Minister of Science and Technology (2021YFA1201602), and the China Postdoctoral Science Foundation (2023M730978).

## Author contributions

S.H.L. and Y.Q. conceived the project. S.H.L. designed the experiments. Q.H.Y. and R.G. performed the experiments. S.H.L., Q.H.Y., Y.Q., J.W. and Q.X. analyzed the results, wrote the manuscript. S.H.L., Q.H.Y., Y.Q., J.W., R.G., and Z.G.L. revised the manuscript. All authors have discussed the results, commented on the manuscript, and approved the final version of the manuscript.

## Competing interests

The authors declare no competing interests.

## Additional information

Shuhai Liu or Yong Qin.

