## [Peer Review File · Nature Communications]

REVIEWER COMMENTS

Reviewer #1 (Remarks to the Author):

This research demonstrated an electric pulse-tuned piezotronic effect by fabricating an Ag/HfO₂/n-ZnO piezotronic tunnelling junction, in which the interface engineering by local piezoelectric polarization can be reversibly and accurately modulated by electric pulse.

I would like to go with a minor revision before it is accepted.

These are my comments/suggestions:

1. There are some earlier reports on HfO₂/n-ZnO-based piezotronic tunnelling junction. The authors should point out the main findings by comparing those previous studies.
2. Why HfO₂ is chosen as an insulating layer in this study?
3. Did the author check the effect of post-annealing of as-prepared ZnO nano/microstructures on the piezotronic performances? The formation of Schottky junction is affected by the density of oxygen vacancy near the interface and hence the barrier height. The author should show the experimental evidence and comment on the presence and nature of the oxygen vacancies of as prepared ZnO and their influence on the electric pulse-induced oxygen vacancy migration.

Reviewer #2 (Remarks to the Author):

The authors reported a novel approach of interface engineering by applying electric pulses on Ag/HfO₂/n-ZnO piezotronic tunneling junction device. The interface barrier height can reversibly be tuned up to 168.11 meV by 80 V electric pulses. And the piezotronic modification on interface barrier tuned by electric pulse can be up to 148.81 meV under a strain of 1.34%. The article demonstrated sufficient experimental results on the phenomenon of tunable interface barrier height by the stimulation of electric pulse and bending-induced strain, and provided a reasonable theoretical analysis of interfacial engineering. It shows a novelty in the further investigation of piezotronics, and gives a guide to design and optimize novel piezotronic devices based on low-dimensional materials (e.g., ZnO, GaN, MoS₂). The reviewer would like to recommend it be published in Nature Communications, however, some critical issues remain to be addressed.

Comments

- (1) In the Introduction, the authors introduced “flexoelectronics and ferroelectric electronics” and discussed their working principles in Fig. 1. However, only piezotronics was investigated in the manuscript. It seems to be redundant to talk about flexoelectronics and ferroelectric electronics.
- (2) Why did use an electric pulse in the investigation on piezotronics? The authors need to give more discussion on the advantages or disadvantages of piezotronic devices.
- (3) The electric pulse has a large amplitude of 80 V and a long period. Actually, the electric pulse with a range of 1 ns to 100 ms would be widely applied on various electronic devices. Is it possible to reduce the period time of the pulse, making it equivalent to that of a long pulse condition? And the different frequency of pulse needs to be conducted in the experiments. In addition, the voltage amplitude of 80 V is high enough and would easily cause device breakdown.
- (4) AFM has the ability to implement the electric pulse mode. Could it conduct in-situ measurement of tunable interfacial barrier height by AFM?
- (5) What is the thickness of ALD HfO₂? The authors claimed “when an electric pulse is applied to cause oxygen vacancy migration inside the ZnO nanowire, there exists accumulation of oxygen vacancies near the left interface barrier”. If the HfO₂ thickness is less than 3-5 nm, tunneling is possible as the dominant mechanism for interfacial engineering. Or the HfO₂ layer would have some defects that could play an important role in trapping electrons from the ZnO nanowire. Please give more discussion about that.
- (6) Will the recovery time of the interface barrier change after heating? The authors need to give more explanation about the working principle. Additionally, please comment on the difference of interfacial barrier height tuned by the electric-pulse or TENG, which were previously reported.
- (7) How to achieve a long retention of the tunable barrier height? Could you comment on that? Because the well-controlled interfacial barrier height would be beneficial for the optimization of electronic devices for practical applications.

Reviewer #3 (Remarks to the Author):

In this work, Qin et al. reported in depth investigation of modulation of interface engineering using electrical pulse by the piezotronic effect. They prove that the proposed model is valid for the piezotronic device using literature data. Therefore, I recommend its publication, with suggested major modifications, as below:

- 1) The author should show the high-resolution cross-section image of the Ag/HfO₂/ZnO junction.
- 2) There are many insulator layers, why the author used HfO₂ insulator layer for device fabrication. What is the thickness of the HfO₂ layer.
- 3) Why author measured the I-V characteristics (Fig. 2e) at low voltages. Did author measure higher (> 1 V) voltages or not. If yes, please include it in the manuscript. If not, please explain the reason.

4) Discussion needs about the fundamental understanding of piezotronics and why the contact the author used is Ag.

5) In this paper, the PDMS is piezoelectric ceramics materials, the authors should discuss whether the PDMS coating has influence on the performance of the Ag/HfO₂/ZnO junction.

Point-to-Point Response to Reviewers' comments:

We are grateful to thank the reviewers for their in-depth reviews and important/constructive questions/suggestions regarding our manuscript. In the pages that follow, we provide our responses to each of the reviewer's questions.

Reviewer #1 (Remarks to the Author):

This research demonstrated an electric pulse-tuned piezotronic effect by fabricating an Ag/HfO₂/n-ZnO piezotronic tunnelling junction, in which the interface engineering by local piezoelectric polarization can be reversibly and accurately modulated by electric pulse. I would like to go with a minor revision before it is accepted. These are my comments/suggestions:

Response:

We are particularly grateful to you for the detailed, in-depth, and constructive comments/suggestions on our work. These comments are highly valuable to our research, and we have largely improved the manuscript according to your suggestions.

Question #1

There are some earlier reports on HfO₂/n-ZnO-based piezotronic tunnelling junction. The authors should point out the main findings by comparing those previous studies.

Response:

We highly appreciate you for this suggestion. To reveal the novelty of this work, it is really important to clarify the differences between the study of tunneling junction in this work and the study of tunneling junction in previous works. Previous studies on piezotronic tunneling junctions have proposed that quantum tunneling can be tuned by externally applied mechanical stimuli^{1, 2}. These works utilized the strain-induced piezoelectric potential to regulate the tunneling barrier's height and width in parallel, thereby synergistically modulating the electrical transport process, and achieving a highly sensitive strain sensor based on piezotronic tunneling junctions. In this work, we report a strategy of tuning piezotronic effect reversibly and accurately by electric pulse, using Ag/HfO₂/n-ZnO piezotronic tunneling junction. In the main text, our results show

that for Ag/HfO₂/n-ZnO piezotronic tunneling junction, the interface barrier height can be reversibly tuned as high as 168.11 meV by electric pulse, and the strain (0~1.34‰) modulated current range by piezotronic effect can be switched from 0~18 nA to 44~72 nA. This study provides opportunities to achieve reversible control of piezotronics and even other interface engineering by polarizations.

In order to explain the novelty of using HfO₂/n-ZnO-based piezotronic tunneling junction to achieve the tunable interface barrier in this work more clearly, here we will further describe (1) the piezotronic tunneling junction with high strain sensitivity, and (2) the novelty of piezotronic tunneling junction with *tunable* interface barrier. It is expected that the novelty of this work can be better revealed through this discussion.

(1) Piezotronic tunneling junctions with high strain sensitivity in previous works

In 2019, the research team (including our group) led by Zhong Lin Wang coupled the piezoelectric effect with the quantum tunneling effect and used atomic force microscopy (AFM) to verify a giant switching ($>10^5$), and fast response (≈ 4.38 ms) in a novel piezotronic tunneling junction¹. This investigation offers initial evidence that the high performance of this tunneling junction is achieved by simultaneously adjusting the height and width of the tunneling junction barrier through the piezotronic effect in principle.

Building upon the aforementioned groundwork, in 2022, our group developed Ag/HfO₂/n-ZnO piezotronic tunneling strain sensor and Ag/n-ZnO piezotronic Schottky strain sensor using ZnO nanowires². The experimental results indicate that the piezotronic tunneling strain sensor outperforms the Schottky strain sensor. Specifically, as the tensile strain on the device increases from 0.00% to 0.10%, the forward current of the piezotronic tunneling strain sensor consistently increases by two orders of magnitude, which is 300.5 times greater than that of the Schottky strain sensor. Additionally, the strain sensitivity factor (Gauge factor) of Ag/HfO₂/n-ZnO reaches 3.8×10^5 , significantly higher than that of Ag/n-ZnO, as depicted in **Figure R1**². Moreover, this study initiates the control of the tunneling barrier height and width through the piezotronic effect on a macroscopic device. The dynamic current response

of Ag/HfO₂/n-ZnO piezotronic tunneling strain sensor shows that the piezotronic effect affects the interface carrier tunneling transport at both ultra-low strain (<0.01%) and moderate strain (0.01~0.10%) stages. This indicates that these two stages correspond to the polarization-dominated regulation of barrier width and barrier height, respectively (**Figure R2**²).

The above is the main research content of tunneling junction in previous works. Next, we will introduce the main finding in this work based on HfO₂/n-ZnO-based piezotronic tunneling junction.

(2) Novelty of piezotronic tunneling junction with tunable interface barrier in this work

Different from previous works focusing on piezotronic modification of tunneling junctions, the innovation of this work lies in utilizing electric pulse stimulation to achieve *tunable* barrier of tunneling junction and thus regulate the piezotronic effect in the tunneling junction. Under the driving force of the electric field produced by electric pulse stimulation, positively charged oxygen vacancies are gathered towards the interface, thereby enabling the modulation of the barrier, and altering the range over which piezoelectric polarization can be controlled. To provide a clearer explanation of the novelty of this work, we will further elaborate on ① working principle of piezotronic effect, demonstrate ② dependence of piezotronic devices on interface barriers, and then illustrate ③ mechanism of modulating interface barriers by electric pulse, and ④ modulation of piezotronic effect by electric pulse. Through this discussion, we aim to better reveal the novelty of tunable piezotronic interfaces by electric pulses in this work.

① *Working principle of piezotronic effect*

Currently, the piezotronic effect is mainly applied to the two most fundamental structures in semiconductor devices: the Schottky interface and the *p-n* interface^{3,4}. In order to provide a detailed explanation of the piezotronic effect and its working principle, we will focus on discussing the regulation principle of the Schottky interface by piezotronic effect.

For n -type semiconductors, when they come into contact with a metal with high work function, the resulting interface typically forms a Schottky barrier, causing the Fermi surfaces at both ends of the interface to become flat, thus achieving dynamic thermal equilibrium. If the semiconductor exhibits piezoelectric characteristics and is subjected to the appropriate strain, a corresponding piezoelectric polarization charge will be generated at the interface. This polarization charge is the result of the superposition of dipole moments generated by materials with non-centrosymmetric crystal structures when subjected to strain, commonly referred to as piezoelectric polarization bound charge or simply piezoelectric polarization charge, which also lacks migration characteristics. As depicted in **Figure R3**, because the piezoelectric polarization charge is not entirely shielded by the free carriers in the semiconductor, these unshielded polarization charges will create a Coulomb electrostatic field at the interface^{3,4}. This, in turn, leads to a redistribution of the free carriers and a modification of the interface band^{4,5}. If the positive piezoelectric polarization bound charge is generated at the interface, the Schottky barrier will decrease, as the positive potential lowers the electron band. Conversely, if the negative piezoelectric polarization bound charge is generated at the interface, the Schottky barrier will increase accordingly. At this time, as shown in **Figure R3b** and **R3c**, if a bias voltage is applied to the Schottky barrier, then we will find that the carrier transport at the interface will be completely different from those without bias. When the height of the Schottky barrier is reduced due to piezotronic effect (**Figure R3**, left), both forward and reverse bias will make it easier for electrons to cross the barrier and interface, resulting in higher current; Conversely, when the height of the Schottky barrier is increased due to piezotronic effect (**Figure R3**, right), the electrons will need more energy to cross the barrier and interface, resulting in a decrease of current. For the Schottky interface formed by p -type piezoelectric semiconductor and metal with small work function, the regulation mechanism of piezotronic effect is similar.

© *Dependence of piezotronic devices on interface barriers*

Piezotronic devices based on piezotronic effect offers significant advantages.

Primarily, the piezotronic effect mainly regulates the electrical transport of the interface, which typically exhibit high sensitivity³⁻⁵. Piezotronic devices are often utilized for actively sensing mechanical stimuli. This allows for the regulation and optimization of detector performance through the application of driving forces, as observed in applications such as light-emitting diodes⁶, biomolecular detection^{7, 8}, ultraviolet detection⁹, and lasers¹⁰, *etc.* Additionally, the measurement of electrical properties of devices can be utilized to detect mechanical stimuli, as demonstrated in logic units¹¹⁻¹³, vibration sensing¹⁴, and tactile sensing¹⁵. In summary, the advantage of piezoelectric polarization interface modulation lies in its capability to achieve fine regulation and optimization of the performance of piezotronic devices.

The basic structure of piezotronic devices, as shown in **Figure R4**, is mainly composed of piezoelectric semiconductor and electrodes at both ends. In this structure, the interface barrier forms between the piezoelectric semiconductor and the electrode, and it is regulated by the piezoelectric polarization charges induced by strain and the corresponding piezoelectric potential, thereby controlling the device. The performance of piezotronic devices is dependent on the engineered interface barrier and is highly sensitive to its characteristics. This sensitivity arises from the fact that the piezotronic effect is produced by strain-induced piezoelectric charges that modulate the energy band at the interface, resulting in charge rearrangement at the interface and influencing the transport of carriers³⁻⁵. Consequently, the construction and precise control of the interface barrier play a crucial role in the operation of piezotronic devices, as even minor adjustments to the interface barrier can have a substantial impact on device performance.

Furthermore, the disadvantage of piezotronic devices lies in the fact that once they are fabricated, the internal interface barrier is fixed, leading to the inability to tune the device performance. This means that the range of response current for the same strain will also be fixed, which may render piezotronic devices unsuitable for certain applications. Since polarized interface engineering highly depends on the nature of the interface barrier, once the interface potential barrier is established, its fixed value and uncontrollability will limit the performance and application scenarios of

polarization interface engineering.

If tunable performance of piezotronic devices can be achieved, then the research on interface engineering of piezoelectric, flexoelectric, and ferroelectric polarization in semiconductor devices will have significant importance for their applications in the fields of electronics, optoelectronics, and catalysis⁶⁻¹⁰. In addition, polarization interface engineering is very important for tunable electronic devices (such as piezotronic devices, flexoelectronic devices and ferroelectric electronic devices)¹⁶⁻¹⁸. Therefore, the search for interface barrier modulation strategies suitable for various application scenarios is of great scientific and engineering significance.

③ *Mechanism of modulating interface barriers by electric pulse*

Natural defects of oxygen vacancies are inevitably present during the preparation of ZnO nano/microwires¹⁹. Theoretically, oxygen vacancies can exist in different valence states (Vo^{2+} , Vo^+ and Vo^0), and at the interface between metal and zinc oxide, neutral oxygen vacancies (Vo^0) can easily transform into positively charged ionized oxygen vacancies (Vo^{2+})^{20, 21}. The formation of Schottky barriers is influenced by the density of oxygen vacancies (Vo) near the interface. A high electric field can exert a force on the positively charged ionized oxygen vacancies, leading to ionic migration at the interface²². In this work, continuous electric pulses can drive the migration of positively charged oxygen vacancies and their accumulation at the contact interface of the tunneling junction (**Figure R5a**), forming a narrow region of positive charge, bending the energy bands, and reducing the barrier height²⁰. Consequently, when a piezotronic device is stimulated by electric pulse, the response current will increase.

Next, with the help of the band diagram in **Figure R5b**, we will discuss in detail the physical process of the aggregation and recovery of ionic oxygen vacancies at the tunneling junction interface after electric pulse stimulation. In the initial state, the metal, insulator, and piezoelectric semiconductor contact to form a tunneling junction with a certain barrier height and width, and the concentration of ionic oxygen vacancies near the interface is low (i-1). When the device is stimulated by electric pulses for a certain period of time, the positively charged ionic oxygen vacancies will gather at the interface driven by the electric field, and the concentration of ionic oxygen vacancies at the

interface will increase, leading to the decrease of the height of the tunneling junction barrier ($\Delta\phi_{\text{pulse}}$) (ii-2). When the pulse voltage is removed, due to the imbalance of oxygen distribution caused by the external electric field ²³, the high concentration oxygen vacancy at the interface begins to diffuse to the interior of ZnO, and the barrier will gradually recover (i-1). Because the oxygen vacancy is rapidly gathered to the tunneling junction interface under the drive of electric field, and the recovery process of the barrier height depends on the slow diffusion of oxygen vacancy, so the recovery process of device performance after electric pulse stimulation needs a period of time.

Furthermore, as the electric pulse stimulation time increases, the density of ionic oxygen vacancies near the ZnO contact interface also increases, leading to a gradual decrease in the barrier within the tunneling junction area. Additionally, oxygen vacancies in low electric field areas exhibit slower movement compared to those in high electric field areas ²⁴. Consequently, a sufficiently strong electric field can induce the accumulation of high-density oxygen vacancies at the tunneling junction interface, thereby altering the effective barrier. This phenomenon explains why higher electric pulse voltage or longer stimulation time result in more pronounced increases in the device current. Subsequently, after a certain duration of pulse voltage stimulation, a dynamic equilibrium is established between the migration of ionic oxygen vacancies to the tunneling junction interface driven by the electric field and the diffusion away from the interface due to concentration differences of oxygen vacancies. As stimulation time increases, the barrier undergoes less significant changes, leading to nearly unchanged measured I - V curves and current values.

④ *Modulation of piezotronic effect by electric pulse*

As presented in the manuscript, we fabricated an Ag/HfO₂/n-ZnO piezotronic tunneling junction as a demonstration illustrated in **Figure R6a**. As bending the substrate, a strain will be induced along ZnO nanowire to make it produce opposite piezoelectric charges at both terminals of ZnO (**Figure R6c-i**) that acts as a ‘gate’ voltage to control the charge transport of interface (piezotronic effect). As shown in **Figure R6a-ii**, with electric pulse stimulation, oxygen vacancies migration occurs in ZnO nanowire to tune the barrier height at the tunneling interface, which will then

determine the range of working current modulated by piezotronic effect. By using a +80 V electric pulse, this work shifted the operating current range of Ag/HfO₂/n-ZnO piezotronic tunneling junction strain sensor from 0.068~18.3 nA (**Figure R6b**) to 43.7~71.9 nA (**Figure R6c**). So, electric pulse can effectively tune the piezotronic effect in tunneling junctions.

Furthermore, the polarization interface engineering (including piezotronic effect) is very important for tunable electronic devices (such as piezotronic devices, flexoelectronic devices and ferroelectric electronic devices)²⁵⁻²⁸, and the search for interface barrier modulation strategies suitable for various application scenarios is of great scientific and engineering significance. Considering the obvious regulation of electric pulse on polarization interface engineering (piezotronic effect), the electric pulse-tuned piezotronic effect for interface engineering studied in this work should be important.

Through above discussions, we can find that the main finding of this work is totally different from previous works based on piezotronic tunneling junctions. The novelty of this work lies in **applying electric pulse stimulation to achieve tunable barrier of tunneling junction and thus regulate the piezotronic effect** in the tunneling junction.

We thank you again for this constructive suggestion. In order to improve the quality and understanding of our work, we have included the relevant content in the Supplementary Information and highlighted it in red.

Figure R1. Device structure and interface electrical characteristics of Ag/HfO₂/n-ZnO piezotronic tunneling strain sensor and Ag/n-ZnO Schottky strain sensor². **a**, Schematic diagram of the mechanism for piezoelectric polarization modulation of band in piezotronic tunneling junctions. **b**, Optical image of the piezotronic tunneling strain sensor. **c**, Strain-dependent I - V characteristics of Ag/HfO₂/n-ZnO piezotronic tunneling strain sensor and Ag/n-ZnO Schottky strain sensor. **e**, Gauge factor of piezotronic tunneling strain sensor and Schottky strain sensor.

Figure R2. Preliminary analysis of the regulation of the height and width of the

tunneling junction barrier by piezotronic effects². **a**, Five cycles of current response to continuously changed tensile strain. **b**, Energy band profile of tunneling junction. **c**, $\ln(I_{\text{strain}}/I_{\text{free}})$ as a function of strain. The regions that change drastically (blue region, strain $<0.01\%$) and slowly (red region, strain $>0.02\%$) with strain correspond to different dominant mechanisms: the former is dominated by $(\Delta BW)_{\text{eff}}$, and the latter is dominated by $(\Delta BH)_{\text{eff}}$, both of which caused by the piezotronic effect. **d**, Change of the effective barrier height and width derived from c as a function of applied strain.

Figure R3. Band structure and carrier transport characteristics of metal-semiconductor Schottky interface under the control of piezoelectric polarization bound charge. a, $V_{\text{bias}} = 0$ (thermal equilibrium). **b**, $V_{\text{bias}} < 0$ (reverse bias). **c**, $V_{\text{bias}} > 0$ (forward bias).

Figure R4. The basic typical structure of piezotronic devices.

Figure R5. Mechanism diagram of electric field regulating oxygen vacancies. a, Migration of oxygen vacancies in ZnO promoted by electric field. **b,** Regulation of energy band of tunneling junction by electric pulse voltage.

Figure R6. Regulation of piezotronic effect by electric pulse. **a**, Introducing a strain (i) and an electric pulse (ii) to the Ag/HfO₂/n-ZnO piezotronic tunneling junction to tune the interface barrier. **b**, **c**, Strain-dependent I - V curves of piezotronic tunneling strain sensor before (**b**), immediately after (**c**) electric pulse of 80 V for 30 s.

Question #2

Why HfO₂ is chosen as an insulation layer in this study?

Response:

We thank you very much for this comment. It's indeed essential to give reasons for specifically choosing HfO₂ as an insulation layer. So we have merged our investigation about the influence of different insulation materials on the performance of piezotronic tunneling junctions, together with the relevant literature reports.

Firstly, there are many literature reports that dielectric insulation layer with high k values and wide band gaps can not only reduce the leakage current and the overall power consumption during device operation, but also has good stability to withstand breakdown. As compared with SiO₂, Al₂O₃, MgO and some other insulation materials, HfO₂ has a good advantage in terms of k value and band gap, as shown in **Table R1**²⁹⁻³³. Based on our previous survey, we chose HfO₂ at the beginning as the dielectric

insulation layer of the piezotronic tunneling junction.

Secondly, in order to study the influence of insulation materials on carrier transport in piezotronic tunneling junctions, we characterized the performance of piezotronic tunneling devices using HfO₂, ZrO₂, Al₂O₃ and SiO₂ as insulation layers with a same thickness of ~1.8 nm under the conditions of tensile strain of 0.00%, 0.033%, 0.067% and 0.10%, respectively. As shown in **Figure R7**, the current values measured at +3 V bias for four kinds of piezotronic tunneling devices were selected. The number of devices used for each type is 20. It can be clearly seen from the figure that when the strain is 0.00%, the current values of the four types of piezotronic tunneling devices are randomly distributed between 0.01 and 7 nA, and when the strain is 0.10%, their current values are also irregularly distributed between 30 and 250 nA. According to the current data in **Figure R7**, the current on/off ratio and change of Schottky barrier height of four types of piezotronic tunneling devices under 0.10% strain were calculated, counted and compared in **Figure R8**. The comparison results show that changing the dielectric insulation materials such as HfO₂, ZrO₂, Al₂O₃ and SiO₂ has almost no influence on the sensing performance of piezotronic tunneling devices. The classical carrier transport of metal-insulator-semiconductor (MIS) tunneling junction can be expressed as ¹⁸:

$$I = SA^*T^2 \exp(-\alpha_T d \sqrt{q\varphi_T}) \exp\left(-\frac{q\varphi_B}{kT}\right) \left[\exp\left(\frac{qV}{nkT}\right) - 1 \right]$$

where $\exp(-\alpha_T d \sqrt{q\varphi_T})$ is the tunneling probability term, S is the contact area of the tunneling barrier, A^* is the effective Richardson constant, q represents the charge of an electron, k is the Boltzmann constant, T is the absolute temperature, φ_B represents the height of the Schottky barrier, n is the ideality factor, d is the thickness of the insulation layer in the tunneling junction, $\alpha_T = 2\sqrt{2qm^*}/\hbar$ (m^* represents the effective mass of the charge carriers, \hbar is the reduced Planck constant), φ_T represents the effective height of the tunneling barrier. It can be seen that the tunneling current is mainly dependent on the thickness of insulator layer (d) and the Schottky barrier height (φ_B) (which is determined by the metal's work function and the semiconductor's electron affinity), and slightly dependent on the band gap of insulator layer (which together with its electron affinity will affect the effective height of the tunneling barrier $q\varphi_T$), and

almost not dependent on the k value. Given that the band gaps of HfO₂, ZrO₂, Al₂O₃ and SiO₂ are not very different, the piezotronic modification of tunneling current has little to do with the choice of dielectric insulation layer, which is consistent with our experimental results in **Figures R7** and **R8**. So, it is no problem to choose either HfO₂, ZrO₂, Al₂O₃ or SiO₂ to construct the tunnel junction.

In summary, the rationale for using HfO₂ as an insulation layer for tunneling junctions at the beginning stems from the advantageous properties of HfO₂ reported in the literature; and then, through our comparative experiments, we find that it is possible to choose either HfO₂, ZrO₂, Al₂O₃ or SiO₂, so we select HfO₂ for convenience. It should be noted that HfO₂ exhibits good thermal stability and is compatible with silicon technology²⁹, rendering it a suitable choice for tunneling junctions, while other dielectric materials such as ZrO₂, Al₂O₃, and SiO₂ could also serve as potential candidates for HfO₂. It is helpful for readers to understand our design and improve the quality and readability of the paper by showing our ideas of insulation layer selection. We have incorporated above discussions into the revised Supplementary Information, marked in red. We express our sincere appreciation for your constructive and valuable comment once again.

Table R1. Dielectric constants (k) and band gaps (E_g) of several typical insulation materials²⁹⁻³³.

Insulation materials	Dielectric constants k	Band gaps E_g (eV)
HfO ₂	20~25	5.6~6.0
ZrO ₂	17~25	5.1~7.8
SiO ₂	3.9	9
Si ₃ N ₄	7~7.5	5~5.3
Al ₂ O ₃	9	8.8
MgO	8.4~9.8	8.7~8.8

Figure R7. Influence of insulation materials on piezotronic modification of the carrier transport of piezotronic tunneling devices. a-d, Piezotronic modification of the carrier transports of four types of piezotronic tunneling devices using HfO₂ (a), ZrO₂ (b), Al₂O₃ (c) and SiO₂ (d) as insulation layers under the conditions of tensile strain of 0.00%, 0.033%, 0.067% and 0.10%, respectively. The above currents were measured at bias of +3.0 V.

Figure R8. Sensing performance of piezotronic tunneling devices. a, b, On/off ratio (a) and change of effective Schottky barrier height (b) of the devices based on

piezotronic tunneling junction with different insulation materials of HfO₂, ZrO₂, Al₂O₃ and SiO₂. The error bars denote standard deviations of the mean.

Question #3

Did the author check the effect of post-annealing of as-prepared ZnO nano/microstructures on the piezotronic performances? The formation of Schottky junction is affected by the density of oxygen vacancy near the interface and hence the barrier height. The author should show the experimental evidence and comment on the presence and nature of the oxygen vacancies of as prepared ZnO and their influence on the electric pulse-induced oxygen vacancy migration.

Response:

We appreciate you so much for this valuable and in-depth suggestion. In theory, post-annealing of as-prepared ZnO nano/microstructures will affect the vacancy concentration, which can affect the interface barrier height. Here, we conduct relevant experiments to check the influence of post-annealing of ZnO on the piezotronic performances of devices and their influence on the electric pulse-induced oxygen vacancy migration. In order to answer this question more logically, we discuss it in the following three parts.

(1) Influence of post-annealing on different valence oxygen vacancies in ZnO

We annealed ZnO in air and characterized the influence of annealing on different valence oxygen vacancies in ZnO. In ZnO and oxygen poor condition, the formation energies of oxygen vacancies in different charge states are calculated as a function of the Fermi level, as shown in **Figure R9**. The valence band maximum (E_{VBM}) is set as zero. The formation energy of Vo^0 is 1.0 eV and generally larger than that of Vo^{2+} , until the Fermi level surpasses the thermal transition energy $\varepsilon(2+/0)$ (0.5 eV below the conduction band maximum, E_{CBM}). Vo^{1+} is generally unstable due to its formation energy lies between the neutral and bivalent oxygen vacancy defect formation energy^{34, 35}. Due to the difference in formation energy, the transition of oxygen vacancy between different states, especially that between Vo^0 and Vo^{2+} is possible: Vo^0 can

transform to Vo^{2+} thermally, and the transformation would be reversed when the Fermi level is larger than the thermal transition energy.

The relative concentration of specific oxygen vacancy states cannot be directly obtained experimentally except that of Vo^{1+} , which can be characterized by EPR (**Figure R10**). The peak position in EPR spectra that associated with Vo^{1+} is still controversial, as connections with $g \sim 1.990, 1.960, 1.996$ and 2.003 are reported³⁶⁻⁴⁰, and its relative intensity can be used to indicate the relative concentration of Vo^{1+} ³⁶. This indicates that the VO^{1+} concentration in ZnO annealed in different temperatures is quite low, and the conductivity is correlated with the concentrations of Vo^0 and Vo^{2+} .

As a relative indirect method, PL can be used to characterize V_O in ZnO⁴⁰⁻⁴³. However, due to the complex defect structure in ZnO, various sources are proposed to explain the PL spectra, especially the visible emission of ZnO: shallow donor, oxygen vacancy with different states, zinc vacancy, or oxygen and zinc interstitials⁴⁰⁻⁴⁶. To pinpoint the origin of different defect emissions, PL results must be analyzed in combination with other methods. The PL spectra of ZnO annealed in different temperatures are shown in **Figure R11a**. The intensity is normalized by the intensity of UV emission. The spectra can be divided into three emission regions: UV emission (**Figure R11b**), blue emission (**Figure R11c**) and red emission (**Figure R11d**). Interestingly, a similar pattern is found in these emission regions. The UV emission shows a red shift (380 to 395 nm, 3.26 to 3.13 eV) with the increase of annealing temperature from 200 to 400 °C and the shift is reversed (395 to 385 nm, 3.13 to 3.22 eV) with the further increase of the annealing temperature. For the blue/green emission (430–490 nm), the intensity increases with the increase of annealing temperature from 200 to 400 °C, and the trend is reversed with the further increase of annealing temperature. For the orange/red emission (610 to 720 nm), the intensity decreases with the increase of annealing temperature from 200 to 350 °C, and the correlation is reversed with further increase of the annealing temperature (the peak at 600 nm is caused by the 300 nm excitation source used). The annealing temperature around 400 °C is a turning point for all three emission regions.

The UV emission is related with the exciton emission near the band gap. The red shift of UV emission has been observed previously⁴⁰, and is due to the increase of carrier density introduced by oxygen vacancies. In other words, when the annealing temperature increases from 200 to 400 °C, the concentration of Vo^{2+} increases, and further increase of temperature would result to the decrease of Vo^{2+} concentration. As to the blue/green emission (430–490 nm), it is attributed to the defect emission of oxygen vacancy^{42, 43, 45, 47-49}, transition between Vo^{1+} and photoexcited holes^{40, 44, 50}, transition involves Vo^{2+} ⁵¹, or surface defects⁵². Since the EPR spectra indicates that the concentration of Vo^{1+} is negligible, the transition that involves Vo^{1+} can be excluded. As to the orange/red emission, transitions related with interstitial oxygen^{53, 54}, surface defects⁵⁵, and oxygen vacancy are proposed⁵⁶⁻⁵⁸.

(2) Influence of the vacancy concentration on the interface carrier transport in Schottky junctions

We performed annealing on the fabricated ZnO nano/microwires at various temperatures: 200 °C, 300 °C, 400 °C, and 450 °C. The current-voltage (I - V) properties of both untreated and annealed nanowires were systematically investigated. **Figures R12a-e** illustrate the I - V curves for Schottky barrier devices incorporating ZnO nano/microwires, evaluated under two distinct compressive strain conditions: 0.00% and 0.10%. Notably, the I - V characteristics exhibit a notable piezotronic effect on the interface regulation across the different annealing temperatures. Under a positive bias, devices subjected to 0.10% strain consistently demonstrated higher current responses compared to the unstrained (0.00%) state. Conversely, the devices' responses to a negative bias were diminished under 0.10% strain, in alignment with the behavior of piezotronic effect. **Figure R12f** delineates the I - V characteristic trends across varying annealing temperatures, revealing an increasing response current up to the threshold of 400 °C. Beyond this peak, a reduction in current response at 450 °C suggests that the Schottky barrier height is influenced by the density of oxygen vacancies.

By correlating results from **Figures R9-R11**, we infer that a higher prevalence of positively charged oxygen vacancies corresponds with a reduction in the Schottky

barrier height at the interface, enhancing the current response observed in the I - V curves. Conversely, a lower vacancy concentration minimizes the influence on the interface barrier. Comprehensively summarized in **Figure R12g** are the current values under ± 1 V bias for the 0.00% and 0.10% strain conditions, reaffirming the established I - V behavior patterns. The current on/off ratio, charted in **Figure R12h**, decreases progressively with each escalation in annealing temperature up to 400 °C, followed by an uptick at 450 °C. The Schottky barrier height modulation, depicted in **Figure R12i**, echoes the current on/off ratio trend and is consistent with alterations in the annealing temperature. These results suggest that the influence of piezoelectric polarization charge on barrier height is mitigated at an annealing temperature of 400 °C, primarily owing to augmented oxygen vacancy concentration, which in turn augments the shielding of piezoelectric polarization charges. However, at 450 °C, a reduction in positively charged oxygen vacancies (Vo^{2+}) concentration lessens the shielding effect, thereby reinstating the regulation effect of piezoelectric polarization charge on the barrier, culminating in the heightened current on/off ratios and Schottky barrier heights.

To summarize, our experimental data indicates that elevating the annealing temperature from 200 °C to 400 °C diminishes the piezotronic effect, whereas at 450 °C, enhances the effect. The positively charged oxygen vacancies (Vo^{2+}) is proved the most factor. Additionally, we discovered that an increase in positively charged oxygen vacancies (Vo^{2+}) concentration reduces the Schottky barrier height at the interface, while a decrease in their concentration results in an increased barrier height.

(3) Influence of the vacancy concentration on regulation of electric stimulation

Our experimental data underscore that annealing fundamentally influences the interface barrier's recovery dynamics. Initially, we subjected ZnO nano/microwires to annealing at disparate temperatures: 200 °C, 300 °C, 400 °C, and 450 °C. These treated nano/microwires were then utilized to construct ZnO devices. As illustrated in **Figure R13a**, we exposed the devices to electric pulse stimulation, characterized by an 80 V amplitude and a 10 Hz frequency. **Figures R13b-f** profile the I - V characteristics of these devices both prior to and following a 60-second regimen of periodic electric pulses.

The resulting I - V curves unequivocally indicate that such electric pulse stimulation markedly amplifies the devices' current response. Capture in **Figure R13g** is the curve depicting the relationship between the current value at a 1 V positive bias and the annealing temperature. Meanwhile, **Figure R13h** charts the variations in current ratio before and after electrical stimulation as a function of annealing temperature. Notably, this relationship curve peaks at 400 °C, designating it as a critical inflection point, beyond which (at 450 °C) the current ratio significantly diminishes. **Figure R13i** encapsulates data on the recovery time of the device's current post electrical stimulation. Observations detailed here validate that electric pulse stimulation notably extends the duration required for the device to revert to its baseline current state.

Here, we give more explanation about the working principle. Based on previous research, the natural defects of oxygen vacancies are inevitably present during the preparation of ZnO nano/microwires¹⁹. Theoretically, oxygen vacancies can exist in different valence states (Vo^{2+} , Vo^+ and Vo^0), and at the interface between metal and zinc oxide, neutral oxygen vacancies (Vo^0) can easily transform into positively charged ionized oxygen vacancies (Vo^{2+})^{20, 21}. The formation of Schottky barriers is influenced by the density of oxygen vacancies near the interface (**Figure R14a**). A high electric field can exert a driving force on the positively charged ionized oxygen vacancies, leading to ionic migration at the interface²². In this work, continuous electric pulses can drive the migration of positively charged oxygen vacancies and their accumulation at the contact interface of the Ag/HfO₂/n-ZnO tunneling junction (**Figure R14b**), forming a narrow region of positive charge, bending the energy bands, and reducing the barrier height²⁰. After the pulse voltage is removed, the high concentration of oxygen vacancies gathered at the interface will undergo a slow diffusion process, gradually returning to their original state (**Figure R14c**).

Next, with the help of the band diagram in **Figure R14a-1** and **R14c-1**, we will discuss in detail the physical process of the aggregation and recovery of ionic oxygen vacancies at the tunneling junction interface after electric pulse voltage stimulation. In the initial state, the metal, insulator, and piezoelectric semiconductor contact to form a tunneling junction with a certain barrier height and width, and the concentration of ionic

oxygen vacancies near the interface is low (**Figure R14a-1**). When the device is stimulated by electric pulses for a certain period of time, the positively charged ionic oxygen vacancies will gather at the interface driven by the electric field, and the concentration of ionic oxygen vacancies at the interface will increase, leading to the decrease of the height ($\Delta\varphi_{\text{pulse}} = BH0 - BH1$) and width ($\Delta d_{\text{pulse}} = BW0 - BW1$) of the tunneling junction barrier (**Figure R14b-1**), where ‘ BH ’ represents barrier height and ‘ BW ’ represents barrier width. When the pulse voltage is removed, due to the imbalance of oxygen distribution caused by the external electric field ²³, the high concentration oxygen vacancy at the interface begins to diffuse to the interior of ZnO, and the barrier will gradually recover (**Figure R14c-1**). Because the oxygen vacancy is rapidly gathered to the tunneling junction interface under the drive of electric field force, and the recovery process of the barrier height depends on the slow diffusion of oxygen vacancy, which affected by the concentration gradient, so the recovery process of device performance after electric pulses stimulation needs a period of time. Furthermore, as the electric pulse voltage stimulation time increases, the density of ionic oxygen vacancies near the ZnO contact interface also increases, leading to a gradual decrease in the barrier within the tunneling junction area. Additionally, oxygen vacancies in low electric field areas exhibit slower movement compared to those in high electric field areas ²⁴. Consequently, a sufficiently strong electric field can induce the accumulation of high-density oxygen vacancies at the tunneling junction interface, thereby altering the effective barrier. This phenomenon explains why higher electric pulse voltages or longer stimulation times result in more pronounced increases in device current. Subsequently, after a certain duration of pulse voltage stimulation, a dynamic equilibrium is established between the migration of ionic oxygen vacancies to the tunneling junction interface driven by the electric field force and the diffusion away from the interface due to concentration differences of oxygen vacancies.

Subsequently, strain is initially introduced to the device without the application of an electric pulse stimulation, as illustrated in **Figure R14a-1** to **Figure R14a-2**. The modulation of the barrier height induced by the piezoelectric polarization charge resulting from the strain is labeled as $\Delta\varphi_{\text{piezo}}$ ($\Delta\varphi_{\text{piezo}} = BH0 - BH0'$). Following

this, strain is introduced to the device after the application of the electric pulse stimulus, as depicted in **Figure R14a-1** to **Figure R14a-2**. The modulation of the barrier height induced by the piezoelectric polarization charge resulting from the strain is labeled as $\Delta\varphi_{\text{piezo1}}$ ($\Delta\varphi_{\text{piezo1}} = BH1 - BH1'$). Through the introduction of electrical stimulation, we exploit the piezotronic effect to expand the tunable range of the tunneling junction barrier height from $\Delta\varphi_{\text{piezo}}$ to $\Delta\varphi_{\text{piezo1}}$, thus achieving the controllable piezotronic effect through electric pulses.

In summary, with the increase of post-annealing temperature, the vacancy Vo^{2+} concentration increases as a whole, which makes the device current increases (under a same bias) but weakens the interface engineering by the piezoelectric polarization and the electric pulse. These results from high vacancy Vo^{2+} concentration by post-annealing are very similar to the weakening of piezotronic effect caused by doping in the past. Your kindly suggestion has brought us a lot of inspiration and deepen the understanding of oxygen vacancy's influence. We are grateful to you again for your nice comment.

Figure R9. Relationship between the formation energy of oxygen vacancy in different states and the Fermi level.

Figure R10. EPR spectra of ZnO annealed in different temperatures.

Figure R11. PL spectra of ZnO annealed in different temperatures. a, Total PL spectra of ZnO in different annealing temperatures. **b-d,** PL spectra in UV region (**b**), blue region (**c**) and red region (**d**). The spectra are normalized by the intensity of near-band emission.

Figure R12. Characterization of the influence of annealing temperature on piezotronic properties. **a-e**, I - V curves for Schottky barrier devices incorporating untreated and annealed ZnO nano/microwires, evaluated under two distinct compressive strain conditions: 0.00% and 0.10%. The annealing temperatures are 200 °C, 300 °C, 400 °C, and 450 °C, respectively. **f**, I - V characteristic trends across varying annealing temperatures. **g**, Current values under ± 1 V bias for the 0.00% and 0.10% strain conditions. **h**, **i**, Current on/off ratio (**h**) and change of Schottky barrier height (**i**) as functions of different annealing temperatures under ± 1 V bias.

Figure R13. Carrier transport characteristics before and after electrical stimulation after annealing. **a**, Electric pulse characterized by an 80 V amplitude and a 10 Hz frequency. **b-f**, I - V characteristics before and after electrical stimulation under different heating temperature. **g**, The relationship between the current value at a 1 V positive bias and the annealing temperature. **h**, The variations in current ratio before and after electrical stimulation as a function of annealing temperature. **i**, The recovery time of the device's current after electrical stimulation

Figure R14. Polarization model and band diagram of barrier variation. **a**, Model of an MIS contact consisted of Ag, HfO₂ and *n*-type ZnO with tunneling junction contact. **a-1**, Corresponding band diagram with the barrier high of $BH0$. **a-2**, Regulation of energy band by piezoelectric polarization. **b**, Electric pulses can drive the migration of positively charged oxygen vacancies and their accumulation at the contact interface of the Ag/HfO₂/*n*-ZnO tunneling junction. **b-1**, Positively charged oxygen vacancies (Vo^{2+}) accumulated at interface result in the bend of energy band and SBH decrease (from $BH0$ to $BH1$). **b-2**, Regulation of energy band by piezoelectric polarization after electric pulses stimulation. **c**, Ionized oxygen vacancies diffuse away from Ag/HfO₂/*n*-ZnO interface slowly after withdrawing the treatment of electrical stimulation. **c-1**, Density of positively charged oxygen vacancies (Vo^{2+}) at the interface decrease, which results in the barrier to recover (from $BH1$ to $BH2$).

Reference:

1. Liu S, Wang L, Feng X, *et al.* Piezotronic tunneling junction gated by mechanical stimuli. *Advanced Materials* **31**, 1905436 (2019).

2. Yu Q, Ge R, Wen J, *et al.* Highly sensitive strain sensors based on piezotronic tunneling junction. *Nature communications* **13**, 778 (2022).
3. Shi J, Starr MB, Wang X. Band structure engineering at heterojunction interfaces via the piezotronic effect. *Advanced Materials* **24**, 4683-4691 (2012).
4. Wang ZL, Wu W. Piezotronics and piezo-phototronics: fundamentals and applications. *National Science Review* **1**, 62-90 (2014).
5. Wu W, Wang ZL. Piezotronics and piezo-phototronics for adaptive electronics and optoelectronics. *Nature Reviews Materials* **1**, 1-17 (2016).
6. Pan C, Dong L, Zhu G, *et al.* High-resolution electroluminescent imaging of pressure distribution using a piezoelectric nanowire LED array. *Nature Photonics* **7**, 752-758 (2013).
7. Yu R, Pan C, Wang ZL. High performance of ZnO nanowire protein sensors enhanced by the piezotronic effect. *Energy & Environmental Science* **6**, 494-499 (2013).
8. Cao X, Cao X, Guo H, *et al.* Piezotronic effect enhanced label-free detection of DNA using a Schottky-contacted ZnO nanowire biosensor. *ACS Nano* **10**, 8038-8044 (2016).
9. Bai S, Wu W, Qin Y, *et al.* High-performance integrated ZnO nanowire UV sensors on rigid and flexible substrates. *Advanced Functional Materials* **21**, 4464-4469 (2011).
10. Yang X, Dong L, Shan C, *et al.* Piezophototronic-effect-enhanced electrically pumped lasing. *Advanced Materials* **29**, 1602832 (2017).
11. Wu W, Wei Y, Wang ZL. Strain-gated piezotronic logic nanodevices. *Advanced materials* **22**, 4711-4715 (2010).
12. Yu R, Wu W, Ding Y, *et al.* GaN nanobelt-based strain-gated piezotronic logic devices and computation. *ACS Nano* **7**, 6403-6409 (2013).
13. Dan M, Hu G, Li L, *et al.* High performance piezotronic logic nanodevices based on GaN/InN/GaN topological insulator. *Nano Energy* **50**, 544-551 (2018).
14. Zhang S, Ma B, Zhou X, *et al.* Strain-controlled power devices as inspired by human reflex. *Nature Communications* **11**, 1-9 (2020).

15. Wu W, Wen X, Wang ZL. Taxel-addressable matrix of vertical-nanowire piezotronic transistors for active and adaptive tactile imaging. *Science* **340**, 952-957 (2013).
16. Meng J, Li Z. Schottky-contacted nanowire sensors. *Advanced Materials* **32**, 2000130 (2020).
17. Wang L, Wang ZL. Advances in piezotronic transistors and piezotronics. *Nano Today* **37**, 101108 (2021).
18. S.M. Sze KKN. *Physics of Semiconductor Devices*, 3rd edn. John Wiley & Sons (2006).
19. Meng J, *et al.* Triboelectric nanogenerator enhanced Schottky nanowire sensor for highly sensitive ethanol detection. *Nano Letters* **20**, 4968-4974 (2020).
20. Zhao L, *et al.* Combining triboelectric nanogenerator with piezoelectric effect for optimizing Schottky barrier height modulation. *Science Bulletin* **66**, 1409-1418 (2021).
21. Li H, *et al.* Triboelectric-polarization-enhanced high sensitive ZnO UV sensor. *Nano Today* **33**, 100873 (2020).
22. Zhang B, *et al.* Breath-based human-machine interaction system using triboelectric nanogenerator. *Nano Energy* **64**, 103953 (2019).
23. Ding Y, *et al.* Pyroelectric-field driven defects diffusion along c-axis in ZnO nanobelts under high-energy electron beam irradiation. *Journal of Applied Physics* **116**, 3435 (2014).
24. Jiang W, *et al.* Mobility of oxygen vacancy in SrTiO₃ and its implications for oxygen-migration-based resistance switching. *Journal of Applied Physics* **110**, 034509 (2011).
25. Arunkumar N, Senathipathi N, Dhanasekar S, *et al.* An ultra-low-power static random-access memory cell using tunneling field effect transistor. *International Journal of Engineering* **33**, 2215-2221 (2020).
26. Dai M, Chen H, Wang F, *et al.* Ultrafast and sensitive self-powered photodetector featuring self-limited depletion region and fully depleted channel with van der Waals contacts. *ACS Nano*, **14**, 9098-9106 (2020).

27. Lv L, Yu J, Hu M, *et al.* Design and tailoring of two-dimensional Schottky, PN and tunnelling junctions for electronics and optoelectronics. *Nanoscale* **13**, 6713-6751 (2021).
28. Peller D, Kastner LZ, Buchner T, *et al.* Sub-cycle atomic-scale forces coherently control a single-molecule switch. *Nature* **585**, 58-62 (2020).
29. Nahar R, Singh V, Sharma A. Study of electrical and microstructure properties of high dielectric hafnium oxide thin film for MOS devices. *Journal of Materials Science: Materials in Electronics* **18**, 615-619 (2007).
30. Gilmer D, *et al.* Compatibility of polycrystalline silicon gate deposition with HfO₂ and Al₂O₃/HfO₂ gate dielectrics. *Applied physics letters* **81**, 1288-1290 (2002).
31. Kang S, *et al.* Effect of deposition conditions of poly Si_{1-x}Ge_x films and Ge atoms on the electrical properties of poly Si_{1-x}Ge_x (x= 0, 0.6)/HfO₂ gate stack. *Journal of applied physics* **94**, 4608-4613 (2003).
32. Hinkle C, Fulton C, *et al.* Enhanced tunneling in stacked gate dielectrics with ultra-thin HfO₂ (ZrO₂) layers sandwiched between thicker SiO₂ layers. *Applied Surface Science* **234**, 240-245 (2004).
33. Gerritsen E, *et al.* Evolution of materials technology for stacked-capacitors in 65 nm embedded-DRAM. *Solid-State Electronics* **49**, 1767-1775 (2005).
34. Oba F, Togo A, Tanaka I, *et al.* Defect energetics in ZnO: a hybrid Hartree-Fock density functional study. *Physical Review B* **77**, 245202 (2008).
35. Agoston P, Albe K, Nieminen RM, *et al.* Intrinsic n-type behavior in transparent conducting oxides: a comparative Hybrid-functional study of In₂O₃, SnO₂, and ZnO. *Physical review letters*, **103**, 245501 (2009).
36. Reddy AJ, Kokila MK, *et al.* Structural, optical and EPR studies on ZnO:Cu nanopowders prepared via low temperature solution combustion synthesis. *Journal of Alloys and Compounds* **509**, 5349-5355 (2011).
37. Kaftelen H, Ocakoglu K, *et al.* EPR and photoluminescence spectroscopy studies on the defect structure of ZnO nanocrystals. *Physical Review B* **86**, 014113 (2012).

38. Evans SM, Giles NC, *et al.* Further characterization of oxygen vacancies and zinc vacancies in electron-irradiated ZnO. *Journal of Applied Physics* **103**, 043710 (2008).
39. Wang XJ, Vlasenko LS, *et al.* Oxygen and zinc vacancies in as-grown ZnO single crystals. *Journal of Physics D: Applied Physics* **42**, 175411 (2009).
40. Liu H, Zeng F, *et al.* Correlation of oxygen vacancy variations to band gap changes in epitaxial ZnO thin films. *Applied Physics Letters* **102**, 181908 (2013).
41. Li Y, Zu B, Guo Y, *et al.* Surface superoxide complex defects-boosted ultrasensitive ppb-level NO₂ gas sensors. *Small* **12**, 1420-1424 (2016).
42. Yu L, Guo F, Liu S, *et al.* Both oxygen vacancies defects and porosity facilitated NO₂ gas sensing response in 2D ZnO nanowalls at room temperature. *Journal of Alloys and Compounds* **682**, 352-356 (2016).
43. Wei XQ, Man BY, *et al.* Blue luminescent centers and microstructural evaluation by XPS and Raman in ZnO thin films annealed in vacuum, N₂ and O₂. *Physica B: Condensed Matter* **388**, 145-152 (2007).
44. Sambandam B, Michael RJ, *et al.* Oxygen vacancies and intense luminescence in manganese loaded ZnO microflowers for visible light water splitting. *Nanoscale* **7**, 13935-13942 (2015).
45. Børseth TM, Svensson BG, *et al.* Identification of oxygen and zinc vacancy optical signals in ZnO. *Applied Physics Letters* **89**, 262112 (2006).
46. Djurišić AB, Leung YH, *et al.* Green, yellow, and orange defect emission from ZnO nanostructures: Influence of excitation wavelength. *Applied Physics Letters* **88**, 103107 (2006).
47. Wang QP, Zhang DH, *et al.* Photoluminescence of ZnO films prepared by R. F. sputtering on different substrates. *Applied Surface Science* **220**, 12-18 (2003).
48. Kang HS, Kang JS, *et al.* Annealing effect on the property of ultraviolet and green emissions of ZnO thin films. *Journal of Applied Physics* **95**, 1246-1250 (2004).
49. Tong YH, Liu YC, *et al.* The optical properties of ZnO nanoparticles capped with polyvinyl butyral. *Journal of Sol-Gel Science and Technology* **30**, 157-161 (2004).

50. Vanheusden K, Seager CH, *et al.* Correlation between photoluminescence and oxygen vacancies in ZnO phosphors. *Applied Physics Letters* **68**, 403-405 (1996).
51. van Dijken A, Meulenkamp EA, *et al.* The kinetics of the radiative and nonradiative processes in nanocrystalline ZnO particles upon photoexcitation. *The Journal of Physical Chemistry B* **104**, 1715-1723 (2000).
52. Meyer B, Marx D, Density-functional study of the structure and stability of ZnO surfaces. *Physical Review B* **67**, 035403 (2003).
53. Greene LE, Law M, *et al.* Low-temperature wafer-scale production of ZnO nanowire arrays. *Angewandte Chemie, International Edition* **42**, 3031-3034 (2003).
54. Kwok WM, Djurišić AB, *et al.* Time-resolved photoluminescence study of the stimulated emission in ZnO nanoneedles. *Applied Physics Letters* **87**, 093108 (2005).
55. Ong HC, Du GT, The evolution of defect emissions in oxygen-deficient and -surplus ZnO thin films: The implication of different growth modes. *Journal of Crystal Growth* **265**, 471-475 (2004).
56. Kumar V, Swart HC, *et al.* Origin of the red emission in zinc oxide nanophosphors. *Materials Letters* **101**, 57-60 (2013).
57. Studenikin SA, Golego N, *et al.* Fabrication of green and orange photoluminescent, undoped ZnO films using spray pyrolysis. *Journal of Applied Physics* **84**, 2287-2294 (1998).
58. Kong YC, Yu DP, *et al.* Ultraviolet-emitting ZnO nanowires synthesized by a physical vapor deposition approach. *Applied Physics Letters* **78**, 407-409 (2001).

Reviewer #2 (Remarks to the Author):

The authors reported a novel approach of interface engineering by applying electric pulses on Ag/HfO₂/n-ZnO piezotronic tunneling junction device. The interface barrier height can reversibly be tuned up to 168.11 meV by 80 V electric pulses. And the piezotronic modification on interface barrier tuned by electric pulse can be up to 148.81 meV under a strain of 1.34%. The article demonstrated sufficient experimental results

on the phenomenon of tunable interface barrier height by the stimulation of electric pulse and bending-induced strain, and provided a reasonable theoretical analysis of interfacial engineering. It shows a novelty in the further investigation of piezotronics, and gives a guide to design and optimize novel piezotronic devices based on low-dimensional materials (*e.g.*, ZnO, GaN, MoS₂). The reviewer would like to recommend it be published in Nature Communications, however, some critical issues remain to be addressed.

Response:

We sincerely appreciate your acknowledgment of the novelty and importance of our research, particularly your remark: "It shows a novelty in the further investigation of piezotronics." And we have carefully revised the manuscript according to your very valuable comments and suggestions.

Question #1

In the Introduction, the authors introduced "flexoelectronics and ferroelectric electronics" and discussed their working principles in Fig. 1. However, only piezotronics was investigated in the manuscript. It seems to be redundant to talk about flexoelectronics and ferroelectric electronics.

Response:

We appreciate you very much for this constructive comment and kindly reminder. In this work, an interfacial engineering approach using the electric pulse-induced spatial distribution of oxygen vacancies to reversibly and accurately tune the piezotronic modification has been proposed. To test the feasibility of this approach, we select an Ag/HfO₂/n-ZnO piezotronic tunneling junction as a model of the piezoelectric polarization system, which utilizes the piezo-potential generated in the piezoelectric semiconductor to modulate the interface barrier and control the charge carrier transport characteristics¹. The coupling effect enables us to find that the oxygen vacancy migration induced by the electric field regulating the interface properties can be extended to even other interface engineering by local polarization, like flexoelectronics and ferroelectric electronics²⁻⁶, to improve the performance of the tunable electronics

and expand their application scenarios. Considering the theoretical and experimental investigations indeed involve only the piezotronics (the piezotronic tunneling junction) in our research, and leaving only the discussion on piezotronics will also make the introduction and Figure 1 more logical and not easy to cause misunderstanding, the schematic diagram of the interface engineering by local polarizations in flexoelectronics and ferroelectric electronics in Figure 1 has been carefully revised. We thank the reviewer again for this important suggestion, which helps to improve the quality of our work.

As the reviewer's suggestion, we have revised Figure 1 without the working principles of the flexoelectronics and the ferroelectric electronics, with the corresponding contents in the Introduction section modified and marked in red. Figure 1 in the revised manuscript is also listed as the following **Figure R15** for your convenience.

Figure R15. *(Figure 1 in the revised manuscript)* Working principle of electric pulse-tuned piezotronic effect on Ag/HfO₂/n-ZnO piezotronic tunneling junction (PTJ).

a, Energy band diagrams of interface engineering by local polarizations in piezotronics.
b, Effect of electric pulse on the oxygen vacancies in semiconductors (left) and

corresponding modification of interface barrier height ($\Delta\varphi_{\text{pulse}}$) (right). **c**, Introducing a strain (i) and an electric pulse (ii) to the Ag/HfO₂/n-ZnO PTJ to tune the interface barrier. **d**, Corresponding energy bands of Ag/HfO₂/n-ZnO PTJ in **c**. (i) The modulation of interface barrier by piezoelectric polarization ranges from $\varphi_0/2$ to $(\varphi_0 - \Delta\varphi_{\text{piezo}})/2$. (ii) The electric pulse-induced oxygen vacancy migration changes the interface barrier by $\Delta\varphi_{\text{pulse}}/2$, and thus tune the piezotronic modulation range from $(\varphi_0 - \Delta\varphi_{\text{pulse}})/2$ to $(\varphi_0 - \Delta\varphi_{\text{pulse}} - \Delta\varphi'_{\text{piezo}})/2$.

Question #2

Why did use an electric pulse in the investigation on piezotronics? The authors need to give more discussion on the advantages or disadvantages of piezotronic devices.

Response:

We thank you very much for this valuable comment. The field of piezotronics is currently experiencing rapid development and garnering widespread attention. Given their status as core components, the performance and characteristics of piezotronic devices directly influence the progress of this field. Therefore, research and optimization of piezotronic devices, especially concerning their interface properties, hold significant importance. Subsequently, we will amalgamate the advantages and disadvantages of piezotronics to elucidate the rationale for employing electric pulses in piezotronics research, emphasizing the innovative aspects of this endeavor. To provide a more cohesive response to this question, we will address the topic in the following three sections.

(1) Dependence of piezotronic devices on interface barriers

Here, we will discuss the advantages and disadvantages of piezotronic devices based on their dependence on the interface barrier.

Piezotronic devices base on piezotronic effect offers significant advantages. Primarily, the piezotronic effect mainly regulates the electrical transport of the interface, which typically exhibit high sensitivity⁷⁻⁹. Piezotronic devices are often utilized for actively sensing mechanical stimuli. This allows for the regulation and optimization of

detector performance through the application of driving forces, as observed in applications such as light-emitting diodes ¹⁰, biomolecular detection ^{11, 12}, ultraviolet detection ¹³, and lasers ¹⁴, *etc.* Additionally, the measurement of electrical properties of devices can be utilized to detect mechanical stimuli, as demonstrated in logic units ¹⁵⁻¹⁷, vibration sensing ¹⁸, and tactile sensing ¹⁹. In summary, the advantage of piezoelectric polarization interface modulation lies in its capability to achieve fine regulation and optimization of the performance of piezotronic devices.

The basic structure of piezotronic devices, as shown in **Figure R16**, is mainly composed of piezoelectric semiconductor and electrodes at both ends. In this structure, the interface barrier forms between the piezoelectric semiconductor and the electrode, and it is regulated by the piezoelectric polarization charges induced by strain and the corresponding piezoelectric potential, thereby controlling the device. The performance of piezotronic devices is dependent on the engineered interface barrier and is highly sensitive to its characteristics. This sensitivity arises from the fact that the piezotronic effect is produced by strain-induced piezoelectric charges that modulate the energy band at the interface, resulting in charge rearrangement at the interface and influencing the transport of carriers ⁷⁻⁹. Consequently, the construction and precise control of the interface barrier play a crucial role in the operation of piezotronic devices, as even minor adjustments to the interface barrier can have a substantial impact on device performance.

The disadvantage of piezotronic devices lies in the fact that once they are fabricated, the internal interface barrier is fixed, leading to the inability to tune the device performance. This means that the range of response current for the same strain will also be fixed, which may render piezotronic devices unsuitable for certain applications. Since polarized interface engineering highly depends on the nature of the interface barrier, once the interface potential barrier is established, its fixed value and uncontrollability will limit the performance and application scenarios of polarization interface engineering.

If tunable performance of piezotronic devices can be achieved, then the research on interface engineering of piezoelectric polarization in semiconductor devices will

have significant importance for their applications in the fields of electronics, optoelectronics and catalysis. In addition, polarization interface engineering is very important for tunable electronic devices (including piezotronic devices, and even flexoelectronic devices and ferroelectric electronic devices) ^{1, 20, 21}. Therefore, the search for interface barrier modulation strategies suitable for various application scenarios is of great scientific and engineering significance.

(2) Novelty of piezotronic tunneling junction with tunable interface barrier in this work

The novelty of this work lies in utilizing electric pulse stimulation to achieve tunable barrier of the tunneling junction and thus regulate the piezotronic effect within the junction. Under the driving force of the electric field produced by electric pulse stimulation, positively charged oxygen vacancies are gathered towards the interface, enabling the modulation of the barrier and thereby altering the range over which piezoelectric polarization can be controlled.

Specifically, in the process of synthesizing ZnO nano/microwires, oxygen vacancies naturally exist in their respective valence states (Vo^{2+} , Vo^+ , Vo^0) ²²⁻²⁴. These vacancies influence the formation of the Schottky barrier at the metal and zinc oxide interface. Force generated by a high electric field drives these vacancies to migrate towards the interface ²⁵. In this study, continuous electric pulse stimulation triggers the migration of these vacancies to the contact interface of the tunnel junction, adjusting the barrier height, hence controlling the performance of piezotronic devices. As the oxygen vacancies rapidly accumulate at the tunnel junction interface under the drive of the electric field, and because the recovery process of this barrier height depends on the slow diffusion of these vacancies ^{26, 27}, the control over the barrier by way of electric pulse stimulation remains for a certain duration. In other words, a period of time is needed for the device's performance to recover post-electric pulse stimulation.

As presented in the manuscript, we fabricated an Ag/HfO₂/n-ZnO piezotronic tunneling junction as a demonstration illustrated in **Figure R17a**. As bending the substrate, a strain will be induced along ZnO nanowire to make it produce opposite

piezoelectric charges at both terminals of ZnO (**Figure R17c-i**) that acts as a ‘gate’ voltage to control the charge transport of interface (piezotronic effect). As shown in **Figure R17a-ii**, with electric pulse stimulation, oxygen vacancies migration occurs in ZnO nanowire to tune the barrier height at the tunneling interface, which will then determine the range of working current modulated by piezotronic effect. By using a +80 V electric pulse, this work shifted the operating current range of Ag/HfO₂/n-ZnO piezotronic tunneling junction strain sensor from 0.068~18.3 nA (**Figure R17b**) to 43.7~71.9 nA (**Figure R17c**). So, electric pulse can effectively tune the piezotronic effect in tunneling junctions.

Furthermore, the polarization interface engineering (including piezotronic effect) is very important for tunable electronic devices, and the search for interface barrier modulation strategies suitable for various application scenarios is of great scientific and engineering significance. Considering the obvious regulation of electric pulse on polarization interface engineering (piezotronic effect), the electric pulse-tuned piezotronic effect for interface engineering studied in this work should be important.

(3) Necessity of using electric pulses to study piezotronics

There are several key reasons and advantages for using electric pulses (or pulse voltage) instead of direct current voltage to stimulate piezotronic devices²⁸⁻³⁰. ① Avoiding thermal losses and degradation: Direct current voltage generates continuous current in electronic components, producing heat energy, which may lead to material performance degradation or thermal degradation over time. Electric pulses, on the other hand, can provide energy in a shorter time, reducing heat accumulation. ② Improving efficiency: Electric pulses can deliver high peak power in a short time, thereby more efficiently exciting devices, especially in applications requiring rapid response. ③ Reducing power consumption: Compared to continuous direct current voltage, electric pulses provide electrical energy only when stimulation is needed, significantly reducing the average power consumption of devices. ④ Control and precision: The duration, magnitude, and frequency of electric pulses can be precisely controlled, enabling precise signal modulation and processing. ⑤ Device protection: Higher direct current

voltage may damage piezoelectric materials, while pulses can use higher peak voltage without causing long-term penetrative damage to the devices. ⑥ Enhancing signal resolution: In some applications, using pulse signals can improve the resolution of detection signals by optimizing the time resolution of the signal through adjustment of pulse width and shape. ⑦ Interface state control: Rapid electric field changes generated by pulse voltage at the interface of piezoelectric materials provide better control over interface charge states and charge transfer processes. ⑧ Reducing electrode and material degradation: Direct current voltage may cause chemical degradation at the electrode and material interface. Pulse voltage, due to its intermittence, reduces the likelihood of this chemical degradation.

Based on the above discussions, it is evident that the innovation of this research lies in the utilization of electric pulse stimulation to manipulate the tunable interface barrier of the tunneling junction, thereby controlling the piezotronic effect within the junction. We appreciate your valuable suggestion once again. To enhance the comprehensiveness and clarity of our study, we have added the relevant details in the Supplementary Information, which have been highlighted in red.

Figure R16. The basic typical structure of piezotronic devices.

Figure R17. Regulation of piezotronic effect by electric pulse. **a**, Introducing a strain (i) and an electric pulse (ii) to the Ag/HfO₂/n-ZnO piezotronic tunneling junction to tune the interface barrier. **b**, **c**, Strain-dependent *I-V* curves of piezotronic tunneling strain sensor before (**b**), immediately after (**c**) electric pulse of 80 V for 30 s.

Question #3

The electric pulse has a large amplitude of 80 V and a long period. Actually, the electric pulse with a range of 1 ns to 100 ms would be widely applied on various electronic devices. Is it possible to reduce the period time of the pulse, making it equivalent to that of a long pulse condition? And the different frequency of pulse needs to be conducted in the experiments. In addition, the voltage amplitude of 80 V is high enough and would easily cause device breakdown.

Response:

We thank you very much for these detailed, in-depth, and valuable suggestions. The period time and frequency are the two important parameters of electric pulse that determines the strength of provided driving force of vacancy redistribution. And the electric pulse with large amplitude is indeed easy to cause device breakdown. According to your suggestions, we carried out experiments to study the influence of the

period time, frequency, and amplitude of electric pulse on the electrical transport of the Ag/HfO₂/n-ZnO devices. In order to answer the questions clearly, we describe the relevant experimental data and conclusions in detail as follows (1) and (2), respectively. Since we think this part of discussion can help us to better understand the process of electric pulse regulating vacancy movement, and improve the quality and readability of our manuscript, we have added them into the revised Supplementary Information and marked them in red.

(1) Is it possible to reduce the period time of the pulse, making it equivalent to that of a long pulse condition? And the different frequency of pulse needs to be conducted in the experiments.

Here, by changing the period time and frequency of the electric pulses, we performed and analyzed the corresponding current response characteristics in **Figures R18-20**. It is worth noting that during the experiment, a set procedure was used to perform electric pulse stimulation on the Ag/HfO₂/n-ZnO device, and the current signals were captured by the electrical measurement. **Figure R18a** shows the electric pulse voltage of 40 V on the Ag/HfO₂/n-ZnO device, the duration of each pulse is 0.05 s, 0.36 s and 2.35 s, respectively, and the time interval between two electric pulses is about 10.96 s. The corresponding current responded to **Figure R18a** are shown in **Figure R18b** and **R18c**, in which the current flowing through the device is gradually enhanced from 9.45 to 12.00 μ A with the increase of the duration of each pulse, and shows a trend of attenuation with time after the electric pulse stimulating (**Figure R18c**). Based on the data in **Figure R18c**, through the linear fitting analysis and the slope of the recovery curve of the current with time in the period numbered '1' to '3' in **Figure R18d**, in which the recovery slopes are -0.07759, -0.04626 and -0.0249, respectively, we found that the recovery rate of current gradually decreases with the increasing period time of the electric pulse voltage. In general, increasing the duration of the pulse increases the current response and prolonging the recovery time.

Next, we use the electric pulse voltage of 80 V to study the influence of the pulse frequency on the electrical transport of the device. **Figures R19** and **R20** show the

electric pulse stimulation and current response with different pulse frequency of 0.5 Hz, 1 Hz, 5 Hz, 10 Hz and 20 Hz, respectively. And the changing of pulse frequency conducted in the experiments could be discussed in two cases.

For the first case, in **Figure R19**, the pulse frequency is regulated by changing the time interval between two electric pulses, and the duration of each pulse remains constant. At each frequency, the current response gradually increases as the periodic pulse continues, which is the result of the decrease of the interface barrier height due to the accumulation of vacancies near the interface driven by the electric field. Obviously, as shown in **Figure R19**, as the frequency of electric pulses decreases, the interval between two adjacent electric pulses becomes larger and the current response decreases. This can be explained by the fact that the increasing interval between the two electric pulses causes the vacancy to accumulate less at the interface in unit time, resulting in the weakened current response.

For the second case, the frequency is regulated by changing the duration of the electric pulse (**Figure R20**), and the interval between electric pulses remains constant. From the data analysis in **Figure R18**, we already know that increasing the duration of the pulse can increase the current response. Obviously, as the pulse frequency decreases and the duration of the pulse increases, the current response increases in **Figure R20**. This phenomenon can be explained by the increase in the duration of the electric pulse leading to an increase in the accumulation of vacancies at the interface in unit time²²⁻²⁷, thus enhancing the current response.

Thus, by analyzing the current response at different frequencies of electric pulse, we can know that it is possible to reduce the period time of the pulse to make it equivalent to that of a long pulse condition. Your kindly suggestion is of great help to us, so that we can better understand the process of electric pulse regulating vacancy movement.

Figure R18. Current response and recovery characteristics regulated by the electric pulse duration. **a**, Electric pulse voltage of 40 V on Ag/HfO₂/n-ZnO device with the duration of each pulse of 0.05 s, 0.36 s and 2.35 s, respectively, and the time interval between two electric pulses is about 10.96 s. **b**, **c**, Current response corresponding to (a). **d**, Three current recovery curves numbered ‘1’ to ‘3’ in (c), in which the recovery slopes are -0.07759, -0.04626 and -0.0249, respectively.

Figure R19. Electric pulse stimulation and current response of the device with different pulse frequency of 0.5 Hz, 1 Hz, 5 Hz, 10 Hz and 20 Hz, respectively. The frequency is regulated by changing the time interval between two electric pulses.

Figure R20. Electric pulse stimulation and current response of the device with different pulse frequency of 0.5 Hz, 1 Hz, 5 Hz, 10 Hz and 20 Hz, respectively. The frequency is regulated by changing the duration of the pulse.

(2) In addition, the voltage amplitude of 80 V is high enough and would easily cause device breakdown.

The electric pulse with large amplitude is indeed easy to cause device breakdown. It

should be mentioned that during the experiment, the Ag/HfO₂/n-ZnO device used in our work is not damaged by the periodic pulse voltage of 80 V, which still work normally after the pulse voltage stimulation. Besides, we also characterized the electrical transport characteristics under the periodic electric pulse stimulation with the frequency of 10 Hz and the amplitude of 70 V, 80 V, 100 V, 120 V and 150 V, respectively. **Figure R21** shows the electric pulse voltage and the corresponding current response of the device. As can be seen from the figure that under the pulse voltage stimulating from 70 V to 120 V (**Figure R21a-d**), the current response of the device all increase within 60 s (**Figure R21a-1 to R21d-1**). However, when the amplitude of the electric pulse that stimulates the device is 150 V (**Figure R21e**), although the corresponding current response of the device increases within 0.3 s and reaches 118 μ A, then the current response begins to show a decaying trend. And when the periodic pulse voltage lasts for 60 s, the current response has decayed to 71.4 μ A. This indicates that the pulse amplitude of 150 V is high enough to cause device breakdown. In addition, **Figure R21** shows that the performance of the device is not attenuated under the stimulation of 80 V electric pulses.

Figure R22a shows the I - V characteristics after the electric pulse stimulation for 60 s with different pulse amplitudes of 0 V, 70 V, 80 V, 100 V, 120 V and 150 V, respectively in **Figure R21a-d**. Similar to the phenomenon shown in **Figure R21**, when the amplitude of the stimulating pulse voltage increases from 0 V to 120 V in **Figure R22a**, the electrical transport characteristics of the device continue to increase, and after the stimulation of 150 V pulse voltage, the electrical transport characteristic of the device is attenuated. **Figure R22b** shows the current value at +3 V after the electric pulse stimulation, which is extracted from **Figure R22a**. It can be obviously observed that the current value drops significantly at the pulse amplitudes of 150 V.

Based on above experimental results, we can conclude that the Ag/HfO₂/n-ZnO devices used in this work can withstand electric pulse stimulations under 120 V (for 60 s at 10 Hz). We are grateful to you for reminding us to measure whether the devices are broken down during electric pulse stimulations.

Figure R21. Electric pulse voltage and the corresponding current response of the device. a-e, Amplitude of the electric pulse voltage stimulating the device is 70 V, 80 V, 100 V, 120 V and 150 V, respectively. Both of the time interval between electric pulses and the pulse duration are 0.05 s. **(a-1)-(e-1),** Current response corresponding to **(a-e)** within 60 s.

Figure R22. I - V characteristics after the electric pulse stimulation. a, I - V characteristics after the electric pulse stimulation for 60 s with different pulse amplitudes of 0 V, 70 V, 80 V, 100 V, 120 V and 150 V, respectively. **b**, Current value at +3 V obtained from a. When the pulse voltage amplitude increases from 0 V to 120 V, the electrical transport characteristics of the device continue to increase, and it drops significantly at the pulse amplitudes of 150 V. This shows that under the stimulation of 80 V electric pulses, the device is in good condition.

Question #4

AFM has the ability to implement the electric pulse mode. Could it conduct in-situ measurement of tunable interfacial barrier height by AFM?

Response:

We highly appreciate you for this constructive and insightful suggestion, which have significantly contributed to the depth of our investigation. Actually, as shown in Figure 2 of the manuscript, electric pulse stimulation was initially applied to the ZnO nanowires, followed by an analysis utilizing scanning Kelvin probe force microscopy (KPFM model in AFM, Cypher ES, Asylum Research) to assess their surface potential. The observed decrease in the surface potential over time, as depicted in Figure 2g-j, corroborated our hypothesis: oxygen vacancies migrate towards the interior of ZnO after stimulation, influenced by concentration gradients. The result of this KPFM measurement can indirectly indicate the tunable interfacial barrier height.

Following your kindly reminder, we further explored the tuning phenomenon

using in situ atomic force microscopy to verify the modulation of electric pulse on the interface barrier height. Introducing programmed pulse voltages (in the piezo-response force microscope, PFM model in AFM) and employing conductive atomic force microscope (CAFM model in AFM), we are able to stimulate the nanowires firstly and then perform subsequent *I-V* characterizations in situ. **Figure R23a** presents the optical image capturing the ZnO nanowire during probe testing. **Figure R23b-e** delineate the administered periodic pulse voltages, set at amplitudes of 4 V, 6 V, 8 V, 10 V, and maintained at a frequency of 100 Hz. Following a 10 s duration of these voltages, we carried out the *I-V* characteristics of the nanowire. As indicated by the *I-V* curve in **Figure R23f**, a clear presence of enlarged electrical transport of the nanowire was observed, with the effect intensifying alongside the increment of pulse voltage amplitudes. This result supports that electric pulse stimulation alters the oxygen vacancy distribution, effectively tunes the interface barrier, and thus achieves tunable interfacial barrier height.

This nice suggestion, together with your previous Question #4, has brought us lots of inspiration and spawned new research ideas. Because the piezoelectric polarization mainly occurs just at the interface, it will be affected by the vacancy distribution, but the effect is not huge enough. In contrast, the flexoelectric polarization is distributed in a certain width of space, which may be more affected by the vacancy distribution (which is also in a certain width of space). We believe that it should be very interesting to investigate the electric pulse-tuned flexoelectric effect. Thank you very much again for this valuable and inspirable suggestion.

Figure R23. In-situ measurement of tunable interfacial barrier height by AFM. a, Optical image capturing ZnO nanowire during probe testing. **b-e,** Applying periodic pulse voltages at amplitudes of 4 V, 6 V, 8 V, 10 V, and frequency of 100 Hz on the ZnO nanowire. **f,** In-situ measurement of *I-V* characteristics of the ZnO nanowire with various electric pulse voltages.

Question #5

What is the thickness of ALD HfO₂? The authors claimed “when an electric pulse is applied to cause oxygen vacancy migration inside the ZnO nanowire, there exists accumulation of oxygen vacancies near the left interface barrier”. If the HfO₂ thickness is less than 3-5 nm, tunneling is possible as the dominant mechanism for interfacial engineering. Or the HfO₂ layer would have some defects that could play an important role in trapping electrons from the ZnO nanowire. Please give more discussion about that.

Response:

We are grateful to you for these kindly reminders and constructive suggestions. To answer these questions more clearly, we divide them into three parts as follows (1), (2) and (3).

(1) What is the thickness of ALD HfO₂?

Thanks very much for your kindly reminder. In this study, an HfO₂ insulation layer with a thickness of about 1.8 nm was precisely grown on ZnO nanowire by atomic layer deposition (ALD). We used high-resolution transmission electron microscopy (HRTEM) to characterize the thickness of HfO₂ layer. The scanning electron microscopy (SEM) image in **Figure R24a-i** shows that the uniform ZnO nanowire with a typical hexagonal geometry has a clean surface. The HRTEM image of HfO₂/n-ZnO (**Figure R24a-ii**) shows the HfO₂ thickness of about 1.82 nm. **Figure R24b** gives the distribution of HfO₂ thickness along the *x*-axis labeled in **Figure R24a-i** and indicates a uniform HfO₂ thickness of about 1.8 (± 0.3) nm. The inset shows the statistical distribution of the insulation thickness.

Here, we will explain why the 1.8 nm thick insulation was chosen. In order to demonstrate the influence of insulation layer thickness on the performance of piezotronic tunneling devices, Ag/HfO₂/n-ZnO devices with HfO₂ thickness of 0.0 nm, 0.4 nm, 1.1 nm, 1.8 nm, 2.5 nm, 3.6 nm, and 7.3 nm were prepared, respectively. The piezotronic modifications of electrical transport of 140 devices (20 devices of each thickness) under 0.10% tensile strain was studied. The experimental results show that the device with 1.8 nm HfO₂ exhibit the highest current on/off ratio, which makes us choose 1.8 nm HfO₂ as the insulation layer of piezotronic tunneling junctions.

Figure R25 shows the statistical distributions of on-state and off-state currents of 140 Ag/HfO₂/n-ZnO devices (20 devices of each thickness) with 0.0 nm, 0.4 nm, 1.1 nm, 1.8 nm, 2.5 nm, 3.6 nm, and 7.3 nm HfO₂. By summarizing and analyzing the test data, we can obtain the average on-state current (**Figure R25a** and **R25b**) and off-state current (**Figure R25c** and **R25d**) of these devices with different HfO₂ thicknesses. It can be seen from the figure that with the decrease of HfO₂ thickness, both the on-state current and off-state current increase. We further calculated the statistical distribution of the current on/off ratio of these devices, as shown in **Figure R26a**. It can be found that the maximum current on/off ratio occurs at the insulation layer thickness of about 1.8 nm (**Figure R26b**).

Based on the above consideration, we chose HfO₂ with a thickness of ~1.8 nm as

the insulation layer to fabricate the piezotronic tunneling junction in this work.

Figure R24. Morphology characterization of HfO₂/n-ZnO nanowire and thickness characterization of deposited HfO₂ layer. a-i, SEM image of the ZnO nanowire with a diameter of about 8 μm, showing a typical hexagonal geometry and a clean surface. a-ii, High-resolution TEM image of HfO₂ layer deposited on ZnO surface. b, Thickness of HfO₂ along the x-axis labeled in a-i, showing the HfO₂ thickness of about 1.8 (±0.3) nm. The inset shows the statistical distribution of the insulation thickness.

Figure R25. On-state and Off-state currents of Ag/HfO₂/n-ZnO devices. a, c,

Statistical distribution of the On-state and Off-state currents of 140 Ag/HfO₂/ZnO devices (20 devices for each thickness) with 0 nm, 0.4 nm, 1.1 nm, 1.8 nm, 2.5 nm, 3.6 nm and 7.3 nm thick HfO₂. **b**, **d**, On-state current (**b**) and Off-state current (**d**) as a function of HfO₂ thickness. The error bars denote standard deviations of the mean.

Figure R26. The current on/off ratio of Ag/HfO₂/n-ZnO devices. **a**, The statistical distribution of current on/off ratios of Ag/HfO₂/n-ZnO devices with 0 nm, 0.4 nm, 1.1 nm, 1.8 nm, 2.5 nm, 3.6 nm and 7.3 nm thick HfO₂. **b**, The current on/off ratio as a function of HfO₂ thickness. The error bars denote standard deviations of the mean.

(2) The authors claimed “when an electric pulse is applied to cause oxygen vacancy migration inside the ZnO nanowire, there exists accumulation of oxygen vacancies near the left interface barrier”. If the HfO₂ thickness is less than 3-5 nm, tunneling is possible as the dominant mechanism for interfacial engineering.

Thank you for this valuable comment. We applied a certain frequency of electric pulses to the device first, then test its *I-V* characteristics, and study the modulation of piezotronic effect by the electric pulses. There are two processes in the experiments. The first process involves the stimulation of electric pulse on tunneling junctions, which drives the vacancy migration inside the ZnO. In this process, there are indeed some carriers (electrons) tunneling from the electrode (Ag) to the semiconductor (ZnO) to neutralize the migration of vacancies. This process in turn can also indicate that the vacancy migration does change the interface energy band, particularly the interface barrier height and width. This can be analogous to the formation process of the Schottky

barrier depicted in the Chapter 3 of the book ‘Physics of Semiconductor Devices’ by S. M. Sze, Y. Li and K. K. Ng²¹. It is worth noting that we suggest there is few or even no vacancy passing through the tunneling junction from ZnO into Ag. Our judgment is based on the recoverability and repeatability of the I - V characteristics after electric pulse stimulations (Figure 2f and Figure 3b-d in the manuscript). If the vacancy is indeed tunneled in the electric pulse stimulation, then after removing the electric pulse stimulation, the vacancy will not have enough energy to tunnel back to ZnO, which will lead to a non-recoverable I - V characteristic of Ag/HfO₂/ n -ZnO. The second process involves the piezotronic effect on the post-stimulated Ag/HfO₂/ n -ZnO. In this process, the tunneling is also the dominant transport, which is modulated by the strain-induced piezoelectric polarization and the post-stimulation-induced vacancy migration.

In order to reveal the regulation process more clearly, we describe the above two processes in detail as follows.

When discussing the transport mechanism of charge carriers, it is essential to consider the process of electric field driving oxygen vacancies to move towards the tunneling junction interface, as well as the mechanism of electric pulse modulation of the tunneling junction interface barrier. Natural defects of oxygen vacancies will inevitably occur in ZnO nano/microwires during the preparation process²². In theory, an oxygen vacancy can present different valence states (Vo^{2+} , Vo^+ and Vo^0), while the neutral oxygen vacancy (Vo^0) at the contact interface between metal and zinc oxide can easily be converted into a positively charged ionized oxygen vacancy (Vo^{2+})^{23,24}. The formation of Schottky junction is affected by the density of oxygen vacancy (Vo) near the interface. A high electric field can exert a driving force on the positively charged ionic oxygen vacancies, resulting in ion migration at the interface²⁵. In this work, continuous electric pulses can drive positively charged oxygen vacancies to migrate and accumulate at the contact interface of the tunneling junction (Figure R27a), forming a narrow space positive charge area, bending the energy band, and reducing the barrier height of tunneling junction²⁶. As a result, when the piezotronic device is stimulated by an electric pulse, the response current will increase, incorporating a short-term tunneling current (which will disappear as the electric pulse is removed).

Next, with the help of the band diagram in **Figure R27b**, we will discuss in detail the physical process of the aggregation and recovery of ionic oxygen vacancies at the tunneling junction interface after electric pulse stimulation. In the initial state, the metal, insulator, and piezoelectric semiconductor contact to form a tunneling junction with a certain barrier height and width, and the concentration of ionic oxygen vacancies near the interface is low (i-1). When the device is stimulated by electric pulses for a certain period of time, the positively charged ionic oxygen vacancies will gather at the interface driven by the electric field, and the concentration of ionic oxygen vacancies at the interface will increase, leading to the decrease of the height and width of the tunneling junction barrier (ii-2). When the pulse voltage is removed, due to the imbalance of oxygen distribution caused by the external electric field ²⁶, the high concentration oxygen vacancy at the interface begins to diffuse to the interior of ZnO, and the tunneling junction barrier will gradually recover to its original state (i-1). Due to the rapid accumulation of oxygen vacancies at the tunneling junction interface under the driving force of the electric field, the recovery process of the barrier height and width relies on the slow diffusion of oxygen vacancies. Therefore, the restoration of device performance after electric pulse stimulation requires some time. In other words, the modulation of the barrier by electric pulses exhibits a certain level of persistence. This represents the long-term effect induced by electric pulse stimulation in tunneling junction devices.

In addition, during a certain electric pulse voltage stimulation time, the density of ionic oxygen vacancies near the ZnO contact interface increases with the increase of stimulation time, and the barrier in the tunneling junction area will gradually decrease. Moreover, the oxygen vacancy in the low electric field area moves more slowly than that in the high electric field area ²⁷, so a strong enough electric field can induce the accumulation of high-density oxygen vacancy on the tunneling junction interface, which leads to the change of the effective barrier, including changes in barrier height and width. This explains the phenomenon where the device current increases more significantly as the electric pulse voltage becomes higher or the stimulation time becomes longer. It's important to note that the device current here includes tunneling

current. At the same time, after a certain time of pulse voltage stimulation, a dynamic balance will be reached between the migration of ionic oxygen vacancies to the tunneling junction interface caused by electric pulse stimulation and the diffusion away from the tunneling junction interface caused by the concentration difference of oxygen vacancies. With the increase of stimulation time, the barrier no longer changes significantly, resulting in the measured I - V curves and current values almost unchanged. During the testing process of the I - V curves, the main transport mechanism of charge carriers is tunneling, due to the thickness of our insulation layer being 1.82 nm.

The function of the electric pulse is to facilitate the migration of vacancies, resulting in alteration of the tunneling junction barrier. During the electric pulse stimulation process, the migration of vacancies and carrier tunneling occur simultaneously, and the electron shielding by oxygen vacancies is indeed the cause of the band structure change. The non-uniform electron concentration leads to band restructuring. After the application of the electric pulse to the device, we performed I - V characteristic testing, where carrier tunneling was confirmed as the primary carrier transport mechanism.

Figure R27. Mechanism diagram of electric field regulating oxygen vacancies. a, Migration of oxygen vacancies in ZnO promoted by electric field. **b,** Regulation of energy band of tunneling junction by electric pulse voltage.

(3) Or the HfO₂ layer would have some defects that could play an important role in trapping electrons from the ZnO nanowire. Please give more discussion about that.

We highly appreciate you for this reminder. Defects in HfO₂ can introduce defective surface, which may make a big influence on the tunneling junctions. In order to verify the influence of possible defective surfaces, we characterized the surface of ZnO and conducted capacitance measurements on Ag/HfO₂/n-ZnO tunneling junctions to assess the interface traps formed in the tunneling junction, thereby demonstrating the impact of defects in the HfO₂ layer.

The ZnO nano/microwires, synthesized using the chemical vapor deposition (CVD) method³¹, were characterized by scanning electron microscopy (SEM), transmission electron microscopy (TEM), and X-ray diffraction spectroscopy (XRD) (**Figure R28**). SEM analysis, as depicted in **Figure R28a**, revealed that the ZnO nano/microwire had a diameter of approximately 8 μm, displaying a regular hexagonal geometry with a clean and flat surface. The TEM image in **Figure R28b** depicted surface lattice images of a single ZnO microwire at various positions with high magnification. The distinct boundary of the high-resolution surface and clear lattice fringe in each inset indicated that the ZnO nano/microwire possessed a smooth surface and good crystallinity. Furthermore, **Figure R28c** presented the X-ray diffraction spectra of the ZnO nano/microwires, demonstrating characteristic peaks corresponding to different crystal planes of ZnO. These results prove the excellent crystalline properties of the prepared ZnO nano/microwires.

Traditionally, capacitance measurements have been used to quickly assess the impact of interface traps on the tunneling properties of metal-insulator-semiconductor (MIS) tunneling junctions²¹. In our study, we conducted *C-V* characterizations on 40 fabricated devices, as illustrated in **Figure R29**. The experimental results displayed two typical *C-V* curves of Ag/HfO₂/n-ZnO tunneling junctions without strain (**Figure R29a**). According to the research and theories of Tamm³², Shockley^{33,34}, and others regarding interface traps³⁵ (also historically referred to as interface states, interface defects, surface states, and so on), it is evident that interface traps significantly influence the *C-V* curve, causing it to elongate in the voltage direction²¹. This is attributed to the

additional charges required to fill the traps, resulting in a greater total charge or applied voltage needed to achieve the same surface potential or band bending. Furthermore, interface traps also impact the total capacitance of the tunneling junction. With a fixed bias, the presence of interface traps reduces the remaining charge available to be placed in the depletion layer, thereby diminishing the band bending or surface potential. As a result, the fundamental characteristics of the C - V curve with interface traps shift in both the voltage and capacitance directions, leading to a wider curve opening. Consequently, the blue curve in **Figure R29a** closely resembles the ideal C - V characteristics of an MIS tunneling junction with few interface traps²¹, while the red curve, elongated in the voltage direction, represents a device with noticeable interface traps. **Figure R29b** presents the characterization results of the 40 newly fabricated devices, revealing that only a small percentage ($\sim 3.85\%$) exhibited obvious interface traps.

The process of atomic layer deposition (ALD) of HfO_2 is a self-limiting growth process, with each cycle depositing only one atomic layer or a few atomic layers. Once the surface is occupied by precursor molecules, no further reactions occur, indicating that the growth of the thin film is highly controllable, thus reducing the generation of defects. In this study, the 1.8 nm-thick HfO_2 film deposited by atomic layer deposition is extremely thin, approaching the thickness of a monolayer or a few atomic layers. For the 1.8 nm-thick HfO_2 film, we have rigorously controlled and optimized all steps of the ALD process and used high-purity precursors, therefore, the grown HfO_2 film should have a lower defect density.

Combining the surface quality characterization of ZnO microwires and the C - V characteristics of $\text{Ag}/\text{HfO}_2/n\text{-ZnO}$ devices in our experiment, we suggest that the influence of interface traps (including interface defects in ZnO and HfO_2) on $\text{Ag}/\text{HfO}_2/n\text{-ZnO}$ can be neglected.

Overall, the thickness of ALD HfO_2 in this work is 1.82 nm. During the process of electric pulse stimulation, the current response is accompanied by transient tunneling currents, and the modulation of the barrier by this tunneling current is temporary. In the I - V characteristic tests conducted after the electric pulse stimulation, the main carrier transport mechanism in the device is tunneling. The influence of interface traps in the

HfO₂ layer on Ag/HfO₂/n-ZnO can be considered negligible. Thank you very much again for this reminder, which helps us to understand the interface regulation and possible key factors more deeply.

Figure R28. Scanning electron microscopy (SEM), transmission electron microscopy (TEM) images, and X-ray diffraction spectra of ZnO microwire. **a, b,** SEM (**a**) and TEM (**b**) images of ZnO microwire used in this work. The insets in (**b**) exhibit the surface quality of the ZnO microwire at different locations labeled in black box. The scale bars of insets represent 2 nm. **c,** Corresponding peaks of (100), (002), (101), (102), (110), (112) and (201) planes indicate that the ZnO microwires were highly crystalline in nature.

Figure R29. Influence of interface traps on C - V curves of Ag/HfO₂/n-ZnO tunneling junctions. **a,** Two typical C - V curves of Ag/HfO₂/n-ZnO devices under strain

free condition. The red curve is close to the ideal C - V characteristics of Ag/HfO₂/ n -ZnO tunneling junction. The blue curve is stretched out in the voltage direction, which represent the device with obvious interface traps. **b**, Percentages of 40 devices with few interface traps and obvious interface traps occurred in experiments.

Question #6

Will the recovery time of the interface barrier change after heating? The authors need to give more explanation about the working principle. Additionally, please comment on the difference of interfacial barrier height tuned by the electric-pulse or TENG, which were previously reported.

Response:

We thank you very much for this constructive comment. Based on your suggestion, we conducted relevant experiments to study the effect of annealing on the modulation of the interface barrier height by electrical pulses. We also provided further elucidation on the working principles in this study. Additionally, we discussed the interface modulation of the barrier by TENG and electric pulses. In accordance with these, we will respond to the two questions as follows (1) and (2), respectively.

(1) Will the recovery time of the interface barrier change after annealing? The authors need to give more explanation about the working principle.

We greatly appreciate your insightful comments. Our experimental data clearly demonstrate the significant influence of annealing on the recovery dynamics of the interface barrier. Initially, we subjected ZnO nano/microwires to distinct temperature anneals at 200 °C, 300 °C, 400 °C, and 450 °C. These treated nano/microwires were then used to construct ZnO nano/microwire devices based on Schottky contact principles. As shown in **Figure R30a**, the devices were subjected to electrical pulse stimulation characterized by an amplitude of 80V and a frequency of 10 Hz. **Figures R30b-f** depict the I - V characteristics of these devices, both before and after a 60-second stimulation of periodic electrical pulses. The resulting I - V curves unequivocally indicate that such electrical pulse stimulation significantly enhances the devices' current

response. **Figure R30g** shows the curve representing the relationship between the current value at a 1 V positive bias and the annealing temperature, while **Figure R30h** outlines the changes in the current ratio before and after electrical stimulation as a function of the annealing temperature. Notably, this relationship curve reaches a peak at 400 °C, implying it a critical turning point, beyond which (at 450 °C) the current ratio significantly diminishes. **Figure R30i** captures the data on the recovery time of the device's current after electrical stimulation. These observations confirm that electric pulse stimulation significantly extends the period required for the device to return to its baseline current state.

During the fabrication of ZnO nano/microwires, oxygen vacancies in various valence states (Vo^{2+} , Vo^+ , and Vo^0) are inevitably present ²². At the junction of metal and zinc oxide, neutral oxygen vacancies (Vo^0) readily convert into positively charged ionized oxygen vacancies (Vo^{2+}) ^{23, 24}, thereby impacting the formation of the Schottky barrier. This research exemplifies that the continuous electric pulses can propel the positively charged oxygen vacancies to migrate towards the contact interface of the Ag/HfO₂/n-ZnO tunneling junction ³⁶, creating a region of positive charge, changing the band structure, and decreasing the barrier height ²³. Upon the removal of the pulsed voltage, the dense concentration of oxygen vacancies at the interface slowly diffuses into the ZnO interior until it reverts to its original state. This migration of oxygen vacancies to the interface under an electric field is well understood as an electromigration phenomenon, capable of altering the material's electrical properties. This process can be managed through the regulation of the annealing temperature.

In the following discussion, we will leverage the band diagram to delve into the detailed physical processes of the aggregation and restoration of ionized oxygen vacancies at the interface of the tunneling junction after stimulation by electric pulse voltage. We will also explore the implications of annealing on this process and the restoration of the interface barrier. In the starting state, the contact among the metal, insulator, and piezoelectric semiconductor forms a tunneling junction with a specific barrier height and width, with a low concentration of ionized oxygen vacancies near the interface (as illustrated in **Figure R31a** and **R31a-1**). After the device is stimulated by

the electric pulse for a certain period, driven by the electric field, the positively charged ionized oxygen vacancies will aggregate at the interface, increasing the concentration of ionized oxygen vacancies at the interface (**Figure R31b**), resulting in a decrease in the barrier height ($\Delta\phi_{pulse} = BH0 - BH1$) and width ($\Delta d_{pulse} = BW0 - BW1$) of the tunneling junction (**Figure R31b-1**). Upon removal of the pulsed voltage, due to the uneven distribution of oxygen caused by the external electric field, the high concentration of oxygen vacancies at the interface begins to diffuse into the interior of ZnO, and the barrier will gradually recover²⁶ (**Figure R31c** and **R31c-1**).

The annealing temperature can affect the oxygen vacancies' concentration in semiconductor materials. For oxide semiconductors, at lower annealing temperatures, oxygen vacancies may attach to lattice defects, leading to higher oxygen vacancy concentrations²⁷(as depicted in **Figure R31d**). However, as the annealing temperature increases, oxygen vacancies are more likely to detach from lattice defects or merge with other defects to form more stable configurations, effectively lowering the concentration of oxygen vacancies. Compared to **Figure R31a** and **R31a-1**, the higher concentration of oxygen vacancies in **Figure R31d** has a greater impact on the interface. Therefore, under the same conditions, $BH0$ will be greater than $BH0''$. Under the influence of an electric field, more positively ionized oxygen vacancies (Vo^{2+}) accumulate at the tunneling junction interface (**Figure R31e**), significantly reducing the interface barrier (**Figure R31e-1**). Due to the increased concentration of oxygen vacancies, thus, $BH1$ will be greater than $BH1''$. After annealing, there will be a decrease in the barrier height ($\Delta\phi_{pulse1} = BH0'' - BH1''$) and width ($\Delta d_{pulse1} = BW0'' - BW1''$) of the tunneling junction (**Figure R31e-1**). Upon removal of the pulse voltage, the oxygen vacancies gradually recover under the drive of the concentration gradient (**Figure R31f**), and simultaneously, the barrier is restored (**Figure R31f-1**). This explains the relationship between the recovery time of the current in **Figure R30i** and the annealing temperature: when the annealing temperature rises to 400 °C, the recovery time of the current increases, while when the temperature rises to 450 °C, the recovery time of the current decreases. This further suggests that an increase in the annealing temperature to 400 °C results in an increased concentration of oxygen vacancies. Conversely, when

the annealing temperature rises to 450 °C, the concentration of oxygen vacancies decreases. The device's electrical transport characteristics are intimately bound to the interface barrier, thus, the period required for interface barrier recovery correlates with the concentration of oxygen vacancies. During the recovery of the interface barrier, lower annealing temperatures will result in longer recovery times. However, as the annealing temperature ascends to a certain value, the recovery time will reduce.

Figure R30. Carrier transport characteristics before and after electrical stimulation after annealing. **a**, Electric pulse characterized by an 80 V amplitude and a 10 Hz frequency. **b-f**, I - V characteristics before and after electrical stimulation under different annealing temperature. **g**, The relationship between the current value at a 1 V positive bias and the annealing temperature. **h**, The variations in current ratio before and after electrical stimulation as a function of annealing temperature. **i**, The recovery time of the device's current after electrical stimulation.

Figure R31. Polarization model and band diagram of SBH variation with the influence of annealing. **a, d,** Model of an MIS contact consisted of Ag, HfO₂ and *n*-type ZnO with tunneling junction contact before (**a**) and after (**d**) annealing. The concentration of oxygen vacancies in (**d**) has increased due to the influence of annealing treatment. **a-1, d-1,** Band diagram with the barrier high of BH0 (**a-1**) and BH0''(**d-1**) before and after annealing, respectively. **b, e,** The electric pulse drives the migration and accumulation of positively charged oxygen vacancies at the interface of the Ag/HfO₂/*n*-ZnO tunneling junction before (**b**) and after (**e**) annealing, respectively. **b-1, e-1,** Positively charged oxygen vacancies (Vo²⁺) accumulated at interface result in the bend of energy band and SBH decrease. Under the same conditions, a higher concentration of oxygen vacancies in (**e-1**) leads to more significant modulation of the

interface barrier. **c, f**, Ionized oxygen vacancies diffuse away from Ag/HfO₂/n-ZnO interface slowly after withdrawing the treatment of electrical stimulation before (**c**) and after (**f**) annealing. **c-1, f-1**, Density of positively charged oxygen vacancies (Vo²⁺) at the interface decrease, which results in the barrier to recover. During the recovery of the interface barrier, lower annealing temperatures will result in longer recovery times. However, as the annealing temperature ascends to a certain value, the recovery time will reduce.

(2) Additionally, please comment on the difference of interfacial barrier height tuned by the electric-pulse or TENG, which were previously reported.

We are grateful for your suggestion. Indeed, our methodology is inspired by precedent studies focused on the effective modulation of the Schottky barrier height (SBH) utilizing the triboelectric nanogenerator (TENG) ²²⁻²⁴. It is established that both TENG and electrical pulses serve to modulate the interface barrier, leveraging an electric field to precipitate oxygen vacancy accumulation at the interface, thus influencing the barrier height ²²⁻²⁴. As delineated in **Figure R32a-1**, the friction-based electric signals generated by TENG are converted into electrical pulses capable of stimulating the device, a process visually represented. The electrical pulse signals generated by an external source are shown in **Figure R32b-1**. Subsequent **Figure R32a-2** and **R32b-2** illustrate how the current values within the tested *I-V* curves incrementally rise with prolonged electrical pulse stimulation.

Nevertheless, the attributes of pulses derived from these two methodologies diverge significantly, influencing their efficacy in barrier height modulation. The approach of harnessing TENG for this purpose capitalizes on its synergy with the piezoelectric effect to finesse SBH modulation. However, the generation of pulse signals via TENG presents challenges. The TENG's frictional electricity is particularly prone to environmental disturbances due to electrostatic induction, leading to inconsistent peak outputs ³⁷⁻³⁹. This instability hinders precise investigations into the effects of steady pulse amplitude on oxygen vacancies and their subsequent impact on the interface barrier. Moreover, the transient nature and limited pulse width of frictional

electric signals confine the scope for controllable pulse width modulation. Coupled with a low adjustable frequency range, the current incapacity for high-frequency dynamic stimulation curtails the potential for achieving desired frictional electricity outputs at higher frequencies.

Conversely, utilizing an external source to generate pulse signals offers enhanced stability, including amplitude consistency and temporal continuity. This approach facilitates precise control over amplitude, frequency, pulse width, amplitude voltage duration, and pulse period duration, enabling both transient and sustained pulse delivery. This versatility is instrumental in examining the dynamics of electrical pulse stimulation—including the effects of time, frequency, and amplitude—on the modulation of the interface barrier, which is critically important for us to find the typical modulation of piezotronics modulation. And the finding of obvious modulation and the effective technical way for this modulation constitute the important progress.

We greatly appreciate your valuable insights and important suggestions.

Interface barrier tuned by TENG

Interface barrier tuned by electric pulse

Figure R32. TENG and electrical pulses serve to modulate the interface barrier. a-1, b-1, The friction-based electric signals generated by TENG and the electrical pulse signals generated by an external source capable of stimulating the device. **a-2, b-2,** The current values within the tested $I-V$ curves incrementally rise with prolonged electrical pulse stimulation.

Question #7

How to achieve a long retention of the tunable barrier height? Could you comment on that? Because the well-controlled interfacial barrier height would be beneficial for the optimization of electronic devices for practical applications.

Response:

We greatly appreciate your insightful comment. The distinctiveness of our work resides in our ability to modulate interface barrier height via electric pulse stimulation. Indeed, your point regarding the achievement of prolonged maintenance of this variable barrier

height certainly merits thoughtful deliberation and exploration: The well-controlled interfacial barrier height would be beneficial for the optimization of electronic devices for practical applications. Considering our analysis and relevant literature research, we have suggested three potential approaches to securing a durable retention of the tunable barrier height.

(1) Appropriately increasing the amplitude and duration of electric pulse stimulation

By increasing the amplitude and duration of electric pulses, sufficient energy can be introduced into the device, leading to changes in the distribution of carriers and defects within the semiconductor, thereby affecting the height of the barrier⁴⁰⁻⁴². The long-term maintenance capability of this method can be attributed to several factors. Firstly, increasing the amplitude and duration of electric pulse stimulation can alter the carrier concentration and distribution, as well as the defect distribution within the semiconductor device, thereby adjusting the barrier height. This adjustment can be maintained for a certain period, thus achieving long-term maintenance capability. Secondly, this method can stabilize the barrier height within a certain range, meeting the specific requirements of certain applications. However, it is important to note that increasing the amplitude and duration of electric pulse stimulation may also have some negative effects, such as increased power consumption, thermal effects, and impacts on the long-term stability of the device. Therefore, in practical applications, material characteristics, power consumption, thermal effects, and device durability need to be comprehensively considered to ensure the realization of long-term maintenance capability.

(2) Proper annealing of semiconductor materials

Proper annealing processes can adjust the internal structure and properties of the material, enabling the stable and long-term adjustability of the barrier height within the device. Here are some possible reasons⁴³⁻⁴⁵: ① Crystal defect repair: During annealing, high temperatures can promote the migration and binding of lattice defects, thereby

reducing or repairing crystal defects within the material. This helps improve the crystalline quality and grain boundary conditions, stabilizing the adjustability of the barrier height. ② Carrier redistribution: Proper annealing temperatures can induce carrier redistribution within the material, thereby adjusting the barrier height. This redistribution can be maintained for a certain period, achieving long-term stability. ③ Barrier interface adjustment: Through annealing processes, the interface properties between semiconductor materials and metals or other semiconductor materials can be adjusted, affecting the formation and height of the barrier. This contributes to the long-term stability of the barrier height. ④ Lattice stress release: Proper annealing temperatures can help release lattice stress within the material, improving the crystal structure and stability, thus affecting the long-term maintenance capability of the barrier height.

In summary, proper annealing can improve the internal structure and properties of semiconductor materials, thereby achieving long-term maintenance capability of the barrier height. However, it is important to note that the temperature and duration of annealing need to be precisely controlled to avoid negative impacts on material performance and device stability.

(3) Placing the device in an electric field environment

Existing literature has reported that electric fields can influence the distribution of carriers and potential energy within semiconductor materials, thereby adjusting the barrier height⁴⁶⁻⁴⁸. Here are some possible reasons: ① Carrier distribution regulation: Electric fields can influence the distribution of carriers within semiconductor materials, affecting the formation and height of the barrier. By adjusting the strength and direction of the electric field, carrier distribution within the material can be regulated, thereby affecting the adjustability of the barrier height. ② Potential energy adjustment: Electric fields can alter the potential energy distribution within semiconductor materials, affecting the formation and height of the barrier. By adjusting the strength and direction of the electric field, the potential energy distribution within the material can be changed, thus achieving long-term maintenance capability of the barrier height. ③ Barrier

stability: In an electric field environment, an appropriate electric field can stabilize the barrier height within the device within a certain range, meeting the specific requirements of certain applications. ④ Energy input: Electric fields can provide sufficient energy to the device, allowing for continuous adjustment of the barrier height over a long period. Through the action of the electric field, long-term adjustment of the barrier height can be achieved, meeting the specific requirements of certain applications.

In conclusion, by placing the device in an electric field environment, the distribution of carriers and potential energy within semiconductor materials can be altered, thus achieving long-term maintenance capability of the barrier height. However, it is important to note that the strength and direction of the electric field need to be precisely controlled to ensure long-term and stable adjustment of the barrier height.

We once again express our grateful to you. It guides us to further consider how to utilize electric pulse polarization to achieve long-term adjustability of interface barrier height, which can improve the importance of electric pulse-tuned piezotronic effect.

Reference:

1. Wang L, Wang ZL, Advances in piezotronic transistors and piezotronics. *Nano Today* **37**, 101108 (2021).
2. Wang L, *et al.* Flexoelectronics of centrosymmetric semiconductors. *Nature Nanotechnology* **15**, 661-667 (2020).
3. Wen Z, Li C, Wu D, *et al.* Ferroelectric-field-effect-enhanced electroresistance in metal/ferroelectric/semiconductor tunnel junctions. *Nature Materials* **12**, 617-621 (2013).
4. Cheng G, *et al.* Large anelasticity and associated energy dissipation in single-crystalline nanowires. *Nature Nanotechnology* **10**, 687-691 (2015).
5. Das S, *et al.* Controlled manipulation of oxygen vacancies using nanoscale flexoelectricity. *Nature Communications* **8**, 615 (2017).
6. Huang B, *et al.* Schottky barrier control of self-polarization for a colossal ferroelectric resistive switching. *ACS Nano* **17**, 12347-12357 (2023).
7. Shi J, Starr MB, Wang X. Band structure engineering at heterojunction interfaces

- via the piezotronic effect. *Advanced Materials* **24**, 4683-4691 (2012).
8. Wang ZL, Wu W. Piezotronics and piezo-phototronics: fundamentals and applications. *National Science Review* **1**, 62-90 (2014).
 9. Wu W, Wang ZL. Piezotronics and piezo-phototronics for adaptive electronics and optoelectronics. *Nature Reviews Materials* **1**, 1-17 (2016).
 10. Pan C, Dong L, Zhu G, *et al.* High-resolution electroluminescent imaging of pressure distribution using a piezoelectric nanowire LED array. *Nature Photonics* **7**, 752-758 (2013).
 11. Yu R, Pan C, Wang ZL. High performance of ZnO nanowire protein sensors enhanced by the piezotronic effect. *Energy & Environmental Science* **6**, 494-499 (2013).
 12. Cao X, Cao X, Guo H, *et al.* Piezotronic effect enhanced label-free detection of DNA using a Schottky-contacted ZnO nanowire biosensor. *ACS Nano* **10**, 8038-8044 (2016).
 13. Bai S, Wu W, Qin Y, *et al.* High-performance integrated ZnO nanowire UV sensors on rigid and flexible substrates. *Advanced Functional Materials* **21**, 4464-4469 (2011).
 14. Yang X, Dong L, Shan C, *et al.* Piezophototronic-effect-enhanced electrically pumped lasing. *Advanced Materials* **29**, 1602832 (2017).
 15. Wu W, Wei Y, Wang ZL. Strain-gated piezotronic logic nanodevices. *Advanced materials* **22**, 4711-4715 (2010).
 16. Yu R, Wu W, Ding Y, *et al.* GaN nanobelt-based strain-gated piezotronic logic devices and computation. *ACS Nano* **7**, 6403-6409 (2013).
 17. Dan M, Hu G, Li L, *et al.* High performance piezotronic logic nanodevices based on GaN/InN/GaN topological insulator. *Nano Energy* **50**, 544-551 (2018).
 18. Zhang S, Ma B, Zhou X, *et al.* Strain-controlled power devices as inspired by human reflex. *Nature Communications* **11**, 1-9 (2020).
 19. Wu W, Wen X, Wang ZL. Taxel-addressable matrix of vertical-nanowire piezotronic transistors for active and adaptive tactile imaging. *Science* **340**, 952-957 (2013).

20. Meng J, Li Z. Schottky-contacted nanowire sensors. *Advanced Materials* **32**, 2000130 (2020).
21. S.M. Sze KKN. *Physics of Semiconductor Devices*, 3rd edn. John Wiley & Sons (2006).
22. Meng J, *et al.* Triboelectric nanogenerator enhanced Schottky nanowire sensor for highly sensitive ethanol detection. *Nano Letters* **20**, 4968-4974 (2020).
23. Zhao L, *et al.* Combining triboelectric nanogenerator with piezoelectric effect for optimizing Schottky barrier height modulation. *Science Bulletin* **66**, 1409-1418 (2021).
24. Li H, *et al.* Triboelectric-polarization-enhanced high sensitive ZnO UV sensor. *Nano Today* **33**, 100873 (2020).
25. Zhang B, *et al.* Breath-based human-machine interaction system using triboelectric nanogenerator. *Nano Energy* **64**, 103953 (2019).
26. Ding Y, *et al.* Pyroelectric-field driven defects diffusion along c-axis in ZnO nanobelts under high-energy electron beam irradiation. *Journal of Applied Physics* **116**, 3435 (2014).
27. Jiang W, *et al.* Mobility of oxygen vacancy in SrTiO₃ and its implications for oxygen-migration-based resistance switching. *Journal of Applied Physics* **110**, 034509 (2011).
28. Jiang F, Thangavel G, Zhou X, *et al.* Ferroelectric modulation in flexible lead-free perovskite schottky direct-current nanogenerator for capsule-like magnetic suspension sensor. *Advanced Materials* **35**, 2302815 (2023).
29. Dong J, Xu C, Zhu L, *et al.* A high voltage direct current droplet-based electricity generator inspired by thunderbolts. *Nano Energy* **90**, 106567 (2021).
30. Chen JC, Bhave G, Alrashdan F, *et al.* Self-rectifying magnetoelectric metamaterials for remote neural stimulation and motor function restoration. *Nature materials* **23**, 39-146 (2024).
31. Meng L, *et al.* Enhancing the performance of room temperature ZnO microwire gas sensor through a combined technology of surface etching and UV illumination. *Materials Letter* **212**, 296-298 (2018).

32. Tamm I. Uber eine mogliche Art der Elektronenbindung an Kristalloberflächen *Physikalische Zeitschrift Der Sowjetunion* **1**, 733 (1933).
33. Shockley W, Pearson G. Modulation of conductance of thin films of semiconductors by surface charges. *Physical Review* **74**, 232 (1948).
34. Shockley W. On the surface states associated with a periodic potential. *Physical Review* **56**, 317 (1939).
35. Nicollian E, Brews J. MOS Physics and Technology, John Wiley & Sons (1982).
36. Zhang B, *et al.* Breath-based human-machine interaction system using triboelectric nanogenerator. *Nano Energy* **64**, 103953 (2019).
37. Kim WG, Kim DW, Tcho IW, *et al.* Triboelectric nanogenerator: Structure, mechanism, and applications. *Acs Nano* **15**, 258-287 (2021).
38. Zhang SL, Roach DJ, *et al.* Electromagnetic pulse powered by a triboelectric nanogenerator with applications in accurate self-powered sensing and security. *Advanced Materials Technologies* **5**, 2000368 (2020).
39. Lone SA, Lim KC, Kaswan K, *et al.* Recent advancements for improving the performance of triboelectric nanogenerator devices. *Nano Energy* **99**, 107318 (2022).
40. Chaigne S, Sigg DC, Stewart MT, *et al.* Reversible and irreversible effects of electroporation on contractility and calcium homeostasis in isolated cardiac ventricular myocytes. *Circulation: Arrhythmia and Electrophysiology* **15**, e011131 (2022).
41. arquez-Chin C, Popovic MR. Functional electrical stimulation therapy for restoration of motor function after spinal cord injury and stroke: a review. *Biomedical engineering online* **19**, 1-25 (2020).
42. Yin Q, Zhu P, Liu W, *et al.* A conductive bioengineered cardiac patch for myocardial infarction treatment by improving tissue electrical integrity. *Advanced Healthcare Materials* **12**, 2201856 (2023).
43. Won D, Bang J, Choi SH, *et al.* Transparent electronics for wearable electronics application. *Chemical Reviews* **123**, 9982-10078 (2023).

44. Xu X, Guo T, Lanza M, *et al.* Status and prospects of MXene-based nanoelectronic devices. *Matter* **6**, 800-837 (2023).
45. Han X, Ji Y, Yang Y. Ferroelectric photovoltaic materials and devices. *Advanced Functional Materials* **32**, 2109625 (2022).
46. Nguyen CV, Idrees M, Phuc HV, *et al.* Interlayer coupling and electric field controllable Schottky barriers and contact types in graphene/PbI₂ heterostructures. *Physical Review B* **101**, 235419 (2020).
47. Ma X, Bo H, Gong X, *et al.* Tunable Schottky barrier of WSi₂N₄/graphene heterostructure via interface distance and external electric field. *Applied Surface Science* **615**, 156385 (2023).
48. Shaik S, Danovich D, Joy J, *et al.* Electric-field mediated chemistry: uncovering and exploiting the potential of (oriented) electric fields to exert chemical catalysis and reaction control. *Journal of the American Chemical Society* **142**, 12551-12562 (2020).

Reviewer #3 (Remarks to the Author):

In this work, Qin *et al.* reported in depth investigation of modulation of interface engineering using electrical pulse by the piezotronic effect. They prove that the proposed model is valid for the piezotronic device using literature data. Therefore, I recommend its publication, with suggested major modifications, as below:

Response:

We highly appreciate you for acknowledging the importance of our work and offering the detailed and constructive suggestions, which prompts us to reflect on our shortcomings in principle formulation, material selection, and characterization of important structures, and improve the research details better. And we also thank you very much for the time and effort in helping us to significantly improve the quality of our manuscript.

Question #1

The author should show the high-resolution cross-section image of the Ag/HfO₂/ZnO

junction.

Response:

We highly appreciate you for this kindly reminder and constructive suggestion. It is indeed necessary to give the high-resolution cross-section images of the Ag/HfO₂/*n*-ZnO junctions to show their internal structure, which is the base of investigations in this work. In order to better understand the structure of Ag/HfO₂/*n*-ZnO tunneling junction, we will first outline the preparation process of Ag/HfO₂/*n*-ZnO tunneling junctions, and then provide the high-resolution cross-section image.

We initially synthesized the one-dimensional ZnO nano/microwires *via* the chemical vapor deposition (CVD) method as the key piezoelectric semiconductor to fabricate piezotronic tunneling junctions. Then, the ZnO surface was deposited with an HfO₂ insulation layer by the method of atomic layer deposition (ALD). The critical process parameters involve a pressure of approximately 0.34 Torr, a temperature of around 150 °C, a flow rate of 20 sccm for N₂ carrier gas, a cycle time of 0.015 s for H₂O, a cycle time of 0.1 s for TDMAHf, and a purging duration of 10 s. Ultimately, we successfully achieved precise control in the preparation of HfO₂ insulation layer. In the final step of the tunneling junction preparation, we used the RZF-300 thermal evaporation coating system to prepare the Ag electrode at both ends of the ZnO nanowire/microwire.

In order to show the cross-section image of the Ag/HfO₂/*n*-ZnO junction, we firstly employed high-resolution transmission electron microscopy (HTEM) to determine the thickness of the HfO₂ layer. The scanning electron microscopy (SEM) image presented in **Figure R33a-i** illustrates the clean surface and characteristic hexagonal geometry of the uniform ZnO microwire. Analysis of the HTEM image of the HfO₂/*n*-ZnO structure (**Figure R33a-ii**) yielded a thickness measurement of approximately 1.82 nm for the deposited HfO₂ layer on the ZnO surface.

Given the fact that the ZnO microwires employed in our experiments exhibit diameters in the micrometer range, it necessitates the deposition of thick Ag electrodes at both ends of the ZnO microwire in order to establish electrically conductive electrodes. Consequently, this leads to the thickness of the Ag being considerably larger

than that of the HfO₂. To try our best to show the cross-section image of the tunneling junction, we next used a small tool knife to cut the cross section at one end of the ZnO microwires previously deposited by HfO₂ and Ag. **Figure R33b** presents a high-resolution SEM image that offers a cross-section view of the Ag/HfO₂/*n*-ZnO junction. It clearly depicts the regular hexagonal cross-section of the ZnO nanowire and the Ag electrode coating on the surface of ZnO. Additionally, **Figure R33c** displays an enlarged version of the cross-section image of the Ag/HfO₂/*n*-ZnO junction. The HfO₂ insulation layer with a thickness of approximately 1.82 nm, and the ZnO microwire possessing a diameter in the micron level, and the Ag electrode with a thickness much larger than HfO₂, make it difficult to clearly observe the HfO₂ insulation layer inside the ZnO and Ag in the cross-sectional image shown in **Figure R33b and R33c**. Following this, we conducted an EDX elemental mapping analysis based on the high-resolution cross-section image of the Ag/HfO₂/*n*-ZnO junction in **Figure R33d**. **Figure R33e** displays the EDX element mappings for Zn, O, Hf, and Ag, allowing us to observe the distribution of the Hf element at the boundary between Ag and ZnO.

Following your kindly reminder and suggestion, we realized the importance of the cross-section image of the Ag/HfO₂/*n*-ZnO junction, which is absolutely the key fundament for further investigation on the pulse-tuned piezotronic tunneling junction. So, we added the relevant content into the revised Supplementary Information with red marks.

Figure R33. Cross-section TEM and SEM images of Ag/HfO₂/n-ZnO junctions and the corresponding EDX element mapping image. a-i, SEM image of the ZnO microwire with a diameter of about 8 μm, showing a typical hexagonal geometry and a clean surface. a-ii, High-resolution TEM image of HfO₂ layer with a thickness of about 1.82 nm deposited on ZnO surface. b, Cross-section view SEM images of an Ag/HfO₂/n-ZnO junction. c, Enlarged version of the cross-section SEM image of the Ag/HfO₂/n-ZnO junction in (b). d, High-resolution cross-section view SEM image of the Ag/HfO₂/n-ZnO junction. e, The corresponding EDX element mapping image of Zn, Ag, O and Hf in (d).

Question #2

There are many insulator layers, why the author used HfO₂ insulator layer for device

fabrication. What is the thickness of the HfO₂ layer.

Response:

We thank you very much for these nice and valuable suggestions. Prior to our response, we are sorry that we did not give the thickness value of HfO₂ layer in the manuscript. To better answer these two questions, we will briefly discuss the reason why choose HfO₂ as insulator layer, and then give the thickness (~1.82 nm) of the HfO₂ layer and the reason for choosing its thickness value of 1.82 nm for device fabrication. With your kindly reminder and suggestion, we also think the parameter of HfO₂ thickness and the related discussion important for improving the quality and readability of our work, and have added the relevant content in the revised manuscript and added the following discussion in the Supplementary Information with changes marked in red.

(1) There are many insulator layers, why the author used HfO₂ insulator layer for device fabrication.

Thank you very much for this nice comment. The reasons for choosing HfO₂ insulator layer for device fabrication are really what we should provide. Here, we combine the related literature reports and our further experimental results of studying the influence of insulation materials on the performance of piezotronic tunneling junction to clarify the reason why we choose HfO₂.

At the beginning, we chose HfO₂ as the insulator layer due to its advantages in terms of high dielectric constants (k) and wide band gap (E_g). Many literatures report that dielectric insulation layer with high k values and wide band gaps can not only reduce the leakage current and the overall power consumption during device operation, but also has good stability to withstand breakdown. As compared with SiO₂, Al₂O₃, MgO and some other insulation materials, HfO₂ has a good advantage in terms of k value and band gap, as shown in **Table R2**¹⁻⁵. Based on our previous survey, we chose HfO₂ as the dielectric insulator layer of the piezotronic tunneling junction.

In order to experimentally clarify the influence of insulation materials on piezotronic modification of carrier transport in piezotronic tunneling junctions, we also tested and characterized the strain sensing performance of piezotronic tunneling devices

using HfO₂, ZrO₂, Al₂O₃ and SiO₂ as insulator layers with a same thickness of ~1.8 nm under the conditions of tensile strain of 0.00%, 0.033%, 0.067% and 0.10%, respectively. As shown in **Figure R34**, the current values measured at +3.0 V bias for four kinds of piezotronic tunneling devices were selected. The number of devices used for each type is 20. It can be clearly seen from **Figure R34** that, when the strain is 0.00%, the current values of the four types of piezotronic tunneling devices are randomly distributed between 0.01 and 7 nA, and when the strain is 0.10%, their current values are also irregularly distributed between 30 and 250 nA. According to the current data in **Figure R34**, the current on/off ratio and change of Schottky barrier height of four types of piezotronic tunneling devices under 0.10% strain were calculated, counted, and compared in **Figure R35**. The comparison results show that changing the dielectric insulation materials such as HfO₂, ZrO₂, Al₂O₃ and SiO₂ has almost no influence on the strain sensing performance of piezotronic tunneling devices. The classical carrier transport of metal-insulator-semiconductor (MIS) tunneling junction can be expressed as ⁶:

$$I = SA^*T^2 \exp(-\alpha_T d \sqrt{q\varphi_T}) \exp\left(-\frac{q\varphi_B}{kT}\right) \left[\exp\left(\frac{qV}{nkT}\right) - 1 \right]$$

where $\exp(-\alpha_T d \sqrt{q\varphi_T})$ is the tunneling probability term, S is the contact area of the tunneling barrier, A^* is the effective Richardson constant, q represents the charge of an electron, k is the Boltzmann constant, T is the absolute temperature, φ_B represents the height of the Schottky barrier, n is the ideality factor, d is the thickness of the insulation layer in the tunneling junction, $\alpha_T = 2\sqrt{2qm^*}/\hbar$ (m^* represents the effective mass of the charge carriers, \hbar is the reduced Planck constant), φ_T represents the effective height of the tunneling barrier. It can be seen that the tunneling current is mainly dependent on the thickness of insulator layer (d) and the Schottky barrier height (φ_B) (which is determined by the metal's work function and the semiconductor's electron affinity), and slightly dependent on the band gap of insulator layer (which together with its electron affinity will affect the effective height of the tunneling barrier $q\varphi_T$), and almost not dependent on the k value. Given that the band gaps of HfO₂, ZrO₂, Al₂O₃ and SiO₂ are not very different, the piezotronic modification of tunneling current has

little to do with the choice of dielectric insulation layer, which is consistent with our experimental results in **Figures R34** and **R35**.

Based on above reasons, we chose HfO₂ as the insulator layer for convenience, which can be precisely fabricated by atomic layer deposition (ALD). Actually, other dielectric materials including ZrO₂, Al₂O₃ and SiO₂ can also be candidates as insulation layer. We added above discussions into the revised Supplementary Information marked in red.

Table R2. Dielectric constants (k) and band gaps (E_g) of several typical insulation materials¹⁻⁵.

Insulation materials	Dielectric constants k	Band gaps E_g (eV)
HfO ₂	20~25	5.6~6.0
ZrO ₂	17~25	5.1~7.8
SiO ₂	3.9	9
Si ₃ N ₄	7~7.5	5~5.3
Al ₂ O ₃	9	8.8
MgO	8.4~9.8	8.7~8.8

Figure R34. Influence of insulation materials on piezotronic modification of the carrier transport of piezotronic tunneling devices. a-d, Piezotronic modification of the carrier transports of four types of piezotronic tunneling devices using HfO₂ (a), ZrO₂ (b), Al₂O₃ (c) and SiO₂ (d) as insulation layers under the conditions of tensile strain of 0.00%, 0.033%, 0.067% and 0.10%, respectively. The above currents were measured at bias of +3.0 V.

Figure R35. Sensing performance of piezotronic tunneling devices. a, b, On/off ratio (a) and change of effective Schottky barrier height (b) of the devices based on

piezotronic tunneling junction with different insulation materials of HfO₂, ZrO₂, Al₂O₃ and SiO₂. The error bars denote standard deviations of the mean.

(2) What is the thickness of the HfO₂ layer.

We appreciate you so much for this kindly reminder. In our experiments, an HfO₂ insulator layer with a thickness of about 1.8 nm was precisely deposited on ZnO microwire by atomic layer deposition (ALD). In addition, we used high-resolution transmission electron microscopy (HTEM) to characterize the thickness of HfO₂ layer. The scanning electron microscopy (SEM) image (**Figure R36a-i**) shows the uniform ZnO microwire with a typical hexagonal geometry, and HTEM image of HfO₂/*n*-ZnO (**Figure R36a-ii**) shows the thickness value of the deposited HfO₂ on ZnO surface, which is about 1.82 nm. In **Figure R36b**, the distribution of HfO₂ thickness along the *x*-axis labeled in **Figure R36a-i** demonstrates that the HfO₂ is uniform with a thickness of about 1.8 (± 0.3) nm. The inset shows the statistical distribution of the insulation thickness.

Here, we will explain why the thickness of 1.8 nm was chosen. In order to demonstrate the influence of insulator layer thickness on the sensing performance of piezotronic tunneling devices, Ag/HfO₂/*n*-ZnO devices with HfO₂ thickness of 0.0 nm, 0.4 nm, 1.1 nm, 1.8 nm, 2.5 nm, 3.6 nm, and 7.3 nm were prepared, respectively. The piezotronic modifications of electrical transport of 140 devices (20 devices of each thickness) under 0.10% tensile strain were studied. The experimental results show that the device with 1.8 nm HfO₂ exhibit the highest current on/off ratio, which makes us choose 1.8 nm HfO₂ as the insulator layer of piezotronic tunneling junctions.

Figure R37 shows the statistical distributions of on-state and off-state currents of 140 Ag/HfO₂/*n*-ZnO devices (20 devices of each thickness) with 0.0 nm, 0.4 nm, 1.1 nm, 1.8 nm, 2.5 nm, 3.6 nm, and 7.3 nm HfO₂. By summarizing and analyzing the test data, we can obtain the average on-state current (**Figure R37a** and **R37b**) and off-state current (**Figure R37c** and **R37d**) of these devices with different HfO₂ thicknesses. It can be seen from the figure that with the decrease of HfO₂ thickness, both the on-state current and off-state current increase. We further calculate the statistical distribution of

the current on/off ratio of these devices, as shown in **Figure R38a**. It can be found that the maximum current on/off ratio occurs at the insulation layer thickness of about 1.8 nm (**Figure R38b**).

Therefore, based on the above considerations, we chose HfO₂ with the thickness of about 1.8 nm as the insulator layer to fabricate the piezotronic tunneling junction in this work. We have added the thickness value of HfO₂ into the revised manuscript. Thank you very much again for your kindly reminder.

Figure R36. Morphology of HfO₂/n-ZnO microwire and thickness of deposited HfO₂ layer. a-i, SEM image of the ZnO microwire with a diameter of about 8 μm, showing a typical hexagonal geometry and a clean surface. **a-ii**, High-resolution TEM image of HfO₂ layer deposited on ZnO surface, showing a thickness of 1.82 nm. **b**, Thickness of HfO₂ along the *x*-axis labeled in (a-i), showing the HfO₂ thickness of about 1.8 (± 0.3) nm. The inset shows the statistical distribution of the HfO₂ thickness.

Figure R37. On-state and Off-state currents of Ag/HfO₂/n-ZnO devices. Statistical distributions of On-state currents (a-b) and Off-state currents (c-d) of 140 Ag/HfO₂/n-ZnO devices (20 devices for each thickness) with 0 nm, 0.4 nm, 1.1 nm, 1.8 nm, 2.5 nm, 3.6 nm, and 7.3 nm thick HfO₂. The error bars denote standard deviations of the mean.

Figure R38. Current on/off ratio of Ag/HfO₂/n-ZnO devices. a, Statistical distribution of current on/off ratios of Ag/HfO₂/n-ZnO devices with 0.0 nm, 0.4 nm, 1.1

nm, 1.8 nm, 2.5 nm, 3.6 nm, and 7.3 nm thick HfO₂. **b**, The current on/off ratio as a function of HfO₂ thickness. The error bars denote standard deviations of the mean.

Question #3

Why author measured the I - V characteristics (Fig. 2e) at low voltages. Did author measure higher (>1 V) voltages or not. If yes, please include it in the manuscript. If not, please explain the reason.

Response:

We highly appreciate you for this constructive and in-depth suggestion. In piezotronics, because high voltage will lead to degraded piezotronic modification, I - V characteristics of piezotronic devices are typically measured using low (sweep) voltage. Here, we will provide a detailed explanation for measuring the I - V characteristics at a voltage of 1 V and not exceeding this value.

(1) Why author measured the I - V characteristics (Fig. 2e) at low voltages.

Thank you very much for this nice suggestion. The objective of performing I - V measurements is to evaluate the impact of piezotronic effect on the electrical transport before and after applying electrical stimulation to the Ag/HfO₂/ n -ZnO devices. The I - V characteristics acquired at low voltage levels fulfill the research requirements necessary for this study. While, high voltage will lead to a poor piezotronic modification on the electrical transport of devices, and even induce damage or fatigue at the interface (Ag/HfO₂/ n -ZnO) of devices.

Figure R39 illustrates the comparison of I - V characteristics at different sweep voltages (± 1 V, ± 3 V, ± 5 V, ± 7 V and ± 9 V) for the Ag/HfO₂/ n -ZnO device. It clearly indicates an increasing trend in current values within the I - V curves for compressive strains of 0.00% and 0.10% as the sweep voltage increases. This phenomenon can be attributed to the increased electric field leading to changes in the device's interface band structure, influencing electrons' energy levels and carrier behavior, ultimately resulting in higher current flow. Concurrently, the device's piezoelectric polarization control performance experiences a substantial decline as the sweep voltage increases. A clear

illustration of this is seen in **Figure R39e**, where the I - V curves for compressive strains of 0.00% and 0.10% demonstrate significant overlap under a ± 9 V sweep voltage. Furthermore, **Figure R39f** provides a comprehensive summary of the correlation between the sweep voltage and the current on-off ratio, unequivocally demonstrating a noteworthy decrease as the sweep voltage increases. This behavior arises due to the enhanced shielding capacity of free carriers against piezoelectric polarization charges under high electric fields, leading to a decline in the piezotronic effect.

Figure R40 illustrates the measured I - V characteristics at ± 1 V and ± 5 V sweep voltages, obtained by performing periodic electrical pulse stimulation. The Ag/HfO₂/n-ZnO device was stimulated using electrical pulses of 80 V amplitude and 10 Hz frequency, with stimulation durations of 0 s, 30 s, 60 s and 120 s, respectively. As depicted in **Figure R40a**, the current response within the I - V curve tested under a ± 1 V sweep voltage continues to increase with the extension of the pulse stimulation time. **Figure R40b** reveals the I - V characteristics observed under a ± 5 V sweep voltage when subjected to pulse stimulation times of 0 s, 30 s and 60 s. Notably, during these durations, the device's response current displays an upward trend; however, a pulse stimulation time of 120 s leads to a substantial decrease in the current value within the I - V curve. It is worth mentioning that periodic electrical pulse stimulation and high sweep voltage testing can potentially induce damage or fatigue at the interface, consequently diminishing its performance.

The above results suggest that higher sweep voltages are not ideal for studying the tuning impact of electric pulses on piezotronic modification of tunneling junctions. Consequently, we opted to examine the I - V characteristics at lower voltages.

Figure R39. Comparison of I - V characteristics at different sweep voltage of $\text{Ag}/\text{HfO}_2/\text{n-ZnO}$ device. **a-e**, The I - V curves is measured at different sweep voltage of ± 1 V (**a**), ± 3 V (**b**), ± 5 V (**c**), ± 7 V (**d**) and ± 9 V (**e**), respectively, under the compressive strain of 0.00% and 0.10%. **f**, Relationship between current on/off ratio and sweep voltage.

Figure R40. I - V characteristics under electric pulse stimulations with different duration at sweep voltages of ± 1 V and ± 5 V. **a**, **b**, I - V curves under electric pulse stimulations for 0 s, 30 s, 60 s and 120 s measured at sweep voltages of ± 1 V (**a**) and ± 5 V (**b**), respectively. The electrical pulse used has a frequency of 10 Hz, and the amplitude is set at 80 V.

(2) Did author measure higher (>1 V) voltages or not. If yes, please include it in the manuscript. If not, please explain the reason.

Thank you for this constructive suggestion. Voltages exceeding 1 V were not employed in this study. The findings derived from **Figures R39** and **R40** clearly indicate that high sweep voltages are inappropriate for studying the tuning influence of electric pulses on piezotronic effect at tunneling junctions. Moreover, there are several reasons that support the decision to employ the low-voltage testing method⁷⁻¹¹: Firstly, by avoiding high voltages, potential damage to the interface (Ag/HfO₂/n-ZnO) in the device can be prevented. Excessive voltage can result in material breakdown or degradation, compromising measurement reversibility and reliability. Secondly, operating at high voltages accentuates non-linear effects in the material, complicating the accurate interpretation of I - V characteristics. On the other hand, conducting measurements at low voltages ensures that the device operates within the linear range, facilitating the analysis of its electrical behavior. Lastly, lower voltage measurements conserve energy, enhancing the overall efficiency and sustainability of the testing process.

In summary, measuring I - V characteristics at low voltage levels helps to protect the device, reduce non-linear effects, and conserve energy, thereby providing more accurate and reliable data for the characterization of piezotronic devices. Consequently, considering the analysis of the results depicted in **Figures R39** and **R40**, along with the benefits associated with low voltage testing, we opted to utilize a testing voltage of 1 V in this study, as opposed to selecting a higher voltage.

Question #4

Discussion needs about the fundamental understanding of piezotronics and why the contact the author used is Ag.

Response:

We appreciate you so much for reminding us to discuss the fundamental understanding of piezotronics and the reason of using Ag as electrodes. In order to answer these two questions more clearly, we will discuss the fundamental understanding of piezotronics detailly in the first, and then explain the reason for using Ag as the contact electrode.

Considering these two contents regarding the understanding of piezotronics and fabrication of typical piezotronic devices, we have added the relevant contents to the revised Supplementary Information and marked them in red.

(1) Discussion needs about the fundamental understanding of piezotronics.

Thank you very much for this valuable reminder. Piezotronics is a field that explores the interaction/coupling between piezoelectric polarization and semiconductor properties (**Figure R41**). When a mechanical stimulus is applied to a piezoelectric semiconductor, it can produce piezoelectric polarization at interfaces (formed between the piezoelectric semiconductor and other materials such as metal or another semiconductor) and thus modulate the energy band and carrier transport of interfaces, leading to the development of new types of sensors, and electronic devices. To better clarify what is piezotronics, we will discuss this from the following two aspects:

① *Piezoelectricity in wurtzite semiconductors*

Piezoelectric semiconductor materials with a wurtzite structure, such as ZnO and GaN, possess both piezoelectric and semiconductor properties. The wurtzite structure of these materials allows them to exhibit piezoelectricity due to their non-centrosymmetric crystal structure, while also functioning as semiconductors, enabling the modulation of electrical transport processes under mechanical stimuli. The wurtzite-structured ZnO exhibits piezoelectric characteristics along the *c*-axis when subjected to external force, as depicted in **Figure R42a**, top. In **Figure R42a**, in the absence of strain, the metal cation Zn^{2+} and the anion O^{2-} generally form tetrahedral coordination, and the positive and negative ion centers overlap each other. When the tetrahedral vertices of the unit cell are subjected to external forces, the centers of Zn^{2+} cations and O^{2-} anions are shifted relative to each other, resulting in electric dipole moments and piezoelectric polarization (**Figure R42a**, bottom). In addition, piezoelectric polarization charges will be generated on both surfaces of the ZnO crystal along the *c*-axis, with the thickness of about one to two atomic layers, when subjected to axial stress/strain¹²⁻¹⁵. It is worth to mention that the *c*-axis direction is usually the growth direction of ZnO nanowire/nanosheet. Since all the unit cell in wurtzite-structured ZnO have such an

electric dipole moment under stress/strain, a potential distribution along the c -axis direction, called the piezoelectric potential (**Figure R42b**), is generated on a macroscopic scale, which is the piezoelectric property of ZnO. The stress/strain-induced piezoelectric potential in piezoelectric semiconductors enables the modulation of electrical transport processes across the interface of metal-semiconductor Schottky contacts under mechanical stimuli, representing the fundamental principle of piezotronics.

② *Effect of piezopotential on metal-semiconductor (MS) contact and metal-insulator-semiconductor (MIS) contact*

In the field of piezotronics, the fundamental construction of piezotronic devices typically involves semiconductor-semiconductor contacts (such as p - n junctions) and Schottky junctions based on metal-semiconductor (MS) contacts. Subsequently, we will expound upon the foundational principle of interface regulation in piezotronics, utilizing metal-semiconductor (MS) contact as a paradigmatic illustration. When a metal and a semiconductor come into contact, the free charges near the junction region undergo significant redistribution to establish thermal equilibrium and form either a non-rectifying Ohmic contact or a rectifying Schottky contact. In the case of a Schottky contact, the barrier height and the width of the depletion region of the induced Schottky barrier govern carrier transport across the metal-semiconductor interface when reverse-biased. Upon application of normal stress or strain, the piezoelectric polarization charge induced at the MS contact can effectively regulate the interfacial Schottky barrier and control carrier transport. This exertion of influence on the concentration and distribution of free carriers at the interface is substantial.

Here, we consider the n -type piezoelectric semiconductor as a representative example and, for the sake of simplicity, disregarding surface states, shielding effects, and other crystal anomalies, while maintaining the MS junction at zero bias. **Figure R43** shows the band diagrams of MS and MIS contact regulated by piezoelectric polarization charges. Generally, negative piezoelectric polarization charges repel electrons at the MS interface, leading to further depletion of the depletion region and an increase in the Schottky barrier height (**Figure R43a-i**). Conversely, positive

piezoelectric polarization charges attract electrons to the MS interface, thereby reducing the depletion region width and the Schottky barrier height (**Figure R43a-ii**). The magnitude and polarity of the piezoelectric potential in the piezoelectric semiconductor depend on the crystal orientation and the magnitude and polarity of the applied stress/strain. The above forms the basis for the regulation of the interface energy band by piezoelectric polarization charges, also known as the piezotronic effect, which has been extensively demonstrated in wurtzite-structured semiconductors (such as ZnO, GaN, etc.)¹²⁻¹⁵. The metal-insulator-semiconductor (MIS) contact can be achieved by inserting an insulation layer between the metal and semiconductor. Similarly, the piezoelectric potential induced by the piezoelectric polarization charges also exerts a regulatory effect on the interfacial energy band, thus influencing carrier transport (**Figure R43b**).

In summary, the coupling of piezoelectric properties and the electrical transport properties of piezoelectric semiconductors has given rise to the emerging field of piezotronics (**Figure R41**). It is an interface effect that is grounded in the piezoelectric polarization charges and the corresponding piezoelectric potential caused by external mechanical stimuli, acting as the control signal to regulate the interfaces including MS interface and MIS interface, thus allowing for effective regulation of the electrical transport characteristics of the carriers across these interfaces.

Figure R41. Coupling the piezoelectricity and the semiconducting property resulting in the emerging field of piezotronics.

Figure R42. Piezoelectricity in wurtzite semiconductors. a, Atomic model of the wurtzite-structured ZnO and the origin of piezoelectricity. **b**, Piezopotential distributed along ZnO nanowire under axial forces.

Figure R43. Band diagrams of metal-semiconductor (MS) and metal-insulator-semiconductor (MIS) contact regulated by piezotronic effect. a, b, Band diagrams depicting the contact between MS (a) and MIS (b) regulated by piezoelectric polarization. The presence of stress induces positive and negative piezoelectric polarization charges at the MS and MIS interface, which can decrease and increase the barrier height, respectively.

(2) Why the contact the author used is Ag.

Thank you very much for this valuable suggestion. There are two reasons that motivate us to use Silver (Ag) as electrodes. Firstly, Ag possesses a relative high work function and can form Schottky contacts with ZnO. ZnO is an II-VI group metal-oxide semiconductor with a wurtzite crystal structure that has a direct band gap of 3.37 eV, which has been widely used as the piezoelectric semiconductor in piezotronics. Schottky contacts with ZnO have been fabricated using various metals (Au, Ag, Pt, and Pd) with work functions of 5.1 eV, 4.26 eV, 5.65 eV and 5.12 eV, respectively. In 1965, C.A.M. conducted experiments where vacuum-deposited metal-semiconductor contacts were made on cleaved ZnO single crystals using Cu, In, Al, Ti, Au, Pt, Pd and Ag¹⁶. Experimental findings show that Cu, In, Al and Ti metals have a significantly low barrier height when in contact with ZnO, indicating an ohmic contact configuration. And, Au, Pt, Pd and Ag metals demonstrate high barrier heights when in contact with ZnO, thus establishing a Schottky contact arrangement. Secondly, in the past ten years, Ag has been widely utilized to fabricate Schottky contacts particularly with ZnO in piezotronic devices. Therefore, we selected Ag as electrodes to fabricate Schottky contact with ZnO for preparing piezotronic tunneling devices.

Question #5

In this paper, the PDMS is piezoelectric ceramics materials, the authors should discuss whether the PDMS coating has influence on the performance of the Ag/HfO₂/ZnO junction.

Response:

We thank you very much for this kindly and important reminder, which in fact highly improved our understandings about the influence of PDMS on the piezotronic tunneling devices. In the following paragraphs, we will briefly introduce PDMS, and then investigate the influence of PDMS on the device by calculation.

Polydimethylsiloxane, commonly known as PDMS, is a polymer classified as a silicon elastomer. It is the most widely used silicone based on organic polymers that presents the remarkable proprieties, including flexibility, optical transparency, biocompatibility, and simple fabrication, *etc*¹⁷. PDMS is also chemically inert, thermally stable, permeable to gases, exhibits isotropic and homogeneous properties¹⁸. With all these advantages, PDMS has demonstrated many promising applications in fields such as wearable electronics, microelectromechanical systems (MEMS), medical implants, microfluidics, and many others^{19, 20}. It is worth noting that, due to its electrical insulation capability, PDMS is the most commonly used material to embed or encapsulate electronic components by casting, which can protect the components from environmental factors and mechanical shock, hence extending the lifespan of devices¹⁸. **Table R3** lists some physical properties of PDMS.

Additionally, the manufacturing of PDMS composites mixed with other different types of materials, such as fibers, particles, polymers, or other additives, can modify the mechanical, electrical, optical, thermal, and surface wettability properties of PDMS, which allows the optimization and expansion of its applications in various fields of engineering¹⁷. However, to our best, usually pure PDMS is not a piezoelectric material according to the related studies reported hitherto.

For a traditional piezotronic device (*e.g.*, piezotronic transistor) with a lateral structure, the piezoelectric nano/microwire lies flat on a flexible substrate (such as Polyethyleneterephthalate (PET)), with both ends tightly bonded to the substrate by metal electrodes. In this work, the introduction of the PDMS is mainly to encapsulate the piezotronic tunneling junction and protect the ZnO nano/microwire from being easily damaged when the substrate bends. As the main encapsulation material in the piezotronic device, the properties of PDMS are directly related to the nano/microwire deformation macroscopically. Regarding mechanical properties, pure PDMS usually

shows low mechanical properties, such as low modulus of elasticity and strength, which would vary depending on the curing agent ratio and curing temperature during the manufacturing process^{20,21}. It should be noted that the flexible substrate has much more stiffness (for example Young's modulus of PET, $E = 3\sim 3.5$ GPa) than the thin encapsulation layer (Young's modulus of PDMS, $E = 360\sim 870$ kPa), and thus the mechanical deformation of PET will not be affected by PDMS. It is usually assumed that the ZnO nano/microwire can strictly deform with the bending of PET substrate in a shape close to a circular arc.

To delve into the possible influence of the PDMS structure on the mechanical property and the sensing performance of device, the finite-element method (FEM) simulations are further performed under the same condition as in the experiment. Considering the extremely small diameter of the ZnO nano/microwire (usually in the micron scale, about 8 μm) compared with the thicknesses of the PDMS layer and the PET substrate, the morphology of ZnO nano/microwire in the radial direction can be ignored in this work. For the loading process of mechanical strain, we create the proper boundary condition in the neutral layer of the PET substrate to induce the bending deformation, while the length of neutral layer of the substrate is assumed to remain unchanged in the model. Despite these simplified assumptions, it is reasonable for us to construct a 2D finite element model in plane stress condition to provides a qualitative insight into the role of PDMS structure. As shown in **Figure R44a**, the FEM model is established with a ZnO nano/microwire (1 mm in length) that exists on the central of PET substrate (0.15 mm in thickness and 30 mm in length). Besides, a PDMS layer with thickness 0.5 mm is introduced in a typical model; whereas there is no PDMS structure in the other one for comparison. Here, we list the main parameters of the material properties for the mechanical simulation using the material library database in the software of COMSOL Multiphysics (**Table R4**).

It could be clearly observed that the upper part of the piezotronic device is gradually compressed as the increasing curvature of the substrate, as demonstrated by the displacement distribution profiles under different bending conditions in **Figure R44b** and **R44c**. **Figure R44d** and **R44e** show the corresponding outlines of the bended

nano/microwires obtained from **Figure R44b** and **R44c**, while the calculated strain of the ZnO nano/microwire in the piezotronic device with and without the PDMS layer under the same loading condition is plotted in **Figure R44f**. According to the verification of the simulation results, the existence of PDMS layer would have little influence on the ZnO nano/microwire deformation when the substrate bends, since the nano/microwire strain might be identical for the piezotronic device with and without PDMS layer (**Figure R44f**). Therefore, as expected, the addition of the PDMS layer would not significantly affect the sensing performance of the piezotronic tunneling device.

Following your kindly reminder, we added above relevant discussion into the revised Supplementary Information with changes marked in red.

Table R3. Typical properties of cured PDMS ¹⁸.

Property (Unity)	Value
Transmittance at range 390 nm to 780 nm (%)	75~92
Index of refraction	1.4
Thermal conductivity (W/m·K)	0.2~0.27
Specific heat (kJ/kg·K)	1.46
Dielectric strength (kV/mm)	19
Dielectric constant	2.3~2.8
Volume resistivity (ohm·cm)	2.9×10^{14}
Young's modulus (kPa)	360~870
Poisson ratio	0.5
Tensile strength (MPa)	2.24~6.7
Hardness [Shore A]	41~43
Viscosity (Pa·s)	3.5
Hydrophobicity—contact angle (°)	$\sim 108^\circ \pm 7^\circ$
Melting Point (°C)	-49.9 to -40

Figure R44. Impact of the PDMS encapsulation layer on the performance of the piezotronic tunneling device by FEM simulation. **a**, Schematic illustration of the side view of the piezotronic tunneling device with (top) and without (bottom) PDMS encapsulation layer. **b-e**, Displacement distribution profiles (**b**, **c**) and the corresponding outlines of bended nano/microwires (**d**, **e**) of the piezotronic tunneling devices with (**b**, **d**) and without (**c**, **e**) PDMS layer in response to mechanical strain, respectively. **f**, Calculated ZnO nano/microwire strain as a function of the distance change between the two ends of the substrate in the piezotronic device with and without the PDMS layer, respectively.

Table R4. Material parameters in the finite element method (FEM) simulation.

Materials	Young's modulus, E	Poisson's ratio, ν
PET	4 GPa	0.35
PDMS	750 kPa	0.49
ZnO	210 GPa	0.33

References:

1. Nahar R, Singh V, Sharma A. Study of electrical and microstructure properties of high dielectric hafnium oxide thin film for MOS devices. *Journal of Materials*

- Science: Materials in Electronics* **18**, 615-619 (2007).
2. Gilmer D, *et al.* Compatibility of polycrystalline silicon gate deposition with HfO₂ and Al₂O₃/HfO₂ gate dielectrics. *Applied Physics Letters* **81**, 1288-1290 (2002).
 3. Kang S, *et al.* Effect of deposition conditions of poly Si_{1-x}Ge_x films and Ge atoms on the electrical properties of poly Si_{1-x}Ge_x (x= 0, 0.6)/HfO₂ gate stack. *Journal of Applied Physics* **94**, 4608-4613 (2003).
 4. Hinkle C, Fulton C, *et al.* Enhanced tunneling in stacked gate dielectrics with ultra-thin HfO₂ (ZrO₂) layers sandwiched between thicker SiO₂ layers. *Applied Surface Science* **234**, 240-245 (2004).
 5. Gerritsen E, *et al.* Evolution of materials technology for stacked-capacitors in 65 nm embedded-DRAM. *Solid-State Electronics* **49**, 1767-1775 (2005).
 6. S.M. Sze KKN. *Physics of Semiconductor Devices*, 3rd edn. John Wiley & Sons (2006).
 7. Deshmukh S, Rojo MM, Yalon E, *et al.* Direct measurement of nanoscale filamentary hot spots in resistive memory devices. *Science Advances* **8**, eabk1514 (2022).
 8. Ghittorelli M, Lingstedt L, *et al.* High-sensitivity ion detection at low voltages with current-driven organic electrochemical transistors. *Nature Communications* **9**, 1441 (2018).
 9. Conti S, Pimpolari L, Calabrese G, *et al.* Low-voltage 2D materials-based printed field-effect transistors for integrated digital and analog electronics on paper. *Nature Communications* **11**, 3566 (2020).
 10. Valencia A, Hincapie RA, *et al.* Optimal location, selection, and operation of battery energy storage systems and renewable distributed generation in medium–low voltage distribution networks. *Journal of Energy Storage* **34**, 102158 (2021).
 11. Wu W, Lin B. Benefits of electric vehicles integrating into power grid. *Energy* **224**, 120108 (2021).
 12. Pan C, Zhai J, Wang ZL. Piezotronics and piezo-phototronics of third generation semiconductor nanowires. *Chemical Reviews* **119**, 9303-9359 (2019).
 13. Wang L, Wang ZL. Advances in piezotronic transistors and piezotronics. *Nano*

Today **37**, 101108 (2021).

14. Zhu L, Wang ZL. A perspective on piezotronics and piezo-phototronics based on the third and fourth generation semiconductors. *Applied Physics Letters* **122**, 250501 (2023).
15. An C, Qi H, Wang L, *et al.* Piezotronic and piezo-phototronic effects of atomically-thin ZnO nanosheets. *Nano Energy* **82**, 105653 (2021).
16. Polyakov AY, Smirnov NB, Kozhukhova EA, *et al.* Electrical characteristics of Au and Ag Schottky contacts on *n*-ZnO. *Applied physics letters* **83**, 1575-1577 (2003).
17. Ariati R, Sales F, *et al.* Polydimethylsiloxane composites characterization and its applications: a review. *Polymers* **13**, 4258 (2021).
18. Miranda I, *et al.* Properties and applications of PDMS for biomedical engineering: a review. *Journal of Functional Biomaterials* **13**, 2 (2022).
19. Teixeira I, *et al.* Polydimethylsiloxane mechanical properties: a systematic review. *AIMS Materials Science* **8**, 952-973 (2021).
20. Moučka R, Sedláčik M, *et al.* Mechanical properties of bulk Sylgard 184 and its extension with silicone oil. *Scientific Reports* **11**, 19090 (2021).
21. Peng J, Tomsia AP, *et al.* Stiff and tough PDMS-MMT layered nanocomposites visualized by AIE luminogens. *Nature Communications* **12**, 4539 (2021).

Point-to-Point Response to Reviewers' comments:

We are grateful to the reviewers for their in-depth reviews and important/constructive questions/suggestions regarding our manuscript. In the pages that follow, we provide our responses to each of the reviewer's questions.

Reviewer #1 (Remarks to the Author):

This research demonstrated an electric pulse-tuned piezotronic effect by fabricating an Ag/HfO₂/n-ZnO piezotronic tunnelling junction, in which the interface engineering by local piezoelectric polarization can be reversibly and accurately modulated by electric pulse. I would like to go with a minor revision before it is accepted. These are my comments/suggestions:

Response:

We are grateful to you for the detailed, in-depth, and constructive comments/suggestions on our work. These comments are highly valuable to our research, and we have improved the manuscript according to your suggestions.

Question #1

There are some earlier reports on HfO₂/n-ZnO-based piezotronic tunneling junction. The authors should point out the main findings by comparing those previous studies.

Response:

We thank you for this suggestion. In order to reveal the novelty of this work more clearly, here we will describe (1) the piezotronic tunneling junction in previous works, and (2) the novelty of piezotronic tunneling junction with *tunable* interface barrier.

(1) Piezotronic tunneling junctions in previous works

In 2019, the research team (including our group) led by Zhong Lin Wang coupled the piezoelectric effect with the quantum tunneling effect and used atomic force microscopy (AFM) to verify a giant switching ($>10^5$), and fast response (≈ 4.38 ms) in a novel piezotronic tunneling junction¹. This investigation offers initial evidence that the high performance of this tunneling junction is achieved by simultaneously modulating the height and width of the tunneling junction barrier through the

piezotronic effect in principle. Building upon the aforementioned groundwork, in 2022, our group further developed Ag/HfO₂/n-ZnO piezotronic tunneling strain sensor in the nanodevice scale ². The experimental results prove that the piezotronic tunneling strain sensor in the nanodevice scale can also achieve high strain sensing performance (gauge factor $\sim 3.8 \times 10^5$) as the demonstration under AFM.

These previous works utilized the strain-induced piezoelectric potential to regulate the tunneling barrier's height and width in parallel, thereby synergistically modulating the electrical transport process, and achieving a highly sensitive strain sensor based on piezotronic tunneling junctions. But for a particular tunneling junction, once it is fabricated, the interface barrier is fixed and can't be tuned, it is more impossible to be continuously tuned.

(2) Novelty of piezotronic tunneling junction with tunable interface barrier in this work

Different from previous works focusing on piezotronic modification of tunneling junctions, the main finding of this work lies in utilizing electric pulse stimulation to *tune* the barrier of tunneling junction and thus regulate the piezotronic effect in the tunneling junction.

Piezotronic devices usually possess high sensitivity to response to external mechanical stimuli in theory. As shown in **Figure R1**, the basic structure of piezotronic devices is mainly composed of piezoelectric semiconductor and electrodes at both ends. The electrical transport of the device is dependent on the interface barrier and highly sensitive to the interface barrier height. The strain-induced piezoelectric polarization charges and corresponding piezo-potentials at interfaces are able to linearly modulate the interface barrier height, and hence exponentially control the electrical transport across the interface ^{2, 3}. Because of this natural exponential relationship between input mechanical signal and output electrical signal, piezotronic device has inherent advantages in detecting mechanical stimuli. However, there is a disadvantage for piezotronic devices that once they are fabricated, the internal interface barrier is fixed, leading to the inability to tune the device performance. This means that the range of response current for the same strain will also be fixed, which

may render piezotronic devices unsuitable for certain applications. Since polarized interface engineering highly depends on the nature of the interface barrier, once the interface potential barrier is established, its fixed value and uncontrollability are not beneficial to expand the performance and application scenarios of polarization interface engineering.

If the interface barrier of piezotronic tunneling junction can be tuned, the tunable piezotronic devices can be realized for more scenarios and wider applications. Besides, it may also inspire the researches on tunable electronic devices based on some other interface engineering by local polarizations like flexoelectric, and ferroelectric polarization (flexoelectronic devices and ferroelectric electronic devices) ⁴⁻⁶. Therefore, developing the strategy for piezotronic tunneling junction with tunable interface barrier is important.

As presented in the manuscript (Figure 3, or the following **Figure R2** for convenience), a strategy of tuning piezotronic effect reversibly and accurately by electric pulse is achieved in the constructed Ag/HfO₂/n-ZnO piezotronic tunneling junction. Considering the obvious regulation effect of electric pulse on the interface barrier in piezotronics devices, our work conquered the current disadvantage of fixed interface barrier in piezotronics.

Through above discussions, we can find that the main finding of this work is totally different from previous works on piezotronic tunneling junctions. All piezotronic tunneling junctions in past works are about fixed interface barriers. The novelty of this work lies in **applying electric pulse stimulation to achieve tunable barrier of tunneling junction and thus regulate the piezotronic effect** in the tunneling junction, which gives a new research dimension for this field. We thank you again for this constructive suggestion. In order to improve the quality and understanding of our work, we have added above discussions into the revised Supplementary Information (Supplementary Note 7 and Supplementary Fig. 13) and revised the ‘**Working principle**’ section to make the novelty be clearer expressed.

The ‘**Working principle**’ section in the manuscript has been revised to: “The corresponding energy bands This synergistically modulation makes the

Ag/HfO₂/n-ZnO tunneling junction with high strain sensitivity, which is the advantage of using Ag/HfO₂/n-ZnO tunneling junction here. What needs attention is that the modulation of interface barrier by piezoelectric polarization ranges from $\varphi_0/2$ to $(\varphi_0 - \Delta\varphi_{\text{piezo}})/2$, where $\varphi_0/2$ is the initial interface barrier height. But for a particular tunneling junction, once it is fabricated, the interface barrier is fixed and can't be tuned, it is more impossible to be continuously tuned. This initially fixed barrier height and uncontrollability are not beneficial to expand the performance and application scenarios of piezotronics. The key working principle here lies in utilizing electric pulse stimulation to achieve **tunable** barrier of tunneling junction and thus regulate the piezotronic effect in the tunneling junction (Supplementary Note 7 and Supplementary Fig. 13).”

Figure R1. The basic typical structure of piezotronic devices.

Figure R2. Regulation of piezotronic effect by electric pulse. a, Introducing a strain

(i) and an electric pulse (ii) to the Ag/HfO₂/n-ZnO piezotronic tunneling junction to tune the interface barrier. **b, c**, Strain-dependent *I-V* curves of piezotronic tunneling strain sensor before (**b**), immediately after (**c**) electric pulse of 80 V for 30 s.

Question #2

Why HfO₂ is chosen as an insulation layer in this study?

Response:

We thank you very much for this comment. There are two reasons for specifically choosing HfO₂ as an insulation layer.

Firstly, there are many literature reports that the dielectric insulation layer with high *k* values and wide band gaps not only can reduce the leakage current and the overall power consumption during device operation, but also has good stability to withstand breakdown. As compared with SiO₂, Al₂O₃, MgO and some other insulation materials, HfO₂ has advantages in terms of *k* value and band gap, as shown in **Table R1**⁷⁻¹¹. So, we chose HfO₂ at the beginning as the dielectric insulation layer of the piezotronic tunneling junction.

Secondly, it is also based on our studies about the influence of insulation materials on the performance of piezotronic tunneling junction. We once studied the performance of piezotronic tunneling devices using HfO₂, ZrO₂, Al₂O₃ and SiO₂ as insulation layers with a same thickness of ~1.8 nm under the conditions of tensile strain of 0.00%, 0.033%, 0.067% and 0.10%, respectively. As shown in **Figure R3**, the current values measured at +3 V bias for four kinds of piezotronic tunneling devices were selected. The number of devices used for each type is 20. It can be clearly seen from the figure that when the strain is 0.00%, the current values of the four types of piezotronic tunneling devices are randomly distributed between 0.01 and 7 nA, and when the strain is 0.10%, their current values are also irregularly distributed between 30 and 250 nA. According to the current data in **Figure R3**, the current on/off ratio and change of Schottky barrier height of four types of piezotronic tunneling devices under 0.10% strain were calculated, counted and compared in **Figure R4**. The comparison results show that changing the dielectric insulation

materials such as HfO₂, ZrO₂, Al₂O₃ and SiO₂ has almost no influence on the sensing performance of piezotronic tunneling devices. The classical carrier transport of metal-insulator-semiconductor (MIS) tunneling junction can be expressed as ⁶:

$$I = SA^*T^2 \exp(-\alpha_T d \sqrt{q\varphi_T}) \exp\left(-\frac{q\varphi_B}{kT}\right) \left[\exp\left(\frac{qV}{nkT}\right) - 1 \right] \quad (1)$$

where $\exp(-\alpha_T d \sqrt{q\varphi_T})$ is the tunneling probability term, S is the contact area of the tunneling barrier, A^* is the effective Richardson constant, q represents the charge of an electron, k is the Boltzmann constant, T is the absolute temperature, φ_B represents the height of the Schottky barrier, n is the ideality factor, d is the thickness of the insulation layer in the tunneling junction, $\alpha_T = 2\sqrt{2qm^*}/\hbar$ (m^* represents the effective mass of the charge carriers, \hbar is the reduced Planck constant), φ_T represents the effective height of the tunneling barrier. It can be seen that the tunneling current is mainly dependent on the thickness of insulator layer (d) and the Schottky barrier height (φ_B) (which is determined by the metal's work function and the semiconductor's electron affinity), and slightly dependent on the band gap of insulator layer (which together with its electron affinity will affect the effective height of the tunneling barrier $q\varphi_T$), and almost not dependent on the k value. Given that the band gaps of HfO₂, ZrO₂, Al₂O₃ and SiO₂ are not very different, the piezotronic modification of tunneling current almost doesn't change with the choice of dielectric insulation layer, which is consistent with our experimental results in **Figures R3** and **R4**. So, it is no problem to choose either HfO₂, ZrO₂, Al₂O₃ or SiO₂ to construct the tunnel junction.

Based on above reasons, we chose HfO₂ as the insulation layer for convenience, which can be precisely fabricated by atomic layer deposition (ALD). Actually, other dielectric materials including ZrO₂, Al₂O₃ and SiO₂ can also be candidates as insulation layer. We have added above discussions into the revised Supplementary Information (Supplementary Note 2, Supplementary Figs. 1-2 and Supplementary Table 1). Thank you again for your constructive comment.

The '**Working principle**' section in the manuscript has been revised to: "Based on above principle, we Here, we choose HfO₂ with a thickness of about 1.8 nm

as the insulation layer.....”

The ‘**Methods**’ section in the manuscript has been revised to: “.....Finally, controllable preparation of HfO₂ insulation layer of about 1.8 nm can be realized. Since the PTJ with a 1.8 nm HfO₂ possesses the best piezotronic modification of electrical transport, the thickness of about 1.8 nm is chosen. In addition, our further experiments indicate that the HfO₂ can also be replaced with Al₂O₃, SiO₂, and ZrO₂. More detailed discussions can be found in the Supplementary Note 2, Supplementary Figs. 1-4 and Supplementary Table 1.”

Table R1. Dielectric constants and band gaps of typical insulation materials⁷⁻¹¹.

Insulation materials	Dielectric constants k	Band gaps E_g (eV)
HfO ₂	20~25	5.6~6.0
ZrO ₂	17~25	5.1~7.8
SiO ₂	3.9	9
Si ₃ N ₄	7~7.5	5~5.3
Al ₂ O ₃	9	8.8
MgO	8.4~9.8	8.7~8.8

Figure R3. Influence of insulation materials on piezotronic modification of the carrier transport of piezotronic tunneling devices. a-d, Piezotronic modification of

the carrier transports of four types of piezotronic tunneling devices using HfO₂ (a), ZrO₂ (b), Al₂O₃ (c) and SiO₂ (d) as insulation layers under the conditions of tensile strain of 0.00%, 0.033%, 0.067% and 0.10%, respectively. The above currents were measured at bias of +3.0 V.

Figure R4. Sensing performance of piezotronic tunneling devices. a, b, On/off ratio (a) and change of effective Schottky barrier height (b) of the devices based on piezotronic tunneling junction with different insulation materials of HfO₂, ZrO₂, Al₂O₃ and SiO₂. The error bars denote standard deviations of the mean.

Question #3

Did the author check the effect of post-annealing of as-prepared ZnO nano/microstructures on the piezotronic performances? The formation of Schottky junction is affected by the density of oxygen vacancy near the interface and hence the barrier height. The author should show the experimental evidence and comment on the presence and nature of the oxygen vacancies of as prepared ZnO and their influence on the electric pulse-induced oxygen vacancy migration.

Response:

Thank you very much for your nice suggestion. We have conducted relevant experiments to check the influence of ZnO post-annealing on the piezotronic performances of devices, and their influence on the electric pulse-induced oxygen vacancy migration. We will discuss it in the following three parts. And the relevant contents have been added into the revised manuscript and Supplementary Information (Supplementary Note 18 and Supplementary Figs. 36-37).

(1) Influence of post-annealing on different valence oxygen vacancies in ZnO

We annealed ZnO in air and characterized the influence of annealing on different

valence oxygen vacancies in ZnO. In ZnO and oxygen poor condition, the formation energies of oxygen vacancies in different charge states are calculated as a function of the Fermi level, as shown in **Figure R5**. The valence band maximum (E_{VBM}) is set as zero. The formation energy of Vo^0 is 1.0 eV and generally larger than that of Vo^{2+} , until the Fermi level surpasses the thermal transition energy $\varepsilon(2+/0)$ (0.5 eV below the conduction band maximum, E_{CBM}). Vo^{1+} is generally unstable due to its formation energy lies between the neutral and bivalent oxygen vacancy defect formation energy^{12, 13}. Due to the difference in formation energy, the transition of oxygen vacancy between different states, especially that between Vo^0 and Vo^{2+} is possible: Vo^0 can transform to Vo^{2+} thermally, and the transformation would be reversed when Fermi level is larger than thermal transition energy.

The relative concentration of oxygen vacancy states cannot be directly obtained experimentally except that of Vo^{1+} , which can be characterized by EPR (**Figure R6**). The peak position in EPR spectra that associated with Vo^{1+} is still controversial, as connections with $g \sim 1.990, 1.960, 1.996$ and 2.003 are reported¹⁴⁻¹⁸, and its relative intensity can be used to indicate the relative concentration of Vo^{1+} ¹⁴. This indicates that the Vo^{1+} concentration in ZnO annealed in different temperatures is quite low, and the conductivity is correlated with the concentrations of Vo^0 and Vo^{2+} .

As a relative indirect method, PL can be used to characterize Vo in ZnO¹⁸⁻²¹. However, due to the complex defect structure in ZnO, various sources are proposed to explain the PL spectra, especially the visible emission of ZnO: shallow donor, oxygen vacancy with different states, zinc vacancy, or oxygen and zinc interstitials¹⁸⁻²⁴. To pinpoint the origin of different defect emissions, PL results must be analyzed in combination with other methods. The PL spectra of ZnO annealed in different temperatures are shown in **Figure R7a**. The intensity is normalized by the intensity of UV emission. The spectra can be divided into three emission regions: UV emission (**Figure R7b**), blue emission (**Figure R7c**) and red emission (**Figure R7d**). Interestingly, a similar pattern is found in these emission regions. The UV emission shows a red shift (380 to 395 nm, 3.26 to 3.13 eV) with the increase of annealing temperature from 200 to 400 °C and the shift is reversed (395 to 385 nm, 3.13 to 3.22

eV) with the further increase of the annealing temperature. For the blue/green emission (430–490 nm), the intensity increases with the increase of annealing temperature from 200 to 400 °C, and the trend is reversed with the further increase of annealing temperature. For the orange/red emission (610 to 720 nm), the intensity decreases with the increase of annealing temperature from 200 to 350 °C, and the correlation is reversed with further increase of the annealing temperature (the peak at 600 nm is caused by the 300 nm excitation source used). The annealing temperature around 400 °C is a turning point for all three emission regions.

The UV emission is related with the exciton emission near the band gap. The red shift of UV emission has been observed previously¹⁸, and is due to the increase of carrier density introduced by oxygen vacancies. In other words, when the post-annealing temperature increases from 200 to 400 °C, the concentration of Vo^{2+} increases, and further increase of temperature would result to the decrease of Vo^{2+} concentration. As to the blue/green emission (430–490 nm), it is attributed to the defect emission of oxygen vacancy^{20, 21, 23, 25-27}, transition between Vo^{1+} and photoexcited holes^{18, 22, 28}, transition involves Vo^{2+} ²⁹, or surface defects³⁰. Since the EPR spectra indicates that the concentration of Vo^{1+} is negligible, the transition that involves Vo^{1+} can be excluded. As to the orange/red emission, transitions related with interstitial oxygen^{31, 32}, surface defects³³, and oxygen vacancy are proposed³⁴⁻³⁶.

(2) Influence of the vacancy concentration on the interface carrier transport in Schottky junctions

We performed post-annealing on the ZnO nanowires in air at various temperatures: 200 °C, 300 °C, 400 °C, and 450 °C. And then, the *I-V* properties of devices with untreated and annealed ZnO were analyzed. **Figure R8a-e** show *I-V* curves for the devices under 0.00% and 0.10% compressive strain, revealing the existence of piezotronic effect at various annealing temperatures. At 0.10% compressive strain, higher current responses under positive bias and reduced responses under negative bias were observed. **Figure R8f** gives the changes in *I-V* curves, with peak current at 400 °C, influenced by oxygen vacancies on the Schottky barrier height. **Figure R8g** summarizes the current at ± 1 V bias for 0.00% and -0.10%

strain. The current increases with annealing temperature, indicating that annealing increases the carrier concentration and oxygen vacancy Vo^{2+} density. In addition, the current on/off ratio in **Figure R8h** decreases with temperature increasing from 0 °C to 400 °C. Modulation of Schottky barrier height in **Figure R8i** is consistent with the current on/off ratio trend. The results indicate that the impact of piezoelectric polarization on barrier height diminishes at 400 °C due to the increased oxygen vacancy density and stronger carrier shielding. However, at 450 °C, a decrease in positively charged oxygen vacancies (Vo^{2+} , which will be transited into Vo^0 above 400 °C) weakens this effect, resulting in higher current on/off ratios and Schottky barrier heights.

(3) Influence of the vacancy concentration on regulation of electric stimulation

As illustrated in **Figure R9a**, we exposed the devices to electric pulse stimulation with an 80 V amplitude and a 10 Hz frequency. **Figures R9b-f** show the I - V characteristics of these devices both before and after a 60 s electric pulse stimulation. The measured I - V curves indicate that the electric pulse can significantly increase the devices' current. **Figure R9g** summarizes the relationship between the current value at +1 V and the annealing temperature. The increased current before electric pulse stimulation indicates that the density of vacancy in n -ZnO nanowires increases with the increase of annealing temperature. Meanwhile, **Figure R9h** plots the current ratio before and after electric pulse as a function of the annealing temperature. Notably, this curve peaks at 400 °C, designating it as a critical inflection point, beyond which (at 450 °C) the current ratio slightly increases. The decreasing trend of current ratio indicates that with the increase of annealing temperature, the modulation of electric pulse on interface barrier tends to be weaker. This phenomenon can be easily understood as follows. As the annealing temperature increases, the density of vacancy as well as the free carrier concentration both increases. However, the increase of carrier concentration will electrostatically shield the vacancies and weaken the regulation of vacancy on the interface barrier. This phenomenon is similar to the weakened piezotronic effect or the weakened flexoelectronic effect occurred in high doping semiconductors. **Figure R9i** summaries the recovery time after electric

pulse as a function of the annealing temperature. As can be found that the recovery time tends to increase with the increase of annealing temperature and reaches its peak value at 400 °C. This tell us that with the increase of vacancy density, the gradient of vacancy concentration after electric pulse stimulation is relatively small, and it will take more time to recover to its original state.

In summary, with the increase of post-annealing temperature, the vacancy Vo^{2+} concentration increases as a whole, which makes the device current increases (under a same bias) but weakens the modulations of the piezoelectric polarization and the electric pulse on the interface barrier. The weakening of electric pulse's and piezotronic modulations on the interface barrier by high vacancy Vo^{2+} concentration by post-annealing is very similar to the weakening of piezotronic effect caused by doping in the past. Thank you again for your nice comment.

The '**Electric pulse-tuned interface barrier**' section in the manuscript has been revised to: "In order to clarify In other words, the current will become larger and last longer with a higher electric pulse voltage. Some other ways, such as proper annealing of the piezoelectric semiconductor (*e.g.*, *n*-ZnO), may be also able to achieve a longer retention (Supplementary Note 9 and Supplementary Fig. 17)."

In the '**Electric pulse-tuned piezotronic effect**' section in the manuscript, another paragraph is added: "We also checked the influence of post-annealing of as-prepared ZnO nanowires on the device performances (Supplementary Note 18). The post-annealing with temperature ranged from 0 °C to 400 °C tends to improve the conductivity of devices, indicating an increased carrier concentration and vacancy density inside the ZnO. As can be seen from Supplementary Figs. 36-37, modulations on electrical transport by piezoelectric polarization and electric pulse stimulation are both weakened by increasing post-annealing temperature. These weakened modulation result from the stronger electrostatic shielding effect of carriers. Due to the increased vacancy density, the vacancy gradient caused by electric pulse is smaller, so the recovery time is slightly increased with increased annealing temperature. The post-annealing maybe a method to achieve long retention of the modulation by electric pulse."

Figure R5. Relationship between the formation energy of oxygen vacancy in different states and the Fermi level.

Figure R6. EPR spectra of ZnO annealed in different temperatures.

Figure R7. PL spectra of ZnO annealed in different temperatures. a, Total PL spectra of ZnO in different annealing temperatures. b-d, PL spectra in UV region (b),

blue region (c) and red region (d). The spectra are normalized by the intensity of near-band emission.

Figure R8. Characterization of the influence of annealing temperature on piezotronic properties. a-e, *I-V* curves for Schottky barrier devices incorporating untreated and annealed ZnO nano/microwires, evaluated under two distinct compressive strain conditions: 0.00% and 0.10%. The annealing temperatures are 200 °C, 300 °C, 400 °C, and 450 °C, respectively. f, *I-V* characteristic trends across varying annealing temperatures. g, Current values under ± 1 V bias for the 0.00% and 0.10% compressive strain conditions. h, i, Current on/off ratio (h) and change of Schottky barrier height (i) as functions of different annealing temperatures under ± 1 V bias.

Figure R9. Carrier transport characteristics before and after electrical stimulation after annealing. **a**, Electric pulse characterized by an 80 V amplitude and a 10 Hz frequency. **b-f**, I - V characteristics before and after electrical stimulation under different heating temperature. **g**, The relationship between the current value at a 1 V positive bias and the annealing temperature. **h**, The variations in current ratio before and after electrical stimulation as a function of annealing temperature. **i**, The recovery time of the device's current after electrical stimulation.

Reference:

1. Liu, S. *et al.* Piezotronic tunneling junction gated by mechanical stimuli. *Advanced Materials* **31**, 1905436 (2019).
2. Yu, Q. *et al.* Highly sensitive strain sensors based on piezotronic tunneling junction. *Nature communications* **13**, 778 (2022).
3. Liu, S. *et al.* Statistical piezotronic effect in nanocrystal bulk by anisotropic

- geometry control. *Advanced Functional Materials* **31**, 2010339 (2021).
4. Meng, J., Li, Z. Schottky-contacted nanowire sensors. *Advanced Materials* **32**, 2000130 (2020).
 5. Wang, L., Wang, Z. L. Advances in piezotronic transistors and piezotronics. *Nano Today* **37**, 101108 (2021).
 6. Sze, S. M. & K, K. N. Physics of semiconductor devices, 3rd Edition. (Wiley, 2006).
 7. Nahar, R., Singh, V., Sharma, A. Study of electrical and microstructure properties of high dielectric hafnium oxide thin film for MOS devices. *Journal of Materials Science: Materials in Electronics* **18**, 615-619 (2007).
 8. Gilmer, D. *et al.* Compatibility of polycrystalline silicon gate deposition with HfO₂ and Al₂O₃/HfO₂ gate dielectrics. *Applied physics letters* **81**, 1288-1290 (2002).
 9. Kang, S., *et al.* Effect of deposition conditions of poly Si_{1-x}Ge_x films and Ge atoms on the electrical properties of poly Si_{1-x}Ge_x (x= 0, 0.6)/HfO₂ gate stack. *Journal of applied physics* **94**, 4608-4613 (2003).
 10. Hinkle, C., Fulton, C. *et al.* Enhanced tunneling in stacked gate dielectrics with ultra-thin HfO₂ (ZrO₂) layers sandwiched between thicker SiO₂ layers. *Applied Surface Science* **234**, 240-245 (2004).
 11. Gerritsen, E. *et al.* Evolution of materials technology for stacked-capacitors in 65 nm embedded-DRAM. *Solid-State Electronics* **49**, 1767-1775 (2005).
 12. Oba, F. *et al.* Defect energetics in ZnO: a hybrid Hartree-Fock density functional study. *Physical Review B* **77**, 245202 (2008).
 13. Agoston, P. *et al.* Intrinsic n-type behavior in transparent conducting oxides: a comparative hybrid-functional study of In₂O₃, SnO₂, and ZnO. *Physical review letters*, **103**, 245501 (2009).
 14. Reddy, A. J. *et al.* Structural, optical and EPR studies on ZnO: Cu nanopowders prepared via low temperature solution combustion synthesis. *Journal of Alloys and Compounds* **509**, 5349-5355 (2011).
 15. Kaftelen, H. *et al.* EPR and photoluminescence spectroscopy studies on the

- defect structure of ZnO nanocrystals. *Physical Review B* **86**, 014113 (2012).
16. Evans, S. M. *et al.* Further characterization of oxygen vacancies and zinc vacancies in electron-irradiated ZnO. *Journal of Applied Physics* **103**, 043710 (2008).
 17. Wang, X. J. *et al.* Oxygen and zinc vacancies in as-grown ZnO single crystals. *Journal of Physics D: Applied Physics* **42**, 175411 (2009).
 18. Liu, H. *et al.* Correlation of oxygen vacancy variations to band gap changes in epitaxial ZnO thin films. *Applied Physics Letters* **102**, 181908 (2013).
 19. Li, Y. *et al.* Surface superoxide complex defects-boosted ultrasensitive ppb-level NO₂ gas sensors. *Small* **12**, 1420-1424 (2016).
 20. Yu, L. *et al.* Both oxygen vacancies defects and porosity facilitated NO₂ gas sensing response in 2D ZnO nanowalls at room temperature. *Journal of Alloys and Compounds* **682**, 352-356 (2016).
 21. Wei, X. Q. *et al.* Blue luminescent centers and microstructural evaluation by XPS and Raman in ZnO thin films annealed in vacuum, N₂ and O₂. *Physica B: Condensed Matter* **388**, 145-152 (2007).
 22. Sambandam, B. *et al.* Oxygen vacancies and intense luminescence in manganese loaded ZnO microflowers for visible light water splitting. *Nanoscale* **7**, 13935-13942 (2015).
 23. Børseth, T. M. *et al.* Identification of oxygen and zinc vacancy optical signals in ZnO. *Applied Physics Letters* **89**, 262112 (2006).
 24. Djurišić, A. B. *et al.* Green, yellow, and orange defect emission from ZnO nanostructures: Influence of excitation wavelength. *Applied Physics Letters* **88**, 103107 (2006).
 25. Wang, Q. P. *et al.* Photoluminescence of ZnO films prepared by R. F. sputtering on different substrates. *Applied Surface Science* **220**, 12-18 (2003).
 26. Kang, H. S. *et al.* Annealing effect on the property of ultraviolet and green emissions of ZnO thin films. *Journal of Applied Physics* **95**, 1246-1250 (2004).
 27. Tong, Y. H. *et al.* The optical properties of ZnO nanoparticles capped with polyvinyl butyral. *Journal of Sol-Gel Science and Technology* **30**, 157-161

- (2004).
28. Vanheusden, K. *et al.* Correlation between photoluminescence and oxygen vacancies in ZnO phosphors. *Applied Physics Letters* **68**, 403-405 (1996).
 29. Van Dijken, A. *et al.* The kinetics of the radiative and nonradiative processes in nanocrystalline ZnO particles upon photoexcitation. *The Journal of Physical Chemistry B* **104**, 1715-1723 (2000).
 30. Meyer, B., Marx, D. Density-functional study of the structure and stability of ZnO surfaces. *Physical Review B* **67**, 035403 (2003).
 31. Greene, L. E. *et al.* Low-temperature wafer-scale production of ZnO nanowire arrays. *Angewandte Chemie, International Edition* **42**, 3031-3034 (2003).
 32. Kwok, W. M. *et al.* Time-resolved photoluminescence study of the stimulated emission in ZnO nanoneedles. *Applied Physics Letters* **87**, 093108 (2005).
 33. Ong, H. C., Du, G. T. The evolution of defect emissions in oxygen-deficient and -surplus ZnO thin films: The implication of different growth modes. *Journal of Crystal Growth* **265**, 471-475 (2004).
 34. Kumar, V. *et al.* Origin of the red emission in zinc oxide nanophosphors. *Materials Letters* **101**, 57-60 (2013).
 35. Studenikin, S. A. *et al.* Fabrication of green and orange photoluminescent, undoped ZnO films using spray pyrolysis. *Journal of Applied Physics* **84**, 2287-2294 (1998).
 36. Kong, Y. C. *et al.* Ultraviolet-emitting ZnO nanowires synthesized by a physical vapor deposition approach. *Applied Physics Letters* **78**, 407-409 (2001).

Reviewer #2 (Remarks to the Author):

The authors reported a novel approach of interface engineering by applying electric pulses on Ag/HfO₂/n-ZnO piezotronic tunneling junction device. The interface barrier height can reversibly be tuned up to 168.11 meV by 80 V electric pulses. And the piezotronic modification on interface barrier tuned by electric pulse can be up to 148.81 meV under a strain of 1.34%. The article demonstrated sufficient experimental results on the phenomenon of tunable interface barrier height by the stimulation of electric pulse and bending-induced strain, and provided a reasonable theoretical analysis of interfacial engineering. It shows a novelty in the further investigation of piezotronics, and gives a guide to design and optimize novel piezotronic devices based on low-dimensional materials (*e.g.*, ZnO, GaN, MoS₂). The reviewer would like to recommend it be published in Nature Communications, however, some critical issues remain to be addressed.

Response:

We sincerely appreciate your positive comments and important issues. And we have carefully revised the manuscript according to your very valuable comments and suggestions.

Question #1

In the Introduction, the authors introduced “flexoelectronics and ferroelectric electronics” and discussed their working principles in Fig. 1. However, only piezotronics was investigated in the manuscript. It seems to be redundant to talk about flexoelectronics and ferroelectric electronics.

Response:

We thank you very much for the comment. Considering the theoretical and experimental investigations indeed involve only the piezotronics in this work, the schematic diagrams of the interface engineering by local polarizations in flexoelectronics and ferroelectric electronics in Figure 1 have been removed. According to your suggestion, we have revised Figure 1 by deleting the working principles of the flexoelectronics and the ferroelectric electronics, with the

corresponding contents in the ‘**Abstract**’, ‘**Introduction**’ and ‘**Conclusion**’ section modified. Figure 1 in the revised manuscript is also listed as the following **Figure R10** for your convenience. Thank you again for this important suggestion.

The ‘**Abstract**’ section in the manuscript has been revised to: “Investigating interface engineering This study provides opportunities to achieve reversible control of piezotronics, and extend them to a wider range of scenarios and be better suitable for micro/nano-electromechanical systems.”

The ‘**Introduction**’ section in the manuscript has been revised to: “The fundamental principle **Fig. 1a** illustrates the typical piezotronics, and the corresponding energy bands at interfaces modulated by strain-induced piezoelectric polarization. When a strain is applied on the piezoelectric semiconductor, piezoelectric charges and the corresponding piezoelectric potential will appear at the interface, which will bend the energy band and thus control the electrical transport of the interface. As can be seen, the construction of interface barrier

“Here, we demonstrate These findings give an in-depth understanding for the coupling effect between the electric field-induced redistribution of vacancies and local polarization-controlled interface engineering, and provide the opportunities to achieve reversible control of piezotronics.”

The ‘**Conclusion**’ section in the manuscript has been revised to: “In summary, we have This work not only provides in-depth understanding for the coupling between electric field-induced redistribution of vacancies and local polarization-controlled interface engineering, but also exhibits the possibility to realize tunable piezotronics with a non-destructive technique, which makes it has great application potential for micro/nano-electromechanical systems and many more.”

Figure R10. (Figure 1 in the revised manuscript) Working principle of electric pulse-tuned piezotronic effect on Ag/HfO₂/n-ZnO piezotronic tunneling junction (PTJ). a, Energy band diagrams of interface engineering by local polarizations in piezotronics. b, Effect of electric pulse on the oxygen vacancies in semiconductors (left) and corresponding modification of interface barrier height ($\Delta\varphi_{\text{pulse}}$) (right). c, Introducing a strain (i) and an electric pulse (ii) to the Ag/HfO₂/n-ZnO PTJ to tune the interface barrier. d, Corresponding energy bands of Ag/HfO₂/n-ZnO PTJ in c. (i) The modulation of interface barrier by piezoelectric polarization ranges from $\varphi_0/2$ to $(\varphi_0 - \Delta\varphi_{\text{piezo}})/2$, where $\varphi_0/2$ is the initial interface barrier height. (ii) The electric pulse-induced oxygen vacancy migration changes the interface barrier by $\Delta\varphi_{\text{pulse}}/2$, and thus tune the piezotronic modulation range from $(\varphi_0 - \Delta\varphi_{\text{pulse}})/2$ to $(\varphi_0 - \Delta\varphi_{\text{pulse}} - \Delta\varphi'_{\text{piezo}})/2$.

Question #2

Why did use an electric pulse in the investigation on piezotronics? The authors need to give more discussion on the advantages or disadvantages of piezotronic devices.

Response:

We thank you very much for this valuable comment.

(1) Why did use an electric pulse in the investigation on piezotronics?

Importance of using electric pulse to tune piezotronics

The performance of piezotronic device is strongly dependent on the properties of interface barrier, but the interface barrier is completely fixed and cannot be changed once it is constructed. This makes the performance of the piezotronic device completely fixed once it is fabricated. For example, for a piezotronic sensor, its working current range and sensitivity will be fixed. If the interface barrier of piezotronic device can be tuned, more scenarios and wider applications can be realized for the tunable piezotronic devices. Besides, it may also inspire the researches on tunable electronic devices based on some other interface engineering by local polarizations like flexoelectric, and ferroelectric polarization (flexoelectronic devices and ferroelectric electronic devices) ¹⁻³. Therefore, developing the strategy for piezotronic devices with tunable interface barrier is important.

Considering the obvious modulation of electric pulse on the interface barrier of piezotronic device as presented in the manuscript (Figure 3), we use an electric pulse in the investigation on piezotronics.

Reasons to use electric pulse rather than DC voltage to tune piezotronics

There are several reasons to use electric pulse rather than DC voltage to stimulate devices ⁴⁻⁶. ① Avoiding thermal losses and degradation: DC voltage generates continuous current in electronics, producing heat energy, which may lead to material performance degradation or thermal degradation over time. In comparison, electric pulse can provide energy in a shorter time, reducing heat accumulation. ② Improving efficiency: Electric pulses can deliver high peak power in a short time, thereby more efficiently exciting devices, especially in applications requiring rapid response. ③ Reducing power consumption: Compared to DC voltage, electric pulses provide electrical energy only when stimulation is needed, reducing the average power consumption. ④ Control and precision: The duration, magnitude, and frequency of electric pulses can be precisely controlled, enabling precise signal modulation and processing. ⑤ Device protection: Higher DC voltage may damage piezoelectric

semiconductors, while electric pulse can use higher peak voltage without causing long-term penetrative damage to the devices. ⑥ Enhancing signal resolution: In some applications, using electric pulse can improve the resolution of detection signals by optimizing the time resolution of the signal through adjustment of pulse width and shape. ⑦ Interface state control: Rapid electric field changes generated by pulse voltage at the interface of piezoelectric semiconductors provide better control over interface charge states and charge transfer processes. ⑧ Reducing electrode and material degradation: DC voltage may cause chemical degradation at the electrode and material interface, while electric pulse, due to its intermittence, reduces the likelihood of this chemical degradation.

(2) The authors need to give more discussion on the advantages or disadvantages of piezotronic devices.

Advantages of piezotronic devices:

Piezotronics based on local piezoelectric polarization is an important field in interface engineering. Because of the capacity of constructing adaptive and seamless interactions between electronics/machines and human/ambient, it is important in Internet of Things, artificial intelligence and biomedical engineering. Since strain-induced piezoelectric polarization can control the electrical transport of devices, piezotronic devices can be used as mechanical sensors by measuring the strain-controlled electrical transport, as demonstrated in logic units ⁷⁻⁹, vibration sensing ¹, and tactile sensing ². Moreover, as a strain-induced interface engineering, piezotronics can also introduce regulation and optimization of the electronic or phototronic properties of devices through applying strain/force/pressure, as observed in applications such as light-emitting diodes ³, biomolecular detection ^{10, 11}, ultraviolet detection ¹², and lasers ¹³, *etc.* Besides those, if we can tune piezotronics, it will contribute to the further applications of piezotronics. Compared with other interface engineering methods by local polarizations like flexotronics ^{14, 15}, piezotronics is a typical strategy whose working mechanism is clear, and it is more important for pushing forward the developing the interface engineering.

Disadvantages of piezotronic device:

There is a disadvantage for piezotronic devices that once they are fabricated, the internal interface barrier is fixed, leading to the inability to tune the device performance. This means that the range of response current for the same strain will also be fixed, which may render piezotronic devices unsuitable for certain applications. Since polarized interface engineering highly depends on the nature of the interface barrier, once the interface potential barrier is established, its fixed value and uncontrollability are not beneficial to expand the performance and application scenarios of polarization interface engineering.

In addition, the relevant contents have been added into the revised manuscript and Supplementary Information (Supplementary Note 7 and Supplementary Fig. 13).

The ‘**Introduction**’ section in the manuscript has been revised to: “Here, we demonstrate These findings give an in-depth understanding for the coupling effect between the electric field-induced redistribution of vacancies and local polarization-controlled interface engineering, and provide the opportunities to achieve reversible control of piezotronics.”

The ‘**Working principle**’ section in the manuscript has been revised to: “The corresponding energy bands of Ag/HfO₂/n-ZnO tunneling junction....., and hence synergistically modulate the electrical transport of interface⁶ (Supplementary Note 6 and Supplementary Figs. 11-12). This synergistically modulation makes the Ag/HfO₂/n-ZnO tunneling junction with high strain sensitivity, which is the advantage of using Ag/HfO₂/n-ZnO tunneling junction here. What needs attention is that....., where $\varphi_0/2$ is the initial interface barrier height. But for a particular tunneling junction, once it is fabricated, the interface barrier is fixed and can’t be tuned, it is more impossible to be continuously tuned. This initially fixed barrier height and uncontrollability are not beneficial to expand the performance and application scenarios of piezotronicsThe key working principle here lies in utilizing electric pulse stimulation to achieve **tunable** barrier of tunneling junction and thus regulate the piezotronic effect in the tunneling junction (Supplementary Note 7 and Supplementary Fig. 13).”

The ‘**Conclusion**’ section in the manuscript has been revised to: “This work not only provides in-depth understanding for the coupling between electric field-induced redistribution of vacancies and local polarization-controlled interface engineering, but also exhibits the possibility to realize tunable piezotronics with a non-destructive technique, which makes it has great application potential for micro/nano-electromechanical systems and many more.”

Question #3

The electric pulse has a large amplitude of 80 V and a long period. Actually, the electric pulse with a range of 1 ns to 100 ms would be widely applied on various electronic devices. Is it possible to reduce the period time of the pulse, making it equivalent to that of a long pulse condition? And the different frequency of pulse needs to be conducted in the experiments. In addition, the voltage amplitude of 80 V is high enough and would easily cause device breakdown.

Response:

We thank you very much for these valuable suggestions. According to your comment, we have carried out experiments to study the influence of the period time (frequency) and amplitude of electric pulse on the performance of the devices.

(1) Is it possible to reduce the period time of the pulse, making it equivalent to that of a long pulse condition? And the different frequency of pulse needs to be conducted in the experiments.

As shown in **Figure R11**, by controlling the time interval between two electric pulses, we changed the period time (or frequency) of the electric pulses, and measured its influence on the electrical transport of the device. **Figure R11a** shows the electric pulses with voltage of 80 V, pulse width of 33 ms, and various frequencies of 20 Hz, 10 Hz, 5 Hz, 1 Hz, 0.5 Hz, and the corresponding currents. As can be seen, increasing the frequency (reducing the period time) of the electric pulses can effectively improve the increasing rate of the current. In addition, **Figure R11b** summarizes the recovery curves of the current labeled ‘1’ to ‘5’ in **Figure R11a**, indicating that improving the frequency (reducing the period time) of the electric pulse can enhance the initial

current of the device after stimulations. We also plotted the required time of electric pulse stimulation on the device to tune the device current to reach $18.7 \mu\text{A}$ in **Figure R11c**. It can be observed that the modulation of 20 Hz electric pulse (period time: 50 ms) on the device for 0.23 s is equivalent to that of 0.5 Hz electric pulse (period time: 2 s) for 20.1 s. This means that a high frequency electric pulse with short stimulation duration can make a same effect on the device as a low frequency electric pulse with long stimulation duration. Therefore, reducing the period time of the electric pulse can making the modulation of electric pulse on the device equivalent to that of a long pulse condition.

The above experiment has been added into the revised manuscript and Supplementary Information (Supplementary Note 10 and Supplementary Fig. 21).

The ‘**Electric pulse-tuned interface barrier**’ section in the manuscript has been revised to: “In **Fig. 2e**, Moreover, by increasing the pulse frequency (cycle number per second) and the pulse duration, we can further improve the impact of electric pulse on the current response and increase the recovery time (Supplementary Note 10 and Supplementary Fig. 21).”

Figure R11. Current response of the device under stimulations of electric pulse with different pulse frequency of 0.5 Hz, 1 Hz, 5 Hz, 10 Hz and 20 Hz. The pulse

frequency of 0.5 Hz, 1 Hz, 5 Hz, 10 Hz and 20 Hz also present the period time of 20 s, 1 s, 0.2 s, 0.1 s, 0.05 s, respectively. **a**, Waveforms of the applied pulse voltage and measured response current over time. **b**, Recovery curves of the current labeled '1' to '5' in **(a)**. The decreasing slopes of the current shown in **(b)** for each recovery curves have a unit of $\mu\text{m/s}$. **c**, Required time of electric pulse stimulation on the device to tune the response current to reach $18.7 \mu\text{A}$. The frequency is regulated by changing the time interval between two electric pulses.

(2) In addition, the voltage amplitude of 80 V is high enough and would easily cause device breakdown.

Generally, the electric pulse with large amplitude is easy to cause device breakdown. But for our work, as can be seen from Figure 3b-d in the manuscript, the current response can fully recover to its original state, indicating that the device during the experiments was not damaged after electric pulse stimulation with amplitude of 80 V.

Besides, we additionally studied the electrical characteristics of device under the periodic electric pulse stimulation with a frequency of 10 Hz and amplitudes of 70 V, 80 V, 100 V, 120 V and 150 V, respectively. **Figure R12** shows the electric pulse with various amplitudes (left side) applied on the device and their corresponding current responses (right side). As can be seen from **Figure R12a-d**, the current response to the pulses with amplitude from 70 V to 120 V all increase within 60 s (**Figure R12a-1** to **R12d-1**). However, when the amplitude of the pulse is 150 V (**Figure R12e**), the current response increases to $118 \mu\text{A}$ within 0.3 s, but then shows a decaying trend. And, as the periodic pulse voltage lasts for 60 s, the current response downs to $71.4 \mu\text{A}$. This indicates that the pulse amplitudes of 70 V, 80 V, 100 V, 120 V do not damage the device, but 150 V is high enough to cause device breakdown.

Figure R13a shows the I - V characteristics after 60 s electric pulse with amplitudes of 0 V, 70 V, 80 V, 100 V, 120 V and 150 V in **Figure R13a-d**. Consistent with the phenomenon shown in **Figure R12**, the electrical transport of the device continues to increase as the pulse amplitude increases from 0 V to 120 V, but attenuates after the pulse of 150 V (**Figure R13a**). **Figure R13b** summarizes the

current value at +1 V after the electric pulse with various amplitudes from **Figure R13a**. It can be observed that the current value drops significantly as the pulse amplitude reaching 150 V, which also means that 150 V is high enough to cause device breakdown.

Based on above discussion, we concluded that the devices used in this work can withstand electric pulse with amplitude under 120 V (for 60 s at 10 Hz). We are grateful to you for reminding us to measure whether the devices are broken down during electric pulse. And we have added this information into the revised manuscript and added this part of discussion into the Supplementary Information (Supplementary Note 10 and Supplementary Figs. 19-20).

The '**Electric pulse-tuned interface barrier**' section in the manuscript has been revised to: "In **Fig. 2e**, we studied the electrical transport of the tunneling junction after electric pulse (+80 V) of various durations with a frequency of 10 Hz (Supplementary Fig. 18). It need to be mentioned that the periodic pulse voltage of 80 V will not damage the tunneling junction, which still work normally after the pulse voltage stimulation (Supplementary Note 10 and Supplementary Figs. 19-20). It can be seen....."

Figure R12. Electric pulse voltage and the corresponding current response of the device. a-e, Amplitude of the electric pulse voltage stimulating the device is 70 V, 80 V, 100 V, 120 V and 150 V, respectively. Both of the time interval between electric pulses and the pulse duration are 0.05 s. **(a-1)-(e-1),** Current response corresponding to **(a-e)** within 60 s.

Figure R13. *I-V* characteristics after the electric pulse stimulation. a, *I-V* characteristics after the electric pulse stimulation for 60 s with different pulse amplitudes of 0 V, 70 V, 80 V, 100 V, 120 V and 150 V, respectively. **b, Current value at +3 V obtained from **a**. When the pulse voltage amplitude increases from 0 V to 120 V, the electrical transport characteristics of the device continue to increase, and it drops significantly at the pulse amplitudes of 150 V. This shows that under the stimulation of 80 V electric pulses, the device is in good condition.**

Question #4

AFM has the ability to implement the electric pulse mode. Could it conduct in-situ measurement of tunable interfacial barrier height by AFM?

Response:

Thank you very much for this insightful suggestion. We further performed in-situ measurement by atomic force microscope (AFM) to verify the tunable interface barrier height. Introducing programmed pulse voltages by the model of piezo-response force microscope (PFM model in AFM) and measuring the current by the model of conductive atomic force microscope (CAFM model in AFM), we can stimulate the nanowires firstly and perform subsequent *I-V* characterizations in situ. **Figure R14a** presents the optical image of the ZnO nanowire during in-situ measurement. **Figure R14b-e** shows the waveforms of periodic pulse voltages with amplitudes of 4 V, 6 V, 8 V, 10 V, and a frequency of 100 Hz. Following a pulse duration of 10 s, we carried out the *I-V* characteristics of the nanowire. As shown in **Figure R14f**, a clear presence of tunable current was observed, with the increment of

pulse voltage amplitudes. This indicates that electric pulse stimulation can effectively tune the interface barrier, and thus achieves tunable interface barrier height. In order to assist in proving tunable interface barrier height and improve the quality of our research, we have revised the manuscript and added the in-situ measurement into the revised Supplementary Information (Supplementary Note 11 and Supplementary Fig. 23).

The ‘**Electric pulse-tuned interface barrier**’ section in the manuscript has been revised to: “In **Fig. 2e**,the current gradually increases to a stable value as the stimulation time increases, implying that the interface barrier is indeed tuned by the applied electric pulse. This tunable interface barrier is also verified by an in-situ measurement utilizing Atomic Force Microscope (AFM) (Supplementary Note 11 and Supplementary Fig. 23).”

Figure R14. In-situ measurement of tunable interfacial barrier height by AFM. a, Optical image of ZnO nanowire during in-situ measurement. **b-e,** Applying periodic pulse voltages with amplitudes of 4 V, 6 V, 8 V, 10 V, and a frequency of 100 Hz on the ZnO nanowire. **f,** *I-V* characteristics measured after various electric pulse voltages.

Question #5

What is the thickness of ALD HfO₂? The authors claimed “when an electric pulse is

applied to cause oxygen vacancy migration inside the ZnO nanowire, there exists accumulation of oxygen vacancies near the left interface barrier”. If the HfO₂ thickness is less than 3-5 nm, tunneling is possible as the dominant mechanism for interfacial engineering. Or the HfO₂ layer would have some defects that could play an important role in trapping electrons from the ZnO nanowire. Please give more discussion about that.

Response:

We are grateful to you for these constructive suggestions.

(1) What is the thickness of ALD HfO₂?

In this work, an HfO₂ insulation layer with a thickness of about 1.8 nm was precisely grown on ZnO nanowire by atomic layer deposition (ALD). We used high-resolution transmission electron microscopy (HRTEM) to characterize the thickness of HfO₂ layer. The scanning electron microscopy (SEM) image in **Figure R15a-i** shows that the ZnO nanowire with a typical hexagonal geometry has a clean surface. The HRTEM image of HfO₂/n-ZnO (**Figure R15a-ii**) shows the HfO₂ thickness of about 1.8 nm. **Figure R15b** gives the distribution of HfO₂ thickness along the *x*-axis labeled in **Figure R15a-i** and indicates a uniform HfO₂ thickness of about 1.8 (± 0.3) nm. The inset shows the statistical distribution of the insulation thickness.

Here, we will briefly explain why the 1.8 nm thick insulation layer was chosen. To show the influence of insulation layer thickness on the performance of piezotronic tunneling devices, Ag/HfO₂/n-ZnO devices with HfO₂ thickness of 0.0 nm, 0.4 nm, 1.1 nm, 1.8 nm, 2.5 nm, 3.6 nm, and 7.3 nm were prepared. Next, we studied the piezotronic modifications of electrical transport of 140 devices (20 devices of each thickness) under 0.10% tensile strain. The experimental results show that the device with 1.8 nm HfO₂ exhibit the highest current on/off ratio (**Figure R16**), which makes us choose 1.8 nm HfO₂ as the insulation layer of piezotronic tunneling junctions. We have revised the manuscript to give the thickness of HfO₂ and added above discussion into the revised Supplementary Information (Supplementary Note 2, and Supplementary Figs. 1-4 and Supplementary Table 1).

The ‘**Working principle**’ section in the manuscript has been revised to: “Based

on above principle, Here, we choose HfO₂ with a thickness of about 1.8 nm as the insulation layer and Ag as the electrode to fabricate the Ag/HfO₂/n-ZnO PTJ for this structure possessing good performance on piezotronic modification of electrical transport (Supplementary Note 2, and Supplementary Figs. 1-4 and Supplementary Table 1). The material.....”

The ‘**Methods**’ section in the manuscript has been revised to: “**Fabrication of nanometer-thick HfO₂ insulation layer.** The grown process of HfO₂ Finally, controllable preparation of HfO₂ insulation layer of about 1.8 nm can be realized. Since the PTJ with a 1.8 nm HfO₂ possesses the best piezotronic modification of electrical transport, the thickness of about 1.8 nm is chosen.....”

Figure R15. Morphology characterization of HfO₂/n-ZnO nanowire and thickness characterization of deposited HfO₂ layer. a-i, SEM image of the ZnO nanowire with a diameter of about 8 μm, showing a typical hexagonal geometry and a clean surface. **a-ii,** High-resolution TEM image of HfO₂ layer deposited on ZnO surface. **b,** Thickness of HfO₂ along the x-axis labeled in **a-i**, showing the HfO₂ thickness of about 1.8 (±0.3) nm. The inset shows the statistical distribution of the insulation thickness.

Figure R16. Current on/off ratio of Ag/HfO₂/n-ZnO devices with 0.0 nm, 0.4 nm, 1.1 nm, 1.8 nm, 2.5 nm, 3.6 nm, and 7.3 nm thick HfO₂. The error bars denote standard deviations of the mean.

(2) The authors claimed “when an electric pulse is applied to cause oxygen vacancy migration inside the ZnO nanowire, there exists accumulation of oxygen vacancies near the left interface barrier”. If the HfO₂ thickness is less than 3-5 nm, tunneling is possible as the dominant mechanism for interfacial engineering.

In our experiments, we applied electric pulses to stimulate the device firstly, then test its strain-controlled *I-V* characteristics to study the influence of electric pulse on piezotronic effect. There are two processes here.

The first process involves the stimulation of electric pulse on tunneling junctions. The function of the electric pulse is to facilitate the migration of vacancies, resulting in alteration of the tunneling junction barrier. During this process, because the thickness of HfO₂ is about 1.8 nm, there must be some carriers (electrons) tunneling from the electrode (Ag) to the semiconductor (ZnO) to neutralize (or electrostatic shield) the migration of vacancies. In other words, the migration of vacancies and carrier tunneling occur simultaneously, and the electrostatic shielding of the migrated vacancies by carriers is indeed the cause of the band structure bending. The non-uniform electron concentration leads to band restructuring. This carrier tunneling in turn can also indicate that the vacancy migration does drive carriers to neutralize (or electrostatic shield) them and change the interface energy band, which can be analogous to the formation of the Schottky barrier depicted in the Chapter 3 of the

book ‘Physics of Semiconductor Devices’ by S. M. Sze, Y. Li and K. K. Ng¹⁶. It is worth noting that we think there is few or even no vacancy passing through the tunneling junction from ZnO into Ag. Our judgment is based on the recoverability and repeatability of the *I-V* characteristics after electric pulse stimulations (Figure 2f and Figure 3b-d in the manuscript). If the vacancy indeed tunneled under the electric pulse stimulation, then after removing the electric pulse stimulation, the vacancy will not have enough energy to tunnel back to ZnO, and the *I-V* characteristics of the device in Fig. 3d will not fully recover to its original state.

The second process involves the piezotronic effect on the post-stimulated Ag/HfO₂/*n*-ZnO. In this process, the carrier tunneling is also the dominant transport, which is modulated by the strain-induced piezoelectric polarization and the post-stimulation-induced vacancy migration.

Based on above discussion, the carrier tunneling will run through the two processes, but vacancy tunneling should not occur due to the fully reversable and repeatable *I-V* characteristics after electric pulse stimulations (Figure 3b-d). The relevant discussions have been added into the revised manuscript.

In the ‘**Electric pulse-tuned piezotronic effect**’ section in the manuscript, another paragraph is added: “As can be seen, there are two processes in the above experiment. The first process involves the stimulation of electric pulse on the tunneling junction, which drives the vacancy migration inside the *n*-ZnO and thus achieves tunable barrier of the tunneling junction. Because the thickness of HfO₂ is about 1.8 nm, there must be some electrons tunneling from the Ag into the *n*-ZnO to neutralize the migration of vacancies. The second process involves the piezotronic effect on the post-stimulated Ag/HfO₂/*n*-ZnO. During this process, the electron tunneling is the dominant transport, which is modulated by the strain-induced piezoelectric polarization and the post-stimulation-induced vacancy migration. The electron tunneling will run through the two processes. It should be noted that the vacancy will not pass through the tunneling junction, because if the vacancy under the stimulation of electric pulse passes through the tunneling junction, the *I-V* characteristics in Fig. 3d will not fully recover to its original state.”

(3) Or the HfO₂ layer would have some defects that could play an important role in trapping electrons from the ZnO nanowire. Please give more discussion about that.

Defects in HfO₂ can introduce defective surface, which may make a big influence on the tunneling junctions. In order to verify the influence of possible defective surfaces, we characterized the surface of ZnO and conducted capacitance measurements on Ag/HfO₂/n-ZnO tunneling junctions to assess the interface traps formed in the tunneling junction, thereby demonstrating the impact of defects in HfO₂ layer.

SEM analysis, as depicted in **Figure R17a**, reveals that ZnO nanowire had a clean and flat surface. The TEM image in **Figure R17b** depicted the surface lattice images of a single ZnO microwire at various positions with high magnification. The distinct boundary of the high-resolution surface and clear lattice fringe in each inset indicated that ZnO nano/microwire possessed a smooth surface and good crystallinity. Furthermore, **Figure R17c** presented the X-ray diffraction spectra of the ZnO nano/microwires, demonstrating characteristic peaks corresponding to different crystal planes of ZnO. These results prove the good crystalline properties of ZnO nanowires.

In addition, capacitance measurements have been used to quickly assess the impact of interface traps on the tunneling properties of metal-insulator-semiconductor (MIS) tunneling junctions ¹⁶. Here, we conducted *C-V* characterizations on 40 fabricated devices, as illustrated in **Figure R18**. The results displayed two typical *C-V* curves of Ag/HfO₂/n-ZnO tunneling junctions (**Figure R18a**). According to the research and theories of Tamm ¹⁷, Shockley ^{18, 19}, and others regarding interface traps ²⁰ (also historically referred to as interface states, interface defects, surface states, and so on), it is evident that interface traps significantly influence the *C-V* curve, causing it to elongate in the voltage direction ¹⁶. This is attributed to the additional charges required to fill the traps, resulting in a greater total charge or applied voltage needed to achieve the same surface potential or band bending. Furthermore, interface traps also impact the total capacitance of the tunneling junction. With a fixed bias, the presence of interface traps reduces the remaining charge available to be placed in the depletion layer, thereby diminishing the band bending or surface potential. As a result,

the fundamental characteristics of the C - V curve with interface traps shift in both the voltage and capacitance directions, leading to a wider curve opening. Consequently, the blue curve in **Figure R18a** closely resembles the ideal C - V characteristics of an MIS tunneling junction with few interface traps¹⁶, while the red curve, elongated in the voltage direction, represents a device with noticeable interface traps. **Figure R18b** presents the characterization results of the 40 fabricated devices, revealing that only a small percentage (~3.85%) exhibited obvious interface traps.

The process of atomic layer deposition (ALD) of HfO_2 is a self-limiting growth process, with each cycle depositing only one atomic layer or a few atomic layers. Once the surface is occupied by precursor molecules, no further reactions occur, indicating that the growth of the thin film is highly controllable, thus reducing the generation of defects. In this study, the 1.8 nm-thick HfO_2 film deposited by atomic layer deposition is extremely thin, approaching the thickness of a monolayer or a few atomic layers. For the 1.8 nm-thick HfO_2 film, we have rigorously controlled and optimized all steps of the ALD process and used high-purity precursors, therefore, the grown HfO_2 film should have a lower defect density.

Combining the surface quality characterization of ZnO nanowires and the C - V characteristics of devices, we think the influence of interface traps (including interface defects in ZnO and HfO_2) on $\text{Ag}/\text{HfO}_2/n\text{-ZnO}$ can be neglected.

Thank you very much again for this comment, which helps us to understand the interface regulation and possible key factors more deeply. We have revised the manuscript and added above discussion into the revised Supplementary Information (Supplementary Note 3 and Supplementary Figs. 6-8).

The ‘**Working principle**’ section in the manuscript has been revised to: “Based on above principle, The material characterizations (Supplementary Note 3 and Supplementary Figs. 6-8) show that ZnO nanowire possesses a typical hexagonal geometry with clean surface and good wurtzite crystallinity and the interface formed between HfO_2 and ZnO has few interface traps. The piezoelectricity”

Figure R17. Scanning electron microscopy (SEM), transmission electron microscopy (TEM) images, and X-ray diffraction spectra of ZnO microwire. a, b, SEM (a) and TEM (b) images of ZnO microwire used in this work. The insets in (b) exhibit the surface quality of the ZnO microwire at different locations labeled in black box. The scale bars of insets represent 2 nm. c, Corresponding peaks of (100), (002), (101), (102), (110), (112) and (201) planes indicate that the ZnO microwires were highly crystalline in nature.

Figure R18. Influence of interface traps on C-V curves of Ag/HfO₂/n-ZnO tunneling junctions. a, Two typical C-V curves of Ag/HfO₂/n-ZnO devices under strain free condition. The red curve is close to the ideal C-V characteristics of Ag/HfO₂/n-ZnO tunneling junction. The blue curve is stretched out in the voltage direction, which represent the device with obvious interface traps. b, Percentages of 40 devices with few interface traps and obvious interface traps occurred in experiments.

Question #6

Will the recovery time of the interface barrier change after heating? The authors need to give more explanation about the working principle. Additionally, please comment on the difference of interfacial barrier height tuned by the electric-pulse or TENG, which were previously reported.

Response:

We thank you so much for these comments. In order to answer these questions comprehensively and clearly, we divide them as follows (1), (2) and (3), respectively.

(1) Will the recovery time of the interface barrier change after heating?

We conducted relevant experiments to study the influence of heating *n*-ZnO nanowires (post-annealing) on the tunable interface barrier height by electrical pulses. Initially, we heating ZnO nano/microwires in air under various annealing temperature of 200 °C, 300 °C, 400 °C, and 450 °C. Then, these treated nanowires were used to fabricate nanowire devices. As shown in **Figure R19a**, these devices were stimulated by electric pulse with an amplitude of 80 V and a frequency of 10 Hz. **Figures R19b-f** show the *I-V* characteristics of these devices both before and after a 60 s electric pulse stimulation. The measured *I-V* curves indicate that the electric pulse can significantly increase the devices' current responses. **Figure R19g** summarizes the relationship between the current value at +1 V and the annealing temperature. The high increasement in current before electric pulse stimulation indicates that the density of vacancy in *n*-ZnO nanowires increases with the increase of annealing temperature.

Figure R19h plots the current ratio before and after electric pulse as a function of the annealing temperature. As can be seen, the curve reaches its minimized value at 400 °C, indicating that 400 °C is a critical point, beyond which (at 450 °C) the current ratio increases slightly. The decreasing trend of current ratio indicates that with the increase of annealing temperature, the regulation ability of electric pulse tends to be weaker. This phenomenon can be easily understood as follows. As the annealing temperature increases, the density of vacancy as well as the free carrier concentration both increases. However, the increase of carrier concentration will electrostatically shield the vacancies and weaken the regulation of vacancy on the interface barrier.

This phenomenon is similar to the weakened piezotronic effect or the weakened flexoelectronic effect occurred in high doping semiconductors.

It should be noted that the above-mentioned vacancy actually refers to the positively ionized oxygen vacancy (Vo^{2+}), which can be easily transformed into the neutral oxygen vacancies (Vo^0) under annealing temperature above 400 °C²¹. This is why there exists a critical point at 400 °C.

Furthermore, we also plot the recovery time after electric pulse as a function of the annealing temperature in **Figure R19i**. As can be found that the recovery time tends to increase with the increase of annealing temperature and reaches its peak value at 400 °C. This tell us that with the increase of vacancy density, the gradient of vacancy concentration after electric pulse stimulation is relatively small, and it will take more time to recover to its original state. In summary, the recovery time of the interface barrier change increases after heating with the annealing temperature ranged from 0 °C to 200 °C, but slightly decrease from 400 °C to 450 °C. Thereby, appropriate annealing maybe a good method to achieve a long retention of the tunable interface barrier height. Because above discussion can help us to understand the tunable interface barrier height by electric pulse, we added the relevant content in the revised manuscript and the Supplementary Information (Supplementary Note 18 and Supplementary Figs. 36-37).

The ‘**Electric pulse-tuned interface barrier**’ section in the manuscript has been revised to: “In order to clarify In other words, the current will become larger and last longer with a higher electric pulse voltage. Some other ways, such as proper annealing of the piezoelectric semiconductor (*e.g.*, *n*-ZnO), may be also able to achieve a longer retention (Supplementary Note 9 and Supplementary Fig. 17).”

In the ‘**Electric pulse-tuned piezotronic effect**’ section in the manuscript, another paragraph has been added: “We also checked the influence of post-annealing of as-prepared ZnO nanowires on the device performances (Supplementary Note 18). The post-annealing with temperature ranged from 0 °C to 400 °C tends to improve the conductivity of devices, indicating an increased carrier concentration and vacancy density inside the ZnO. As can be seen from Supplementary Figs. 36-37, modulations

on electrical transport by piezoelectric polarization and electric pulse stimulation are both weakened by increasing post-annealing temperature. These weakened modulation result from the stronger electrostatic shielding effect of carriers. Due to the increased vacancy density, the vacancy gradient caused by electric pulse is smaller, so the recovery time is slightly increased with increased annealing temperature. The post-annealing maybe a method to achieve long retention of the modulation by electric pulse.”

Figure R19. Carrier transport characteristics before and after electrical stimulation after annealing. **a**, Electric pulse characterized by an 80 V amplitude and a 10 Hz frequency. **b-f**, I - V characteristics before and after electrical stimulation under different annealing temperature. **g**, The relationship between the current value at a 1 V positive bias and the annealing temperature. **h**, The variations in current ratio before and after electrical stimulation as a function of annealing temperature. **i**, The recovery time of the device's current after electrical stimulation.

(2) The authors need to give more explanation about the working principle.

Thank you for this kindly suggestion. We have revised the ‘**Working principle**’ section in the manuscript as follows:

“The corresponding energy bands of Ag/HfO₂/n-ZnO tunneling junction are schematically depicted in **Fig. 1d**. If the piezoelectric polar *c*-axis points into the piezoelectric semiconductor, the tunneling junction under a compressive strain can utilize the strain-induced positive piezoelectric charge and the corresponding piezoelectric potential at the interface to lower the barrier height (by $\Delta\varphi_{\text{piezo}}/2$) and the barrier width in parallel by making the surface of semiconductor less depleted (Fig. 1d-i), and hence synergistically modulate the electrical transport of interface⁶ (Supplementary Note 5 and Supplementary Figs. 5-6). This synergistically modulation makes the Ag/HfO₂/n-ZnO tunneling junction with high strain sensitivity, which is the advantage of using Ag/HfO₂/n-ZnO tunneling junction here. What needs attention is that the modulation of interface barrier by piezoelectric polarization ranges from $\varphi_0/2$ to $(\varphi_0 - \Delta\varphi_{\text{piezo}})/2$, where $\varphi_0/2$ is the initial interface barrier height. But for a particular tunneling junction, once it is fabricated, the interface barrier is fixed and can’t be tuned, it is more impossible to be continuously tuned. This initially fixed barrier height and uncontrollability are not beneficial to expand the performance and application scenarios of piezotronics. As shown in Fig. 1d-ii, under the driving force induced by electric pulse (pointing to the left), oxygen vacancy will migrate toward the interface from the inside of the semiconductors and then aggregate at the interface. Because the oxygen vacancies are positively ionized, the aggregation of vacancies at the interface will initially lower the interface barrier by $\Delta\varphi_{\text{pulse}}/2$, and thus tune the piezotronic modulation range from $(\varphi_0 - \Delta\varphi_{\text{pulse}})/2$ to $(\varphi_0 - \Delta\varphi_{\text{pulse}} - \Delta\varphi'_{\text{piezo}})/2$, which can control the working current range of the piezotronic tunneling junction. The key working principle here lies in utilizing electric pulse stimulation to achieve **tunable** barrier of tunneling junction and thus regulate the piezotronic effect in the tunneling junction (Supplementary Note 7 and Supplementary Fig. 13).”

(3) Additionally, please comment on the difference of interfacial barrier height tuned

by the electric-pulse or TENG, which were previously reported.

We are grateful for your suggestion. To clarify the mechanism of utilizing electric pulse stimulation to achieve tunable barrier of tunneling junction and thus regulate the piezotronic effect in the tunneling junction, the applied electric pulse needs to be stable and controllable. Although TENG's output voltage can influence the barrier height, but it is not a good stimulation signal to modulate the barrier, especially it is impossible to use it to study the inner mechanism of the modulation. The TENG's output not only depends on the mechanical stimuli (amplitude, frequency, width) but also depends on the external resistance (the resistance of device or interface barrier in this work), leading to inconsistent peak outputs²³⁻²⁵. This instability hinders the precise investigations into the effect of steady pulse amplitude on oxygen vacancies and their impact on the interface barrier. Moreover, the transient nature and limited pulse width of output voltage confine the scope for controllable pulse width modulation. Coupled with a low adjustable frequency range, the TENG is hard to achieve high frequency electric pulse. These reasons make it temporarily difficult to carry out experiments with TENG. Of course, we believe that with the deepening of research, TENG will provide a highly programmable pulse voltage source. Utilizing an external pulse source to generate electric pulse offers enhanced stability, including amplitude consistency and temporal continuity. This approach facilitates the precise control over amplitude, frequency, pulse width, amplitude voltage duration, and pulse period duration, enabling transient and sustained electric pulse. This versatility is instrumental in examining the dynamics of electric pulse stimulation, including the effects of time, frequency, and amplitude, on the modulation of the interface barrier, which is critically important for us to find the typical modulation of piezotronics. And the finding of obvious modulation and the effective technical way for this modulation constitute the important progress. We have revised the manuscript and Supplementary Information (Supplementary Note 10).

The '**Methods**' section in the manuscript has been revised to: "The measuring equipment As an alternative, triboelectric nanogenerator can also be used to input pulse voltage stimulation signal (Supplementary Note 10). The strain of....."

Question #7:

How to achieve a long retention of the tunable barrier height? Could you comment on that? Because the well-controlled interfacial barrier height would be beneficial for the optimization of electronic devices for practical applications.

Response:

We appreciate your insightful comment. Based on the previous study (response to **Question #3** and **#6**) and relevant literatures, we think there are maybe three potential approaches to increase the retention of tunable barrier height.

(1) Appropriately reducing the period time or increasing the duration of electric pulse

As previously discussed in the response to **Question #3**, by reducing the period time of electric pulses, sufficient electric force and energy per unit time can be introduced to drive vacancy migration and redistribution, and increase the response current, leading to a higher change of the barrier height. After the stimulation of electric pulse with small period time, the vacancies require more time to recover to their origin place/state. As a result, a long retention of the tunable barrier height can be achieved by this method.

Increasing the duration of electric pulse is also an effective way to achieve a long retention of tunable barrier height. As shown in **Figure R20**, electric pulse with a same pulse voltage of 40 V but longer duration will lead to a larger response current of the device (**Figure R20b-c**) and a smaller recovery rate of the current after stimulation (**Figure R20d**). In other words, after the stimulation of electric with long duration, the vacancies require more time to recover to their origin place/state. It is important to note that increasing the duration of electric pulse may also have some negative influences, such as increased power consumption, thermal effect, and negative impacts on the long-term stability of the device.

Thereby, appropriately reducing the period time or increasing the duration of electric pulse can achieve a long retention of the tunable barrier height.

(2) Appropriately annealing the semiconductors

Appropriately annealing may adjust the internal structure and properties of the

material, making it possible to achieve a long retention of the tunable barrier height. As previously studies in the **Question #6**, annealing of the *n*-ZnO nanowire in air can increase the density of vacancy and slightly increase the recovery time (**Figure R19**), indicating annealing is a possible method to achieve long retention. Besides, literatures²⁶⁻²⁸ also give some other reasons to use annealing to improve the retention.

- ① Crystal defect repair: during annealing, high temperatures can promote the migration and binding of lattice defects, thereby reducing or repairing crystal defects within the material. This helps improve the crystalline quality and grain boundary conditions, stabilizing the adjustability of the barrier height.
- ② Lattice stress release: proper annealing can help release lattice stress within the material, improving the crystal structure and stability, thus affecting the long-term maintenance capability.
- ③ Interface barrier adjustment: during annealing, the interface properties between semiconductor materials and metals or other semiconductor materials can be adjusted, affecting the quality of the barrier formation. This contributes to the long-term stability of the barrier height. Thereby, appropriately annealing can improve the structure and properties of semiconductors, possessing possibility to achieve a long retention of the tunable barrier height.

(3) Continuously applying a small electric pulse

After the stimulation of electric pulse with high amplitude, continuously applying a small electric pulse (different from the former electric pulse with high amplitude) together with the sweeping voltage (used to measure the *I-V* characteristics) may maintain the vacancy redistribution. In theory, the form of small electric pulse can be easily distinguished from the sweeping voltage, thereby the measured current containing a DC part and a pulse part can be also distinguished. As a result, the applied small electric pulse to maintain the vacancy redistribution and the tunable interface barrier height will not affect the *I-V* characteristics of the device. This method should be feasible in theory, but a large number of experiments are needed to find the appropriate amplitude, frequency and duration of the small electric pulse, which needs further experimental demonstration in the future.

We again express our grateful to you. It guides us to further consider how to

achieve long retention of the tunable barrier height, which can highly improve the importance of electric pulse-tuned piezotronic effect. This part of discussion may have some inspirations, particularly the third point (‘Continuously applying a small electric pulse’). So, we put them in the revised Supplementary Information (Supplementary Note 9).

The ‘**Electric pulse-tuned interface barrier**’ section in the manuscript has been revised to: “**Electric pulse-tuned interface barrier.** In order to clarify..... In other words, the current will become larger and last longer with a higher electric pulse voltage. Some other ways, such as proper appropriately annealing of the piezoelectric semiconductor (*e.g.*, *n*-ZnO), may be also able to achieve a longer retention (Supplementary Note 9 and Supplementary Fig. 17).”

Figure R20. Current response and recovery characteristics regulated by the electric pulse duration. **a**, Electric pulse voltage of 40 V on Ag/HfO₂/n-ZnO device with the duration of each pulse of 0.05 s, 0.36 s and 2.35 s, respectively, and the time interval between two electric pulses is about 10.96 s. **b**, **c**, Current response corresponding to (**a**). **d**, Three current recovery curves numbered ‘1’ to ‘3’ in (**c**), in which the recovery slopes are -0.07759, -0.04626 and -0.0249, respectively.

Reference:

1. Zhang, S. *et al.* Strain-controlled power devices as inspired by human reflex. *Nature Communications* **11**, 1-9 (2020).
2. Wu, W., Wen, X., Wang, Z. L. Taxel-addressable matrix of vertical-nanowire piezotronic transistors for active and adaptive tactile imaging. *Science* **340**, 952-957 (2013).
3. Pan, C. *et al.* High-resolution electroluminescent imaging of pressure distribution using a piezoelectric nanowire LED array. *Nature Photonics* **7**, 752-758 (2013).
4. Jiang, F. *et al.* Ferroelectric modulation in flexible lead-free perovskite Schottky direct-current nanogenerator for capsule-like magnetic suspension sensor. *Advanced Materials* **35**, 2302815 (2023).
5. Dong, J. *et al.* A high voltage direct current droplet-based electricity generator inspired by thunderbolts. *Nano Energy* **90**, 106567 (2021).
6. Chen, J. *et al.* Self-rectifying magnetoelectric metamaterials for remote neural stimulation and motor function restoration. *Nature materials* **23**, 39-146 (2024).
7. Wu, W., Wei, Y., Wang, Z. L. Strain-gated piezotronic logic nanodevices. *Advanced materials* **22**, 4711-4715 (2010).
8. Yu, R. *et al.* GaN nanobelt-based strain-gated piezotronic logic devices and computation. *ACS Nano* **7**, 6403-6409 (2013).
9. Dan, M. *et al.* High performance piezotronic logic nanodevices based on GaN/InN/GaN topological insulator. *Nano Energy* **50**, 544-551 (2018).
10. Yu, R., Pan, C., Wang, Z. L. High performance of ZnO nanowire protein sensors enhanced by the piezotronic effect. *Energy & Environmental Science* **6**, 494-499 (2013).
11. Cao, X. *et al.* Piezotronic effect enhanced label-free detection of DNA using a Schottky-contacted ZnO nanowire biosensor. *ACS Nano* **10**, 8038-8044 (2016).
12. Bai, S. *et al.* High-performance integrated ZnO nanowire UV sensors on rigid and flexible substrates. *Advanced Functional Materials* **21**, 4464-4469 (2011).
13. Yang, X. *et al.* Piezophototronic-effect-enhanced electrically pumped lasing. *Advanced Materials* **29**, 1602832 (2017).

14. Guo, D. *et al.* Silicon flexoelectronic transistors. *Science Advances* **9**, eadd3310 (2023).
15. Sun, L. *et al.* Mechanical manipulation of silicon-based Schottky diodes via flexoelectricity. *Nano Energy* **83**, 105855 (2021).
16. Sze, S. M. & K, K. N. *Physics of semiconductor devices*, 3rd Edition. (Wiley, 2006).
17. Tamm, I. Über eine mögliche art der elektronenbindung an kristalloberflächen. *Zeitschrift für Physik* **76**, 849-850 (1932).
18. Shockley, W., Pearson, G. Modulation of conductance of thin films of semiconductors by surface charges. *Physical Review* **74**, 232 (1948).
19. Shockley, W. On the surface states associated with a periodic potential. *Physical Review* **56**, 317 (1939).
20. Nicollian, E., Brews, J. *MOS physics and technology*, John Wiley & Sons (1982).
21. Li, G. *et al.* Adjustment of oxygen vacancy states in ZnO and its application in ppb-level NO₂ gas sensor. *Science Bulletin* **65**, 1650-1658 (2020).
22. Meng, J. *et al.* Triboelectric nanogenerator enhanced Schottky nanowire sensor for highly sensitive ethanol detection. *Nano Letters* **20**, 4968-4974 (2020).
23. Kim, W. G. *et al.* Triboelectric nanogenerator: structure, mechanism, and applications. *ACS Nano* **15**, 258-287 (2021).
24. Zhang, S. L. *et al.* Electromagnetic pulse powered by a triboelectric nanogenerator with applications in accurate self-powered sensing and security. *Advanced Materials Technologies* **5**, 2000368 (2020).
25. Lone, S. A. *et al.* Recent advancements for improving the performance of triboelectric nanogenerator devices. *Nano Energy* **99**, 107318 (2022).
26. Won, D. *et al.* Transparent electronics for wearable electronics application. *Chemical Reviews* **123**, 9982-10078 (2023).
27. Xu, X. *et al.* Status and prospects of MXene-based nanoelectronic devices. *Matter* **6**, 800-837 (2023).
28. Han, X., Ji, Y., Yang, Y. Ferroelectric photovoltaic materials and devices. *Advanced Functional Materials* **32**, 2109625 (2022).

Reviewer #3 (Remarks to the Author):

In this work, Qin *et al.* reported in depth investigation of modulation of interface engineering using electrical pulse by the piezotronic effect. They prove that the proposed model is valid for the piezotronic device using literature data. Therefore, I recommend its publication, with suggested major modifications, as below:

Response:

We highly appreciate you for approving the importance of our work and your detailed and constructive suggestions. And we have revised and improved the manuscript according to your important comments.

Question #1

The author should show the high-resolution cross-section image of the Ag/HfO₂/ZnO junction.

Response:

Thank you so much for this suggestion. We have provided the high-resolution cross-section image of the Ag/HfO₂/ZnO junction into the Supplementary Information (Supplementary Note 3 and Supplementary Fig. 7), and revised the manuscript accordingly.

The scanning electron microscopy (SEM) image in **Figure R21a-(i)** shows the clean surface and hexagonal geometry of the ZnO nanowire. Analysis of the high-resolution transmission electron microscopy (HRTEM) image in **Figure R21a-(ii)** reveals that the deposited HfO₂ layer on ZnO surface has a thickness of approximately 1.8 nm. In order to obtain the cross-section image of the tunneling junction, we used a small tool knife to cut the end of the ZnO microwires deposited by HfO₂ and Ag. **Figure R21b** shows the high-resolution SEM image of the Ag/HfO₂/n-ZnO junction, revealing the hexagonal cross-section of ZnO and the Ag electrode coating. Additionally, **Figure R21c** presents an enlarged cross-section image of the junction. The thickness differences between the HfO₂ insulation layer (only 1.8 nm) and the Ag electrode make it difficult to clearly observe the HfO₂ layer on the nanowire surface. Following this, we conducted an EDX elemental mapping analysis based on the high-

resolution cross-section image in **Figure R21d**. The EDX element mappings for Zn, O, Hf, and Ag in **Figure R21e**, allowing us to observe the distribution of the Hf element at the boundary between Ag and ZnO.

The ‘**Working principle**’ section in the manuscript has been revised to: “Based on above principle, The material characterizations (Supplementary Note 3 and Supplementary Figs. 6-8) show that the ZnO nanowire possesses a typical hexagonal geometry with clean surface and good wurtzite crystallinity, and the interface formed between HfO₂ and ZnO has few interface traps.”

Figure R21. Crosse-section TEM and SEM images of Ag/HfO₂/n-ZnO junctions and the corresponding EDX element mapping image. a-i, SEM image of the ZnO microwire with a diameter of about 8 μm, showing a typical hexagonal geometry and a clean surface. a-ii, High-resolution TEM image of HfO₂ layer with a thickness of about 1.8 nm deposited on ZnO surface. b, Crosse-section view SEM images of an Ag/HfO₂/n-ZnO junction. c, Enlarged version of the cross-section SEM image of the Ag/HfO₂/n-ZnO junction in (b). d, High-resolution cross-section view SEM image of the Ag/HfO₂/n-ZnO junction. e, The corresponding EDX element mapping image of Zn, Ag, O and Hf in (d).

Question #2

There are many insulator layers, why the author used HfO₂ insulator layer for device fabrication. What is the thickness of the HfO₂ layer.

Response:

We thank you very much for the important question and suggestion. We will briefly discuss the reason for choosing HfO₂ as insulation layer and given the thickness (~1.8 nm) of the HfO₂ layer. And the relevant details have also been added into the revised manuscript (also listed at the end of this response for your convenience) and the revised Supplementary Information (Supplementary Note 2, and Supplementary Figs. 1-4 and Supplementary Table 1).

(1) There are many insulator layers, why the author used HfO₂ insulator layer for device fabrication.

Here, we combine the related literature reports and our further experimental results about the influence of insulation materials on the performance of piezotronic tunneling junction to explain our choice of HfO₂.

At the beginning, we chose HfO₂ as the insulation layer due to its advantages in terms of high dielectric constants (k) and wide band gap (E_g). Many literatures report that the dielectric insulation layer with high k values and wide band gaps not only can reduce the leakage current and the overall power consumption during device operation, but also has good stability to withstand breakdown. As compared with SiO₂, Al₂O₃, MgO and some other insulation materials, HfO₂ has advantages in terms of k value and band gap, as shown in **Table R2**¹⁻⁵. So, we chose HfO₂ as the dielectric insulation layer of the piezotronic tunneling junction.

Then, we studied the influence of different insulation materials on the performance of piezotronic tunneling junction. Detailly, we measured the strain sensing capabilities of devices with insulation layers of HfO₂, ZrO₂, Al₂O₃ and SiO₂, each with a thickness of ~1.8 nm, under tensile strains of 0.00%, 0.033%, 0.067%, and 0.10%. **Figure R22** presents the current values at a +3.0 V bias for twenty devices of each type, revealing that at 0.00% strain, the currents across the four device types varied randomly between 0.01 and 7 nA, and at 0.10% strain, the variation was

between 30 and 250 nA. Analysis of **Figure R22** allows us to calculate and compare the current on/off ratios and the changes in Schottky barrier height for each device type under 0.10% strain, as summarized in **Figure R23**. The results indicate that the insulation materials (HfO₂, ZrO₂, Al₂O₃, and SiO₂) have almost no influence on the strain sensing performance of piezotronic tunneling devices.

In order to understand this phenomenon, we write the classical carrier transport of metal-insulator-semiconductor (MIS) tunneling junction here ⁶:

$$I = SA^*T^2 \exp(-\alpha_T d \sqrt{q\varphi_T}) \exp\left(-\frac{q\varphi_B}{kT}\right) \left[\exp\left(\frac{qV}{nkT}\right) - 1 \right] \quad (1)$$

where $\exp(-\alpha_T d \sqrt{q\varphi_T})$ is the tunneling probability term, S is the contact area of the tunneling barrier, A^* is the effective Richardson constant, q represents the charge of an electron, k is the Boltzmann constant, T is the absolute temperature, φ_B represents the height of the Schottky barrier, n is the ideality factor, d is the thickness of the insulation layer in the tunneling junction, $\alpha_T = 2\sqrt{2qm^*}/\hbar$ (m^* represents the effective mass of the charge carriers, \hbar is the reduced Planck constant), φ_T represents the effective height of the tunneling barrier. It can be seen that the tunneling current is mainly dependent on the thickness of insulator layer (d) and the Schottky barrier height (φ_B) (which is determined by the metal's work function and the semiconductor's electron affinity), and slightly dependent on the band gap of insulator layer (which together with its electron affinity will affect the effective height of the tunneling barrier $q\varphi_T$), and almost not dependent on the k value. Given that the band gaps of HfO₂, ZrO₂, Al₂O₃ and SiO₂ are not very different, the piezotronic modification of tunneling current has little to do with the choice of insulation materials, which is consistent with our experimental results in **Figures R22** and **R23**.

Based on above reasons, we chose HfO₂ as the insulation layer for convenience, which can be precisely fabricated by atomic layer deposition (ALD). Actually, other dielectric materials including ZrO₂, Al₂O₃ and SiO₂ can also be candidates as insulation layer. We thank you very much for this comment again.

Table R2. Dielectric constants and band gaps of typical insulation materials ¹⁻⁵.

Insulation materials	Dielectric constants k	Band gaps E_g (eV)
--------------------------	----------------------

HfO ₂	20~25	5.6~6.0
ZrO ₂	17~25	5.1~7.8
SiO ₂	3.9	9
Si ₃ N ₄	7~7.5	5~5.3
Al ₂ O ₃	9	8.8
MgO	8.4~9.8	8.7~8.8

Figure R22. Influence of insulation materials on piezotronic modification of the carrier transport of piezotronic tunneling devices. a-d, Piezotronic modification of the carrier transports of four types of piezotronic tunneling devices using HfO₂ (a), ZrO₂ (b), Al₂O₃ (c) and SiO₂ (d) as insulation layers under the conditions of tensile strain of 0.00%, 0.033%, 0.067% and 0.10%, respectively. The above currents were measured at bias of +3.0 V.

Figure R23. Sensing performance of piezotronic tunneling devices. a, b, On/off ratio (a) and change of effective Schottky barrier height (b) of the devices based on

piezotronic tunneling junction with different insulation materials of HfO₂, ZrO₂, Al₂O₃ and SiO₂. The error bars denote standard deviations of the mean.

(2) What is the thickness of the HfO₂ layer.

In our experiments, an HfO₂ insulator layer with a thickness of about 1.8 nm was deposited on ZnO nanowire by ALD. We used high-resolution transmission electron microscopy (HTEM) to characterize the thickness of HfO₂ layer. The scanning electron microscopy (SEM) image (**Figure R24a-(i)**) shows the ZnO nanowire with a hexagonal geometry, and HTEM image of HfO₂/n-ZnO (**Figure R24a-(ii)**) shows a HfO₂ thickness of about 1.8 nm on ZnO surface. In **Figure R24b**, the distribution of HfO₂ thickness along the *x*-axis labeled in **Figure R24a-(i)** shows that the HfO₂ is uniform with a thickness of about 1.8 (± 0.3) nm. The inset shows the statistical distribution of the insulation thickness.

Here, we briefly explain why the thickness of 1.8 nm was chosen. We have fabricated Ag/HfO₂/n-ZnO devices with HfO₂ thicknesses of 0.0 nm, 0.4 nm, 1.1 nm, 1.8 nm, 2.5 nm, 3.6 nm, and 7.3 nm, and then studied the piezotronic modifications of electrical transport of 140 devices (20 devices of each thickness) under 0.10% tensile strain. The experimental results show that the device with 1.8 nm HfO₂ exhibit the highest current on/off ratio (**Figure R25**), which makes us choose 1.8 nm HfO₂ as the insulator layer of piezotronic tunneling junctions.

So, we chose HfO₂ with a thickness of ~1.8 nm as the insulation layer to fabricate the piezotronic tunneling junction.

Figure R24. Morphology of HfO₂/n-ZnO microwire and thickness of deposited

HfO₂ layer. a-i, SEM image of the ZnO microwire with a diameter of about 8 μm , showing a typical hexagonal geometry and a clean surface. **a-ii**, High-resolution TEM image of HfO₂ layer deposited on ZnO surface, showing a thickness of 1.8 nm. **b**, Thickness of HfO₂ along the x -axis labeled in (a-i), showing the HfO₂ thickness of about 1.8 (\pm 0.3) nm. The inset shows the statistical distribution of the HfO₂ thickness.

Figure R25. Current on/off ratio of Ag/HfO₂/ n -ZnO devices with 0.0 nm, 0.4 nm, 1.1 nm, 1.8 nm, 2.5 nm, 3.6 nm, and 7.3 nm thick HfO₂. The error bars denote standard deviations of the mean.

The ‘**Working principle**’ section in the manuscript has been revised to: “Based on above principle, Here, we choose HfO₂ with a thickness of about 1.8 nm as the insulation layer and Ag as the electrode to fabricate the Ag/HfO₂/ n -ZnO PTJ for this structure possessing good performance on piezotronic modification of electrical transport (Supplementary Note 2, and Supplementary Figs. 1-4 and Supplementary Table 1). The material.....”

The ‘**Methods**’ section in the manuscript has been revised to: “The grown process of HfO₂ Finally, controllable preparation of HfO₂ insulation layer of about 1.8 nm can be realized. Since the PTJ with a 1.8 nm HfO₂ possesses the best piezotronic modification of electrical transport, the thickness of about 1.8 nm is chosen. In addition, our further experiments indicate that the HfO₂ can also be replaced with Al₂O₃, SiO₂, and ZrO₂. More detailed discussions can be found in the Supplementary Note 2, Supplementary Figs. 1-4 and Supplementary Table 1.”

Question #3

Why author measured the *I-V* characteristics (Fig. 2e) at low voltages. Did author measure higher (>1 V) voltages or not. If yes, please include it in the manuscript. If not, please explain the reason.

Response:

We highly appreciate you for this in-depth comment. In piezotronics, high voltage will lead to the degrading of piezotronic modification, so *I-V* characteristics of piezotronic devices are typically measured at low sweeping voltage. Also, the objective of performing *I-V* measurements is to evaluate the impact of piezotronic effect before and after electric pulse. The *I-V* curves measured at low voltage fulfill the research requirements. Moreover, high sweeping voltage may lead to damage to the interface barrier. Therefore, we only measured the *I-V* characteristics at low voltages. Additional experiments and explanation are as follows.

Figure R26 shows the comparison of strain-controlled *I-V* characteristics at different sweeping voltages (± 1 V, ± 3 V, ± 5 V, ± 7 V and ± 9 V). It shows an increased current under both strains of 0.00% and -0.10% as the sweeping voltage increases, indicating low interface barrier height under a high sweeping voltage. Because the interface barrier almost disappears under high sweeping voltage, the piezotronic effect is weakened. As shown in **Figure R26e**, the *I-V* curves for 0.00% and -0.10% almost overlap with each other when the sweeping voltage is ± 9 V. **Figure R26f** summarized the current on-off ratio as a function of the sweeping voltage, indicating a degraded piezotronic effect at high sweeping voltage. **Figure R27** shows the *I-V* curves at ± 1 V and ± 5 V sweeping voltages after electric pulse (80 V and 10 Hz) with durations of 0 s, 30 s, 60 s and 120 s, respectively. Under ± 1 V sweeping voltage, the current gradually rises with more time stimulation (**Fig. R27a**). While, under ± 5 V sweeping voltage, the current increases in the range of 0-60 s but suddenly decreases at 120 s (**Fig. R27b**). This indicates that high sweeping voltage testing can potentially induce damage to the interface.

The above results suggest that higher sweeping voltages are not suitable for studying the tuning of electric pulses on piezotronics. So, we opted to examine the *I-V*

characteristics at lower voltages. Relevant discussions have been added into the revised manuscript and Supplementary Information (Supplementary Note 12 and Supplementary Figs. 24-25).

The ‘**Electric pulse-tuned interface barrier**’ section in the manuscript has been revised to: “In **Fig. 2e**, we studied As the electric pulse is stopped, the current gradually decreases to its original magnitude due to the diffusion of oxygen vacancy back to its original state. Moreover, by increasing the pulse frequency (cycle number per second) and the pulse duration, we can further improve the impact of electric pulse on the current response and increase the recovery time (Supplementary Note 10 and Supplementary Fig. 21). It is worth noting that we used sweeping voltages of ± 1 V here and in subsequent *I-V* characteristics, because higher sweeping voltage may degrade the piezotronic effect and cause the damage of tunneling junction (Supplementary Note 12 and Supplementary Figs. 24-25).”

Figure R26. Comparison of *I-V* characteristics at different sweep voltage of Ag/HfO₂/n-ZnO device. a-e, The *I-V* curves is measured at different sweep voltage of ± 1 V (a), ± 3 V (b), ± 5 V (c), ± 7 V (d) and ± 9 V (e), respectively, under the compressive strain of 0.00% and 0.10%. **f,** Relationship between current on/off ratio and sweep voltage.

Figure R27. *I-V* characteristics under electric pulse stimulations with different duration at sweep voltages of ± 1 V and ± 5 V. a, b, *I-V* curves under electric pulse stimulations for 0 s, 30 s, 60 s and 120 s measured at sweep voltages of ± 1 V (a) and ± 5 V (b), respectively. The electrical pulse used has a frequency of 10 Hz, and the amplitude is set at 80 V.

Question #4

Discussion needs about the fundamental understanding of piezotronics and why the contact the author used is Ag.

Response:

Thank you very much for this suggestion.

(1) Discussion needs about the fundamental understanding of piezotronics.

Piezotronics is a field that explores the interaction/coupling between piezoelectric polarization and semiconductor properties (**Figure R28**). When a mechanical stimulus is applied to a piezoelectric semiconductor, it can produce piezoelectric polarization at interfaces and thus modulate the energy band and carrier transport of interfaces, leading to the development of new types of sensors, and electronic devices.

Here, we consider the *n*-type piezoelectric semiconductor as a representative example and, for the sake of simplicity, disregarding surface states, shielding effects, and other crystal anomalies, while maintaining the junctions at zero bias. **Figure R29** shows the band diagrams of metal-semiconductor (MS) and metal-insulation-semiconductor (MIS) contact regulated by piezoelectric polarization charges. Negative piezoelectric polarization charges repel electrons at the MS interface,

leading to further depletion of the depletion region and an increase in the Schottky barrier height (**Figure R29a-i**). Conversely, positive piezoelectric polarization charges attract electrons to the MS interface, thereby reducing the depletion region width and the Schottky barrier height (**Figure R29a-ii**). The magnitude and polarity of the piezoelectric potential in the piezoelectric semiconductor depend on the crystal orientation and the magnitude and polarity of the applied stress/strain. The above considerations form the basis for the regulation of the interface energy band by piezoelectric polarization charges, also known as the piezotronic effect, which has been extensively demonstrated in wurtzite-structured semiconductors (such as ZnO, GaN, etc.)⁷⁻¹⁰. The MIS contact can be achieved by inserting an insulation layer between the metal and semiconductor. Similarly, the piezoelectric potential induced by the piezoelectric polarization charges also exerts a modulation effect on the interfacial energy band, thus influencing carrier transport (**Figure R29b**).

According to your suggestion, we have added the above introduction of piezotronics into the revised Supplementary Information (Supplementary Note 7), and revised the ‘**Introduction**’ and ‘**Working principle**’ section in the manuscript to make piezotronics easier to understand.

The ‘**Introduction**’ section in the manuscript has been revised to: “The fundamental principle **Fig. 1a** illustrates the typical piezotronics, and the corresponding energy bands at interfaces modulated by strain-induced piezoelectric polarization. When a strain is applied on the piezoelectric semiconductor, piezoelectric charges and the corresponding piezoelectric potential will appear at the interface, which will bend the energy band and thus control the electrical transport of the interface. As can be seen, the construction of interface barrier is the key in achieving interface engineering by local polarization³.....”

The ‘**Working principle**’ section in the manuscript has been revised to: “If the piezoelectric polar *c*-axis points into the piezoelectric semiconductor, the tunneling junction under a compressive strain can utilize the strain-induced positive piezoelectric charge and the corresponding piezoelectric potential at the interface to lower the barrier height (by $\Delta\varphi_{\text{piezo}}/2$) and the barrier width in parallel by making

the surface of semiconductor less depleted (Fig. 1d-i), and hence synergistically modulate the electrical transport of interface⁶.....”

Figure R28. Coupling the piezoelectricity and the semiconducting property resulting in the emerging field of piezotronics.

Figure R29. Band diagrams of metal-semiconductor (MS) and metal-insulator-semiconductor (MIS) contact regulated by piezotronic effect. a, b, Band diagrams depicting the contact between MS (a) and MIS (b) regulated by piezoelectric polarization. The presence of stress induces positive and negative piezoelectric polarization charges at the MS and MIS interface, which can decrease and increase the barrier height, respectively.

(2) Why the contact the author used is Ag.

There are two reasons for us to use silver (Ag) as electrodes. Firstly, Ag possesses a relative high work function and can form Schottky contacts with ZnO. As long as the electrode can form a Schottky barrier with ZnO, then the metal-insulation-semiconductor (MIS) tunneling junction can be well constructed. So, Ag meets the requirements. Secondly, in the past ten years, Ag has been widely utilized to fabricate Schottky contacts particularly with ZnO in piezotronic devices⁷⁻¹⁴. Therefore, we selected Ag as electrodes in constructing piezotronic tunneling junction for convenience. This section has been added to the revised Supplementary Information (Supplementary Note 2).

The ‘**Methods**’ section in the manuscript has been revised to: “.....Then, using the RZF-300 thermal evaporation coating system, the Ag electrode as one of the most widely used electrodes for constructing interface barrier in piezotronics is prepared at both ends of the nanowire/microwire by the thermal evaporation method.....”

Question #5

In this paper, the PDMS is piezoelectric ceramics materials, the authors should discuss whether the PDMS coating has influence on the performance of the Ag/HfO₂/ZnO junction.

Response:

Thank you very much for this comment. Generally, pure polydimethylsiloxane (PDMS) is a polymer classified as a silicon elastomer, which usually exhibits no piezoelectric effect. And it is widely used for encapsulating electronics particularly flexible electronics. For example, in the past ten years, most of piezotronic devices are encapsulated by PDMS. In this work, PDMS is used to encapsulate the piezotronic tunneling junction, protecting it from damage during bending. Pure PDMS exhibits low elastic modulus and strength tunable by curing conditions^{15, 16}. Critically, the PET substrate (Young's modulus 3-3.5 GPa) has far greater stiffness than the PDMS film (modulus 360-870 kPa). This enables assuming the nanowire deforms identically to the PET.

We further conducted finite element method simulations under matching experimental conditions to study the possible influence of PDMS encapsulation on performance of the device. Given the micron-scale ZnO nanowire diameter versus millimeter substrate/encapsulation thicknesses, radial nanowire effects are reasonably neglected. Strain loading is modeled by applying proper boundary conditions at the PET substrate neutral layer, inducing bending while preserving neutral layer length. In **Figure R30a**, the FEM model is established with a ZnO nanowire (1 mm in length) that is on the central of PET substrate (0.15 mm in thickness and 30 mm in length). And, a PDMS with thickness 0.5 mm is introduced in a typical model; whereas there is no PDMS structure in the other one for comparison. Here, we list the main parameters of the material properties for the mechanical simulation using the material library database in the software of COMSOL Multiphysics (**Table R3**).

It could be clearly observed that the upper part of the piezotronic device is gradually compressed as the increasing curvature of the substrate, as demonstrated by the displacement distribution profiles under different bending conditions in **Figure R30b** and **R30c**. **Figure R30d** and **R30e** show the corresponding outlines of the bended nanowires obtained from **Figure R30b** and **R30c**, while the calculated strain of the ZnO nanowire in the piezotronic device with and without the PDMS layer under the same loading condition is plotted in **Figure R30f**. As can be seen, the existence of PDMS layer would have little influence on the ZnO nanowire deformation when the substrate bends, since the nanowire strain might be identical for the piezotronic device with and without PDMS layer (**Figure R30f**).

Therefore, as expected, the PDMS encapsulation would not affect the sensing performance of the device. Relevant details have been added into the revised manuscript and Supplementary Information (Supplementary Note 2, Supplementary Fig. 5 and Supplementary Table 2).

The ‘**Methods**’ section in the manuscript has been revised to: “Finally, a thin layer of polydimethylsiloxane (PDMS) is used to package the whole device to prevent ZnO oxidation and device damage under repeated mechanical strain. Our simulations show that the PDMS has little impact on the performance of the piezotronic tunneling

junction, which is the reason we chose PDMS as the encapsulation layer. (Supplementary Note 2, Supplementary Fig. 5 and Supplementary Table 2).”

Figure R30. Impact of the PDMS encapsulation layer on the performance of the piezotronic tunneling device by FEM simulation. **a**, Schematic illustration of the side view of the piezotronic tunneling device with (top) and without (bottom) PDMS encapsulation layer. **b-e**, Displacement distribution profiles (**b**, **c**) and the corresponding outlines of bended nano/microwires (**d**, **e**) of the piezotronic tunneling devices with (**b**, **d**) and without (**c**, **e**) PDMS layer in response to mechanical strain, respectively. **f**, Calculated ZnO nano/microwire strain as a function of the distance change between the two ends of the substrate in the piezotronic device with and without the PDMS layer, respectively.

Table R3. Material parameters in the finite element method (FEM) simulation.

Materials	Young's modulus, E	Poisson's ratio, ν
PET	4 GPa	0.35
PDMS	750 kPa	0.49
ZnO	210 GPa	0.33

References:

1. Nahar, R., Singh, V., Sharma, A. Study of electrical and microstructure properties of high dielectric hafnium oxide thin film for MOS devices. *Journal of Materials Science: Materials in Electronics* **18**, 615-619 (2007).
2. Gilmer, D. *et al.* Compatibility of polycrystalline silicon gate deposition with HfO₂ and Al₂O₃/HfO₂ gate dielectrics. *Applied Physics Letters* **81**, 1288-1290 (2002).
3. Kang, S. *et al.* Effect of deposition conditions of poly Si_{1-x}Ge_x films and Ge atoms on the electrical properties of poly Si_{1-x}Ge_x ($x= 0, 0.6$)/HfO₂ gate stack. *Journal of Applied Physics* **94**, 4608-4613 (2003).
4. Hinkle, C. *et al.* Enhanced tunneling in stacked gate dielectrics with ultra-thin HfO₂ (ZrO₂) layers sandwiched between thicker SiO₂ layers. *Applied Surface Science* **234**, 240-245 (2004).
5. Gerritsen, E. *et al.* Evolution of materials technology for stacked-capacitors in 65 nm embedded-DRAM. *Solid-State Electronics* **49**, 1767-1775 (2005).
6. Sze, S. M. & K, K. N. *Physics of semiconductor devices*, 3rd Edition. (Wiley, 2006).
7. Pan, C., Zhai, J., Wang, Z. L. Piezotronics and piezo-phototronics of third generation semiconductor nanowires. *Chemical Reviews* **119**, 9303-9359 (2019).
8. Wang, L., Wang, Z. L. Advances in piezotronic transistors and piezotronics. *Nano Today* **37**, 101108 (2021).
9. Zhu, L., Wang, Z. L. A perspective on piezotronics and piezo-phototronics based on the third and fourth generation semiconductors. *Applied Physics Letters* **122**, 250501 (2023).
10. An, C. *et al.* Piezotronic and piezo-phototronic effects of atomically-thin ZnO nanosheets. *Nano Energy* **82**, 105653 (2021).
11. Zhao, X. *et al.* Recent progress in ohmic/Schottky-contacted ZnO nanowire sensors. *Journal of Nanomaterials* **2015**, 7-7 (2015).
12. Cao, X. *et al.* Piezotronic effect enhanced label-free detection of DNA using a Schottky-contacted ZnO nanowire biosensor. *ACS Nano* **10**, 8038-8044 (2016).

13. Hu, W., Zhang, C., Wang, Z. L. Recent progress in piezotronics and triboelectronics. *Nanotechnology* **30**, 042001 (2018).
14. Meng, J., Li, Z. Schottky-contacted nanowire sensors. *Advanced Materials* **32**, 2000130 (2020).
15. Moučka, R. *et al.* Mechanical properties of bulk Sylgard 184 and its extension with silicone oil. *Scientific Reports* **11**, 19090 (2021).
16. Peng, J. *et al.* Stiff and tough PDMS-MMT layered nanocomposites visualized by AIE luminogens. *Nature Communications* **12**, 4539 (2021).

REVIEWERS' COMMENTS

Reviewer #1 (Remarks to the Author):

I am now happy with the detailed response of my queries. I recommend this manuscript for the publication.

Reviewer #2 (Remarks to the Author):

The authors have addressed all my concerns. It would be accepted.

Reviewer #3 (Remarks to the Author):

The author has addressed all of my concerns, I recommend publications.

Point-to-Point Response to Reviewer's comments:

We thank the reviewer for the in-depth reviews and important/constructive questions/suggestions regarding our manuscript.

Reviewer #1 (Remarks to the Author):

I am now happy with the detailed response of my queries. I recommend this manuscript for the publication.

Response:

We highly appreciate you for your detailed, very positive, and valuable comments regarding our manuscript during the review process. These important comments and suggestions, along with many other valuable details, have significantly helped us to improve the manuscript. We thank you again for the time and effort on improving the quality of our manuscript.

Reviewer #2 (Remarks to the Author):

The authors have addressed all my concerns. It would be accepted.

Response:

We thank you very much for your in-depth, responsible and valuable reviewing of our manuscript during the review process. By incorporating your comments and suggestions, we have highly enhanced the quality of the paper, making it more readable and comprehensible. This process has also enabled us to gain a deeper understanding of our own work. Once again, we extend our sincere appreciation to you for the time and effort on helping us to improve the quality of our manuscript.

Reviewer #3 (Remarks to the Author):

The author has addressed all of my concerns, I recommend publications.

Response:

We appreciate you very much for your very constructive and valuable comments during the review process. Your insights and suggestions have been immensely helpful to us. By incorporating your recommendations, we have significantly

enhanced the quality of this work, particularly in terms of readability and clarity. We would like to express our sincere thanks to you for dedicating the time and effort on helping us to improve the quality of our manuscript.